# Advanced error diagnostics of the CMAQ and Chimere modelling systems within the AQMEII3 model evaluation framework

Efisio Solazzo[1], Christian Hogrefe[2], Augustin Colette[3], Marta Garcia-Vivanco[3,4], Stefano Galmarini[5]

[1] European Commission, Joint Research Centre (JRC), Directorate for Energy, Transport and Climate, Air and Climate Unit, Ispra (VA), Italy
[2] Atmospheric Model Application and Analysis Branch - Computational Exposure Division - NERL, ORD, U.S. EPA
[3] INERIS, Institut National de l'Environnement Industriel et des Risques, Parc Alata, 60550 Verneuil-en-Halatte, France
[4] CIEMAT, Avda Complutense 40, Madrid, Spain
[5] European Commission, Joint Research Centre (JRC), Directorate for Sustainable Resources, Food and Security Unit, Ispra (VA), Italy

**Abstract.** The work here complements the overview analysis of the modelling systems participating in the third phase of the Air Quality Model Evaluation International Initiative (AQMEII3) by focusing on the performance for hourly surface ozone by two modelling systems, Chimere for Europe and CMAQ for North America.

The evaluation strategy outlined in the course of the three phases of the AQMEII activity, aimed to build up a diagnostic methodology for model evaluation, is pursued here and novel diagnostic methods are proposed. In addition to evaluating the 'base case' simulation in which all model components are configured in their standard mode, the analysis also makes use of sensitivity simulations in which the models have been applied by altering and/or zeroing lateral boundary conditions, emissions of anthropogenic precursors, and ozone dry deposition.

To help understand of the causes of model deficiencies, the error components (bias, variance, and covariance) of the base case and of the sensitivity runs are analysed in conjunction with time-scale considerations and error modelling using the available error fields of temperature, wind speed, and $NO_x$ concentration.

The results reveal the effectiveness and diagnostic power of the methods devised (which remains the main scope of this study), allowing the detection of the time scale and the fields that the two models are most sensitive to. The representation of planetary boundary layers (PBL) dynamics is pivotal to both models. In particular: *i)* The fluctuations slower than ~1.5 days account for 70-85% of the mean square error of the full (undecomposed) ozone time series; *ii)* A recursive, systematic error with daily periodicity is detected, responsible for 10-20% of the quadratic total error; *iii)* Errors in representing the timing of the daily transition between stability regimes in the PBL are responsible for a covariance error as large as 9 ppb (as much as the standard deviation of the network-average ozone observations in summer in both Europe and North America); *iv)* The CMAQ ozone error has a weak/negligible dependence on the errors in $NO_2$, while the error in $NO_2$ significantly impacts the ozone error produced by Chimere; *v)* The response of the models to variations of anthropogenic emissions and boundary conditions show a pronounced spatial heterogeneity, while the seasonal variability of the response is found to be less marked. Only during the winter season the zeroing of boundary values for North America produces a spatially uniform deterioration of the model accuracy across the majority of the continent.

## 1. INTRODUCTION

The vast majority of the research and applications related to the evaluation of geophysical models make use of aggregate statistical metrics to quantify, in some averaged sense, the properties of the residuals obtained from juxtaposing observations and modelled output (typically time series of the variable of interest). This practice is rooted in linear regression analysis and the assumption of normally distributed residuals and has been proven to be reliable when dealing with simple, deterministic and low-order models. Led by the rapid pace of

improved understanding of the underlying physics, the paradigm is however changed nowadays in that models
have grown in complexity and nonlinear interactions and require more powerful and direct diagnostic
methods (Wagener and Gupta, 2005; Gupta, et al., 2008; Dennis et al., 2010; Solazzo and Galmarini, 2016).
Evaluation of geophysical models is typically carried out under the theoretical umbrella proposed by Murphy in
the early 1990s for assessing the dimensions of goodness of a forecast: consistency (*'the correspondence*
*between forecasters' judgments and their forecasts'*), quality (*'the correspondence between the forecasts and*
*the matching observations'*)*,* and value (*'the incremental benefits realised by decision makers through the use*
*of the forecasts'*) (Murphy, 1993). Since 2010, the Air Quality Model Evaluation International Initiative
(AQMEII, Rao et al., 2011) has focused on the quality dimension – the one most relevant to science, according
to Weijs et al. (2010) – of air quality model hindcast products, aiming at building an evalution strategy that is
informative for  modellers as well as to users.
Our claim is that the *value* of a model's result depends strictly on the *quality* of the model that, in turn,
depends on sound evaluation. The scientific problem of assessing the *quality* of a modelling system for air
quality is tackled by Dennis et al. (2010) who distinguish four complementary approaches to support model
evaluation: operational, probabilistic, dynamic and diagnostic, which are also the four founding pillars of
AQMEII. Several studies performed under AQMEII have focused on the operational and probabilistic evaluation
(Solazzo et al., 2012a,b; Solazzo et al., 2013; Im et al., 2015a,b; Appel et al., 2012; Vautard et al., 2012) and
more recently efforts have been expanded to the diagnostic aspect (Hogrefe et al., 2014; Solazzo and
Galmarini, 2016; Kioutsioukis et al., 2016; Solazzo et al., 2017).
Operational metrics usually employed in air quality evalution (cfr. Simon et al., 2012 for a review) have several
limitations as summarised by Tian et al. (2016): *interdependence* (they are related to each other and are
redundant in the type of information they provide), *underdetermination* (they do not describe unique error
features), and *incompleteness* (how many of these metrics are required to fully characterise the error?).
Furthermore, they do not help to determine the *quality* problem set above in terms of diagnostic power.
Gauging (average) model performance through model-to-observation distance leaves open several questions
such as *a)* How much information is contained in the error? In other words, what remains wrong with our
underlying hypothesis and modelling practice? *b)* Is the model providing the correct response for the correct
reason? *c)* What is the degree of complexity of the system models can actually match? These questions have a
straightforward, very practical impact on the use of models, the return they provide (the value) and their
credibility. Answers to these questions are also relevant to the wide-spread practice of bias correction which is
aimed at adjusting the model value to the observed value, rather than correcting the causes of the bias which
might stem from systematic, cumulative errors.
The main aims of this study are to move towards tools devised to enable diagnostic interpretation of model
errors, following the approach of Gupta et al. (2008 and 2009), Solazzo and Galmarini (2016),  and Kioutsioukis
et al. (2016) and to advance the evaluation strategy outlined in the course of the three phases of AQMEII. In
particular, the work presented here is meant to complement the overview analysis of the modelling systems
participating in AQMEII3 (summarised by Solazzo et al., 2017) by concentrating on the performance for surface
ozone modelled by two modelling systems: Chimere for Europe (EU) and CMAQ for North America (NA). This
study attempts to:
• Identify the time scales (or frequencies) of the error of modelled ozone;
• Attribute each type of error to processes by utilizing modelling runs with modified fluxes at the
boundaries (anthropogenic emissions and deposition at the surface, and boundary conditions at the
bounding planes of the domain) and breaking down the mean square error (MSE) into bias, variance
and covariance. This analysis allows us to diagnose the quality of error and to determine if it is caused
by external conditions or due to missing or biased parameterisations or process representations;
•      Investigate the periodicity of the ozone error which can be symptomatic of recursive (either casual or
systematic) model deficiencies;

•      Determine the role of the error of precursor or meteorological fields in explaining the ozone error.
The significance (or the non-significance) of a correlation between the ozone error and that of one of
the explanatory variables can help to understand the impact (or lack of impact) of the latter on the
ozone error as well as the time-scale of the process(es) causing the error.

Among the several models participating in AQMEII3, CMAQ and Chimere have been selected as the analysis
proposed in this study requires additional simulations beyond those performed by all AQMEII3 groups, which
implied additional dedicated resources that were not available to all groups. This of course opens an important
issue connected with the relevance of models in decision making, the adequacy of their contribution, and
consequently the fact that far more resources would be required by the present complexity and state of
development of modelling systems to guarantee that deeper evaluation strategies are put in place. Although
only these two modelling systems are analyzed here, they represent two well-established systems that have
been systematically developed over many years, are in use by a large number of research groups around the
world and also have participated in the various phases of AQMEII.
The data used, model features and error decomposition methodology are summarised in section 2. Results of
the aggregate time series and error decomposition analyses are presented in section 3 and results of the
diagnostic error investigation through wavelet, autocorrelation, and multiple regression analysis are presented
in section 4. Discussion, conclusions and final remarks are drawn in Sections 5 and 6.

## 2. METHODS

### 2.1 DATA AND MODELS

Unless otherwise specified, analyses are carried out and results are presented for the rural receptors of three
sub-regions over each continental area as shown in **Figure 1**. The three sub-regions have been selected based
on similarity analysis of the observed ozone fluctuations slower than ~1.5 days. The regions where the slow
fluctuations showed similar characteristics were selected through unsupervised hierarchical clustering (details
in Solazzo and Galmarini, 2015). Due to the similarity of the observations within these regions which implies
that they experience common physical and chemical characteristics, spatial averaging within these sub-regions
was carried out.
The stations used for the analysis are part of the European (European Monitoring and Evaluation Programme:
EMEP;     http://www.emep.int/;     European     Air     Quality     Database     AirBase;
http://acm.eionet.europa.eu/databases/airbase/) and North American (USEPA Air Quality System AQS:
http://www.epa.gov/ttn/airs/airsaqs/;    Analysis    Facility    operated    by    Environment    Canada:
http://www.ec.gc.ca/natchem/) monitoring networks. Full details are given in Solazzo et al. (2017) and
references therein.
Following the approach used in previous AQMEII investigations, modelled hourly concentrations in the lowest
model layer (~20m for both models) and corresponding observational data are paired in time and space to
provide a verification data sample $\{mod_r^t, obs_r^t;\ t=1,…,8760;\ r=1,…,n_{recs}\}$ of $n_{recs}$ (number of monitoring
stations) record of matched modelled and observational data, where the $r^{th}$-pair $mod^{t0}$ and $obs^{t0}$ is evaluated
at receptor $r$ at a given time $t_0$. Further, while the observations are reported at the hour at the end (for
Europe) or at the beginning (for NA) of the hourly averaging window, the model values available in this study
are provided instantaneously. Therefore, the model concentrations were assumed to be linear between the
instantaneous on-the-hour reporting times; the integration (average) between those times was used to
construct hour starting (or ending) values in order to more directly compare to the averaging used in the
observations. This is of particular relevance when estimating the error due to timing of the diurnal cycle
discussed in section 4.3.
For the analyses conducted in this study, the spatial average of the observed and modelled ozone time series
has been carried out prior to any time aggregation, i.e. the spatial average is created by averaging the hourly
values over all rural stations in each region. Missing values in the time series, prior of the spatial averaging,
have not been imputed. The analysis is restricted to stations with a data completeness percentage above 75%
and located below 1000m above sea level. Time series with more than 335 consecutive missing records (14
days) have been also discarded. The number of rural receptors $n_{recs}$ for ozone is 38, 184, and 40 for EU1, EU2,
and EU3 and 73, 43, and 28 for NA1, NA2, and NA3, respectively. The EU continental domain used for analyses
extends between -30 degree and 60 degree latitude, and between 25 degree and 70 degree longitude,
whereas the NA continental domain extends between -130 degree and -40 degree latitude, and between 23.5
degree and 69 degree longitude.
The configuration of the CMAQ and Chimere modelling systems for AQMEII3 is extensively discussed in
Solazzo, et al. (2017) with respect to resolution, parameterisations, and inputs of emissions, meteorology, land
use, and boundary conditions. For completeness a short summary is provided hereafter.
The CMAQ model (Byun and Shere, 2006) is configured with a horizontal grid spacing of 12 km and 35 vertical
layers (up to 50 hPa) and uses the widely applied CB05-TUCL chemical mechanism (Carbon Bond mechanism,
Whitten et al., 2010) for the representation of gas phase chemistry. Emissions from natural sources are
calculated inline by the Biogenic Emissions Inventory System (BEIS) model. The meteorology is calculated by
the Weather Research and Forecast (WRF) model (Skamarock et al., 2008) with nudging of temperature, wind
and humidity above the planetary boundary layer (PBL) height. In CMAQ, dry deposition is used as a flux
boundary condition for the vertical diffusion equation. A review of CMAQ dry deposition model as well as
other approaches is provided in Pleim and Ran (2011).
Chimere (Menut et al., 2013) is configured with a grid of 0.25 degree (corresponding, approximately, to 25 km
x 18 km over France), 9 vertical layers (up to 500 hPa) and uses the Melchior2 chemical mechanism (Lattuati,
1997) for the representation of gas phase chemistry. Natural emissions are calculated using the MEGAN model
(Guenther 2012). The hourly meteorological fields are retrieved from the Integrated Forecast System (IFS)
operated by the European Centre for Medium-Range Weather Forecast (ECMWF). In Chimere the dry
deposition process is described through a resistance analogy (Wesely, 1989). For each model species, three
resistances are estimated: the aerodynamical resistance, the resistance to diffusivity near the ground and the
surface resistance. For particles, the settling velocity is added. More information is included in Menut et al.
(2013).

Both models are widely used worldwide in a range of applications such as scenario analysis, forecasting,
ensemble modelling, and model inter-comparison studies.
## 2.2 SENSITIVITY RUNS WITH CMAQ AND CHIMERE
The Chimere and CMAQ models have been used to perform a series of sensitivity simulations aimed at a better
understanding of the causes of differences between the base model simulations and observed data. In
particular, the following set of sensitivity runs was performed:
• one annual run with zeroed anthropogenic emissions to provide an indication of the amount of
regional ozone due to boundary conditions and biogenic emissions (referred to as '*zero Emi*');
• one annual run with a constant value of ozone (zero for NA and 35 ppb for EU) at the lateral
boundaries of the model domain to provide an indication of amount of ozone formed due to
anthropogenic and biogenic emissions within the domain (in addition to the constant value for EU)
(referred to as '*zero BC*' and '*const BC*'). All species other than ozone had boundary condition values
of zero for both NA and EU in these sensitivity simulations;
• one annual run where the anthropogenic emissions are reduced by 20%. In addition, the boundary
conditions for this run were prepared from a C-IFS simulation (detail in Galmarini et al., 2017 and

references therein) in which global anthropogenic emissions were also reduced by 20% (referred to as
a '*20% red*');
• one run with ozone dry deposition velocity set to zero, available for the months of January and July
(referred to as '*zero Dep*').

==The analyses presented are not meant to inter-compare the two modelling systems, as the CMAQ and Chimere==
==models are applied to non-comparable contexts (different emissions, meteorology, and observational data).==
==The response of each model to the changes in emissions, boundary conditions and deposition needs to be==
==interpreted independently.==
2.3 ERROR DIAGNOSTIC METRIC
To aid diagnostic interpretation, the ==mean square (or quadratic)== error MSE (MSE = $E[mod\text{-}obs]^2$) is
decomposed according to

$$MSE = \left(\overline{mod} - \overline{obs}\right)^2 + (\sigma_m - \sigma_o)^2 + 2\sigma_m\sigma_o(1 - r) = bias^2 + var + covar \qquad \textbf{Eq 1}$$

where $\sigma_m$ and $\sigma_o$ are the modelled and observed standard deviation, *var* and *covar* are the variance and
covariance operators, *r* is the linear correlation coefficient, and *bias* is the time averaged offset between the
mean modelled and observed ozone concentration. ==The decomposition in Eq.1 (and several variations of it),==
==derived e.g. by Theil (1961), has been extensively discussed in Potempski and Galmarini (2009), Solazzo and==
==Galmarini, (2016), Gupta et al. (2009). The first two moments (mean and variance) relate to the systematic==
==error (unconditional bias) and variability (variance), respectively. All other differences between the statistical==
==properties of modelled and observed chemical species (e.g. the timing of the peaks and autocorrelation==
==features) are quantified by the correlation coefficient, i.e. in the covariance term (Gupta et al., 2009).==
The MSE is a quadratic, parametric metric widely applied in many contexts and occurs because the model does
not account for information that could produce a more accurate estimate. Put in an information theory
context, the MSE provides a measure of the information about the observation that is missing from a Gaussian
model centred at a deterministic prediction (Nearing et al., 2015). Ideally, the deviation of a perfect model
from the observation should be zero or simply white noise (uncorrelated, zero mean, constant variance).
Various flavours of MSE decomposition have been exploited in several geophysical contexts (Enthekabi, et al.,
2010; Murphy, 1988; Wilks, 2011; Wilmott, 1981; Gupta, et al., 2009), all stemming from the consideration
that the bias, the variance, and the covariance characterise different (although not complementary and not
exhaustive) properties of the error – accuracy, precision, and correspondence, respectively.
The relative contribution of each of the MSE components to the overall MSE is summarised by the Theil's
coefficients (Theil, 1961):

$$F_b = bias^2/MSE$$
$$F_v = var/MSE \qquad \textbf{Eq 2}$$
$$F_c = covar/MSE$$

The overall MSE suffers from the limitations of the aggregate metrics discussed in the introductory section,
lacking independence and explanatory power (Tian et al., 2016). When decomposed (e.g according to **Eq 1**),
however, the underdetermination issue is reduced and the MSE coefficients (**Eq 2**) do offer diagnostic aid in
interpreting the modelling error (Gupta et al., 2009).
## 3. SENSITIVITY ANALYSIS TO EMISSIONS AND BOUNDARY CONDITIONS PERTURBATIONS
### 3.1. AGGREGATED TIME SERIES OF OZONE
**Figure 2** and **Figure 3** show monthly and diurnal curves for the base and sensitivity simulations over the three
sub-regions in each continent. Results show that the monthly averaged curves of the zeroed emission runs
peak in April in NA and in July in EU (May to July in EU1 are approximately the same), indicating the periods
when the impact of background concentration (boundary conditions) and biogenic emissions on regional
ozone is largest: springtime in NA and summer in EU. The monthly curves of 'zero BC' and 'zero Emi' for NA are
anti-correlated between the months of April to July-August ('zero Emi' curve decreasing and 'zero BC' curve
raising) and during autumn ('zero Emi' curve rising and 'zero BC' curve decreasing), framing the interplay
among these two factors in terms of total ozone loading: boundary conditions dominating in autumn-winter
and biogenic plus anthropogenic emissions are more important during spring-summer. The springtime peak for
the zero emissions case over NA is consistent with the springtime peak in northern hemispheric background
ozone (Penkett and Brice, 1986; Logan, 1999) and the predominant westerly and north-westerly inflow into
the NA domain. The background ozone springtime peak is thought to be caused by a combination of more
frequent tropospheric/stratospheric exchange   and in-situ photochemical production during that season (Atlas
et al., 2003).
The daily averaged profiles of mean ozone for NA show that the observed peak (occurring between 16-18 LT in
NA1 and NA2 and ~1 hour earlier in NA3) is preceded by the peak in the base run by ~1hour in NA2 and by ~2-
3 hours in NA1, while the timing of the observed minimum (occurring at 8-9 am LT) is captured by the base run
in NA2 and NA3 while it is preceded by the base run by ~1 hour in NA1. The modelled morning transition to
convective conditions is in phase with the observations except for NA1 where the modelled transition occurs
one hour earlier than the observed one. The modelled afternoon transition in NA1 precedes the observed
transition by 3-4 hours, possibly due to errors in the partitioning between sensible and latent surface heat flux
that causes a faster-than-observed collapse of the PBL. One possible reason,  as discussed in Appel et al.
(2016), could reside in the stomatal conductance function and the heat capacity for vegetation in WRF and the
ACM2 vertical mixing scheme in both WRF and CMAQ (relative to the version of WRF and CMAQ used in the
current study). Recent updates to these processes in CMAQ lead to a change in the modelled diurnal cycle of
ozone as well as other pollutants and meteorological variables. In particular, the updates lead to a delay in the
evening collapse of the modelled PBL (Appel et al., 2016).
The shape of the 'zero BC' curve is similar in amplitude to that of the base run, suggesting that the effect of the
regional/background ozone represented through boundary conditions in a limited area model is mainly to shift
the mean concentration upwards while it has no major effect on the frequency modulation. By contrast, the
absence of anthropogenic emissions has a major effect of the amplitude of the signal as well as its magnitude
('zero Emi' curve). As discussed in the next section, these considerations translate into the bias and/or variance
type of error due to the boundary conditions and emissions.
As for EU (**Figure 3**), the observed daily profiles in EU1 and EU2 are closely matched by the Chimere model
between 11 LT and 23 LT (underestimated outside these hours), while in EU3 the daily peak (observed at 19-20
LT) is consistently occurring earlier in the model and its magnitude is overestimated. The morning transition
occurs earlier in the model than the observations and follows a significant model under-prediction of
nighttime and early morning ozone, due to difficulties in reproducing stable or near-stable conditions
(Bessegnet et al., 2016). In EU3, the model displays the poorest performance, with significant underestimation
between midnight and 9 LT (5-7 ppb) and over-estimation in daylight conditions (7-9 ppb).
As opposed to the CMAQ case for NA, the shape of the 'zero Emi' curve of Chimere closely follows the shape
that of the base case (even when considering only the stations classified as 'urban', Figure S2) Due to the long
time average (one year), the daily profiles displayed in **Figure 2** and **Figure 3** do not provide information about
the exact timing of the minima and maxima for each season throughout the year. Figure S3 and Figure S4
report the seasonal average diurnal profiles for the model predictions and the observations (network average
over all stations) and show that the timing of the ozone diurnal cycle varies seasonally.
3.2. ERROR DECOMPOSITION
The plots in **Figure 4** (NA) and **Figure 5** (EU) show the MSE decomposition according to Eq. 1 for the summer
months of June, July, and August for the base case simulation as well as the sensitivity simulations,
distinguishing between daylight (from to 5am to 9pm LT) and night-time hours (the remaining hours, from
10pm to 4am LT). These plots are meant to aid the understanding of the relative impacts of potential errors in
lateral boundary conditions, anthropogenic emissions, and the representation of ozone dry deposition on the
total model error by comparing the magnitude and type of model error from these simulations against the
model error for the base case.
The plots In **Figure 6** And **Figure 7** are complementary to Figures 4-5 and show the error decomposition for
both the summer and winter season in more detail, including the error coefficients *Fb, Fv, Fc* of **Eq 2** (left
vertical axis), the total MSE (right vertical axis), the sign of the bias and variance error (+/- for model over and
under prediction), and the values of the correlation coefficient. Furthermore, the maps in **Figure 8** and **Figure 9**
show the root MSE (RMSE) at the receptors for the 'base' case as well as *ΔRMSE*, i.e. the percentage change of
RMSE of the sensitivity runs with respect to the 'base' case simulation:
*ΔRMSE= 100\*(RMSE$_s$ − RMSE$_{base}$)/RMSE$_{base}$*, where the subscript *s* indicates the zeroed emission or the zeroed
(constant) boundary condition simulations (*ΔRMSE* is measured as percentage).
The CMAQ results for NA are presented in **Figure 4**, **Figure 6**, and **Figure 8** and can be summarised as follows:

- • The MSE of the base case (MSE$_{base}$) during summer daylight is mainly due to bias (~35% in NA1 and
~75% in NA2 and NA3) and the remaining portion is due to covariance error. The fact that there is no
variance error shows that the model is able to replicate the observed 3-month averaged variability.
Possible reasons for the positive model bias (model overestimation) have been discussed in Solazzo et
al. (2017) and includes overestimation of emissions precursors (Travis et al., 2016) and absence of
correct parameterizations of forested areas on surface ozone (Makar et al., 2017);
- • The effect of zeroing the emissions of anthropogenic pollutants on the summer MSE is a rise by a
factor ~2 to 4 (daylight) and by a factor ~6 to 7 during night-time in NA1 and NA2 with respect to
MSE$_{base}$, while during night-time in NA3 the MSE stays approximately the same, indicating that the
emissions have little role in determining the total error in this sub-region during summer night.
Furthermore:
- - All the error components deteriorate in the simulations with zero anthropogenic emissions
except for the bias in NA3. This is particularly true for the variance, signifying the fundamental
role of emissions in shaping the diurnal variation of ozone. Indeed, this suggests that the absence
of a variance error in the base case (see above) is due to the correct interplay between the
temporal/spatial distribution of the emissions, potentially coupled with the variability due to the
meteorology ;
- - The covariance share of the error also increases (although only slightly in NA2) for the zero
emissions case, indicating that the emissions play a role in determining the timing of the
modelled diurnal ozone signal, this increase is more pronounced during night-time.
- • The zeroing of the input of ozone from the lateral boundaries has either no effect or only a limited
effect (e.g. daylight summer in NA2, **Figure 4**) on the variance and covariance shares of the error,
while it has a profound impact on the bias portion. This impact is approximately equal during daylight
and night-time, as expected from the discussion of the daily cycle shown in **Figure 2**.
- • The removal of ozone dry deposition from the model simulations (results based on July only) has the
most profound impact, increasing by one order of magnitude the MSE of the base case which is
approximately double the combined effect of the emissions and boundary conditions perturbation.
This sensitivity gives a gross indication of the relative strength of this process vs external conditions
during summer, while the 'zero BC' case has a larger effect than the 'zero deposition' case in January
(not shown). Similar to the 'zero BC' case, the exclusion of ozone dry deposition from the model
simulations acts as an additive term to the diurnal curve in NA1, leaving almost unaltered the shape
and timing of the signal, while it impacts the variance and covariance error in the other two sub-
regions. The little impact the removal of dry deposition has on the covariance error (timing of the
ozone signal) together with the outweighing offsetting bias might suggest that the correct estimate of
the deposition magnitude is more beneficial than, e.g., the time dependence of surface resistance.
The role of the variance is however unclear and deserves further analyses.
• The instances where the '20% red' bias error is lower than the error of the base case occur when the
mean ozone concentrations were overestimated in the base case (e.g. daylight for all sub-regions and
NA2 and NA3 over night-time summer) as illustrated in **Figure 6**a,b.
• The maps show that there are stations where the error is reduced with zero anthropogenic emissions
(e.g. a reduction of 20-30% in the south coast of the US and in the far North-east during summer,
**Figure 8**d). This suggests the presence of other compensating model errors in both the base and
sensitivity simulations that lead to better agreement with observations when prescribing an
unrealistic emission scenario. The sources of these compensating errors need to be investigated in
future work.
• The 'zero BC' run has profound negative effects over the whole continental area of NA during winter
(**Figure 8**e), while the effects are smaller during summer (**Figure 8**f) especially over the southern coast
due to the relatively higher importance of photochemical formation of ozone during summer.
• The error characteristics of the daily maximum 8-hour rolling mean (DM8h, **Figure 6**e) resemble those
of the daylight base case (Figure 6a, left column), but reduced in magnitude during winter , with
almost null variance error and the same sign of the bias as the base case. The NA1, NA2, and NA3
standard deviations of the summer DM8h is of 7.6, 5.2, and 8.1 ppb and of 7.6, 6.5, and 7 ppb for the
model and the observations, respectively. The model variability is therefore in line with the observed
variability. The error of the DM8h for the sensitivity runs is reported in Figure S5.
• On a network-wide average, removing anthropogenic emissions causes a RMSE increase of 25%
during summer and of 0% (10% at 75[th] percentile) during winter while a zeroing out of input from the
lateral boundaries causes a RMSE increase of 30% during summer and of 180% during winter (median
values, **Figure 8**).
The allocation of the error of the Chimere model for EU varies greatly by sub-region (**Figure 5**, **Figure 7**, and
**Figure 9**):
• The summer daylight RMSE$_{base}$ ranges between ~20 ppb$^2$ (EU1, ~60% covariance and ~20% bias) and
~85 ppb$^2$ (EU3, 95% covariance).  In EU3, the night-time bias of ~75% outweighs the covariance as
seen in **Figure 7**a.
• Removing the anthropogenic emissions had almost no effect on the covariance share of the MSE (if
not a slight reduction with respect to the base case in EU2 and EU3, and also during night-time),
indicating that the error in the timing of the signal is not influenced by the emissions but rather by
other processes.  Moreover, the variance portion is left almost unchanged (1 ppb increase in EU1 and
EU2), in contrast with the CMAQ results for NA. This would indicate that the variability of ozone
concentration is hardly influenced by anthropogenic emissions in Chimere. The bias is the error
component most sensitive to emissions reductions, especially in EU2 and less so in EU3. This is in line
with the discussion of the daily profiles of **Figure 2**b (which showed similar shapes of for the 'zero
Emi' and of the 'Base' profiles) and contrasts with the NA case where the 'zero Emi' daily profiles are
flatter than the base case.
• The effect of imposing a constant ozone boundary condition value of 35 ppb (and of zero for all other
species) has a net small effect on the variance of the ozone error, but significantly reduces the
covariance share of the error in favour of the bias  (**Figure 5** and FIGURE 7d). The total MSE is similar
to that of removing the anthropogenic emissions as far as the total MSE and the bias of EU2 are
concerned. It outweighs the latter for the total MSE, bias and variance in EU3 and covariance and
night-time bias component in EU1. We can infer that the variability of the boundary conditions have a
significant role in determining the timing of the ozone signal in EU1 (close to the western boundary of
the domain) as the correlation coefficient degrades form 0.89 (base case) to 0.66 ('const BC') (**Figure**
**5** and **Figure 7**a and c). The bias staying the same in EU1 daylight summer depends on the magnitude
of the constant value (35 ppb were chosen here) that is in close agreement with that of the base case
while the small variance error (~2ppb) vanishing with respect to the base case might be explainable
with numerical compensation.
- During summer in EU2 and EU3 changing the ozone boundary condition only influences the bias with
marginal impacts on variance and covariance, while in winter (**Figure 7**c) there is also a significant
reduction of the correlation coefficient, meaning that the boundary conditions modulate the timing of
the signal. This also implies that the variability of the boundary conditions themselves become more
important in winter.
- EU3 deserves special consideration as the $RMSE_{zeroEmi}$ is approximately the same as the $RMSE_{base}$,
which mostly consists of covariance error during daylight and bias error during night-time (**Figure 5**e).
Due to the local topography, EU3 is typically characterised by stagnant conditions that are difficult to
model. For example, 50% of the observed wind speed is below 1.65 ms$^{-1}$, while Chimere predicts 1.95
ms$^{-1}$. The largest impact on the total MSE is seen in the 'const BC' run and arises in the bias portion,
pointing to the importance of properly characterising background (regional) concentrations.
- With respect to the base case, the DM8h (**Figure 7**e) shows a reduced share of the covariance error
with respect to the mean ozone (Figure 7a) at the expense of an increase in variance error; the timing
error is now shifted towards seasonal time scales. The variability of the DM8h is governed by synoptic
processes which are likely responsible for the variability error of the DM8h. The EU1, EU2, and EU3
standard deviations of the summer DM8h is of 3, 6.2, and 8.6 ppb and of 6, 11, and 10.2 ppb for the
model and the observations, respectively. The model therefore underestimate the observed
variability (as indicated by the 'minus' sign in the variance share of the error in Figure 7e) by up to
50% in EU1. A range of processes could be responsible for the lack of variability in Chimere, from
emission to chemistry to transport. The error of the DM8h for the sensitivity runs is reported in Figure
S6.
- On a network-wide average, removing anthropogenic emission causes an RMSE increase of 21%
during summer and of 12% during winter (median values, **Figure 9**c,d).
- The effect of setting the dry deposition velocity of ozone to zero (July only, **Figure 5**), increases not
only the bias error but also causes large increases of the variance and covariance shares of the error.
Thus in Chimere the deposition acts not only as a shifting term on the modelled concentration but it
also influences the variability and timing of ozone more profoundly than for the CMAQ case examined
earlier.
## 4. TIME-SCALE ERROR ANALYSIS AND DIAGNOSTIC
The focus of this section is $\Delta O_3$, the time series of the deviation between the base case and observations. The
nature of $\Delta O_3$ is examined for time-frequency patterns using wavelet analysis and for error persistence using
autocorrelation functions (ACF). The causes of $\Delta O_3$ are also tentatively investigated as dependencies on other
fields using multiple regression analysis combined with bootstrapping to sample the relative importance of the
regression variables.
### 4.1. SPECTRAL CONSIDERATIONS
The coefficients of the ACF (Appendix 1) can be interpreted as the Fourier transform of the power spectral
density. Frequency analysis of a signal is often performed by constructing the periodogram (or spectrogram,
see e.g. Chatfield, 2004). This approach has proven useful when dealing with harmonic processes
superimposed on a baseline signal (Mudelsee, 2014) but, at the same time, periodograms often contain high
noise. Therefore, examining a signal at specific frequencies can be instructive, for instance by resorting to
wavelet transform which has the further advantage of enabling a 3-dimensional time-frequency-power
visualisation. Compared to a power spectrum showing the strength of variations of the signal as function of
frequencies, wavelet transformation also allows the allocation of information in the physical time dimension
other than phase space. Here, wavelet analysis of the periodogram of seasonal $\Delta O_3$ is performed using the
Morlet wavelet transform (Torrence and Compo, 1997).
From inspecting **Figure 10** (NA) it emerges that the highest values of spectral energies for $\Delta O_3$ for the three
sub-regions (corresponding to the 99[th] percentile of the spectrum) are observed for periods spanning the
whole year (i.e. the intensity keeps the same high value during the whole year and is associated to a
periodicity higher than ~300 days). These high values of the energy spectrum are likely associated with the
slow variability of the non-zero bias throughout the investigated period that acts as a slow envelop modulation
of the error at shorter time scales. Such a process is more evident in NA1 and NA2 and its magnitude is one
order of magnitude (or more) higher of the 90[th] percentile value.
NA3 and to a lesser extent NA2 show a high spectral power of the error for periodicities of 1-2 months and
lasting from January to May with a weaker wake extending up to the end of the year, potentially pointing to
errors in the characterisation of larger-scale background concentrations associated with boundary conditions.
NA3 also exhibits a high spectral power for errors associated with a periodicity of ~20 days during January-
February and June-July and ~ 15 days during October and December.  This may point to errors in representing
the effects of changing weather regimes on simulated ozone concentrations.
Except for the long-term variations of the model error with periodicities greater than 2 months discussed
above, NA1 is the only sub-region that shows only weak power associated with model errors of shorter
periodicities from June to December. This suggests that fluctuations caused by variations in large scale
background and changing weather patterns are better captured in this region compared to the other two sub-
regions.
The energy associated with the daily error is again higher and more pronounced in NA3 than in the other sub-
regions where it is most pronounced during summer (NA1) or between March to October (NA2). While during
winter and autumn the daily error is likely driven by difficulties in reproducing stable PBL dynamics, during
spring and summer it is also influenced by the chemical production and destruction of ozone, a process
entailing $NO_x$ chemistry, radiation, biogenic emission estimates and chemical transformation, and thus difficult
to disentangle from boundary layer dynamics. Wavelet plots of the ozone error for periods between 12 hours
and 6 days are reported in Figure S7 and Figure S8, allowing to better identify the periods (and/or the
periodicity) affecting the error of the fast fluctuations, e.g the daily error in NA3 (all year) and the high energy
spot towards the end of April in NA2 with a periodicity of ~6 days and above, that could be associated to an
ozone episode, but analysis of episodes is beyond the scope of this investigation.
For the EU (**Figure 11**) a notable feature is the very high daily error energy in EU3 that is present throughout
the year and most pronounced in summer. Such high energy suggests persistent problems in representing
processes having a periodicity of one day. Further, EU3 shows an area of high energy associated with a period
of one to two months and extending from February, peaking in April and May, and ending in September
(mostly model underestimation, **Figure 11**c), while the error of the winter months in EU3 receives high energy
from slower processes, acting on time scales of ~6 months and beyond. Considering that the EU3 region is
surrounded by high mountains, tropopause folding (e.g. Bonasoni et al., 2000; Makar et al., 2010) together
with the lack of modelling mechanisms for the tropopause/stratosphere exchange, could offer an explanation
of the high energy of the error at long time scales (also considering that the higher level modelled by Chimere
is well below the tropopause and that vertical fluxes are those prescribed by the C-IFS model). Errors in the
biogenic emissions also remain a plausible cause of ozone error during spring and summer months.
The similarity of the wavelet spectra for NA3 (**Figure 10**c) and EU1 (**Figure 11**a) (both regions are located on
the Western edge of their domain) at the beginning of the year for periods of 1 to 2 months might be
indicative of the periodicity of the bias induced by the boundary conditions. Compared to CMAQ, the error of
the Chimere model is more concentrated during spring and early summer, with a periodicity of 10-20 days.
Having identified some relevant time-scales for the $\Delta O_3$ error, in the next sections methods are proposed for
its detection and quantification.
4.2. TEMPORAL CHARACTERISTICS OF THE ERROR OF OZONE
In a recent study, Otero et al. (2016) analyzed which synoptic and local variables best characterise the
influence of large scale circulation on daily maximum ozone over Europe. The authors found the majority of
the variance during spring over the entire EU continent is accounted for in the 24 hour lag autocorrelation
while during summer the maximum temperature is the principal explanatory variable over continental EU.
Other influential variables were found to be the relative humidity, the solar radiation and the geopotential
height. Camalier et al. (2007) and Lemaire et al. (2016) found that the near-surface temperature and the
incoming short-wave radiation were the two most influential drivers of ozone uncertainties.
The ACF and PACF (partial autocorrelation function) of $\Delta O_3$ (see Appendix 1 for a definition of both functions)
reveals a strong periodicity for periods that are multiples of 24 hours (Figure 12a And **Figure 13**a) (note that
the first derivative of $\Delta O_3$ is used in this analysis to achieve stationarity). The structure of the error is such that
it repeats itself with daily regularity, indicating either a systematic error in the model physics or a missing
process at the daily scale, possibly related to radiation and/or PBL-related variables. While the presence of a
daily periodic forcing due to the deterministic nature of day/night differences superimposed on the baseline
ozone is expected, the periodicity maintained in the error structure is not and deserves further analysis.
The PACF plots confirm that the error is not simply due to propagation and memory from previous hours, but
arises at 24h intervals and hence stems from daily processes. On average, for NA $corr(\Delta O_3(h), \Delta O_3(h+1))$ (i.e.
the correlation between $\Delta O_3(h)$ and $\Delta O_3(h+1)$) is ~0.45, while the $corr(\Delta O_3(h), \Delta O_3(h+24))$~0.68, for any given
hour $h$. Similarly for EU, $corr(\Delta O_3(h)$ and $\Delta O_3(h+1))$ ranges between 0.31 (EU2) and 0.54 (EU3), while
$corr(\Delta O_3(h),\Delta O_3(h+24))$ ~0.70 for all sub-regions. Thus, the ozone error with a 24h periodicity has a longer
memory than the error with a one hour periodicity. Since the 24h periodicity of the error is present in the
entire annual time series, the periodic error is not associated with particular conditions (e.g. stability), but is
rather embedded into the model at a more fundamental level. Moreover, similar periodicity is observed for:
• The ACF analysis repeated for the 'zero Emi' scenario (Fig S9)
• the ACF of $\Delta WS$ and $\Delta Temp$ for both models (Fig S10),
• The ACF of primary species (PM$_{10}$ for EU and CO for NA) (Fig. S11);
• The ACF of ozone error for the 'zero Emi' scenario at three stations where isoprene emissions are low
(Figure S12). These stations have been selected by looking at the locations where isoprene emissions
accumulated over the months of June, July, and August as provided by the two models analysed here.
In all cases, the error has a marked daily structure, strengthening the notion that a daily process affecting
several model modules is not properly parameterised. The error due to chemical transformation at daily scale
is screened out by the daily periodicity of the ACF of the primary species, while the daily periodicity of the
zeroed emission scenario allows reinforcing the claim that the PBL dynamics is the most probable cause of the
error.
Since the individual daily processes directly or indirectly affecting the PBL dynamics cannot be untangled, here
'PBL error' is meant to encompass errors in the representation of the variables affecting boundary layer
dynamics (i.e. radiation, surface description, surface energy balance, heat exchange processes, development
or suppression of convection, shear generated turbulence, and entrainment and detrainment processes at the
boundary layer top for heat and any other scalar) and their non-linear interdependencies.
By removing the diurnal fluctuations (i.e. by screening out the frequencies between 12 hours and up to ~1.5
days by means of the Kolmogorov-Zurbenko (*kz*) filter, as described in Hogrefe et al., 2000) from the modelled
and observed time series, the daily structure of the ACF disappears (Figure 12b and **Figure 13**b), replaced by a
slow decay and negative (EU1, EU2 and partially NA1, NA2) or fluctuating (NA3, EU3) correlation values. The
PACF plots in Figure 12b and **Figure 13**b suggest that some significant correlation persists up to ~40 hours,
likely due to leakage from the removed diurnal component. As extensively discussed in several earlier works,
the *kz* filter does not allow for a clear separation among components and thus some leakage is expected, (see
e.g. Galmarini et al, 2013; Solazzo et al. 2017). The amount of overlapping variance between the isolated
diurnal fluctuations and the remainder of the time series is of ~4-9%.
The relative strength of the MSE for the undecomposed ozone time series and for the ozone time series with
the diurnal fluctuations removed and with only the diurnal fluctuations is reported in Table 1. With the
exception of NA1 and EU3, the base line error (denoted with 'noDU') accounts for ~70 to 85% of the total
error, while the diurnal fluctuations (denoted with 'DU') are responsible for 10 to 23% of the total error (and
even less during nighttime). The 'DU' error outweighs the 'noDU' error (67% to 26%) only in EU3, where the
daily PBL issue has been pointed out in the previous section.
4.3 COVARIANCE ERROR: PHASE SHIFT OF THE DIURNAL CYCLE
This section explores the nature of the covariance error which occurs, among other reasons, when the two
signals being compared are not in phase. The first and second moments of the error distribution are invariant
with respect to a phase shift between the two signals (Murphy, 1995), i.e. the mean of the signal as well as the
amplitude of the oscillations with respect to the mean value are not affected by a phase shift which therefore
does not have an impact on the bias and variance components of the error. The correlation coefficient, on the
other hand, is impacted by a lagged signal, producing a net increase of the covariance error.
The analysis of the phase lag between the daily component of the modelled and observed cycles is reported in
**Figure 14** (NA) and **Figure 15** (EU), winter and summer are analysed separately.
To perform this analysis, the modelled and observed ozone time series are first filtered to isolate the diurnal
component using a kz filter. Then, the cross-covariance between the two time series is calculated. The time at
which the maximum covariance value occurs is taken as the phase shift between the two signals. The method
has an error of ±0.5 hours.
In NA, the modelled diurnal peak occurs 1-2 hours earlier than the observed diurnal peak at many stations, and
up to 3-4 hours earlier at some Canadian stations. By taking into consideration the 0.5 hour error of the
estimate, the receptors at the western border (approximately corresponding to NA3) are least affected by this
timing error (especially in summer **Figure 14**b), and therefore the covariance share of the error shown in
**Figure 4** is not due to daily phase shift in this region but probably due to the shifting of longer (or shorter)
time periods induced for example by errors in transport (wind speed and/or direction). Figures S7 in the
Supplementary report the same analysis repeated for the 'zero Emi' and 'Zero BC' runs.
In the EU (**Figure 15**), no phase shift (or a phase shift compatible with the 0.5 hour estimation error) is
observed in Romania, Germany and the UK during winter, while a significant phase shift (the modelled peak
occurs up to 6 hours early) is observed in the North of Italy and Austria, with France and Spain oscillating
between positive 3 (model delay up to 5 hours in the south of Madrid) and negative 5 and 6 hour phase shifts,
with the net effect of a spatially aggregated daily cycle that is in phase with the observations (**Figure 3**b).
During summer the phase shift is larger and extends also to the countries where the phase shift was null
during winter. Moreover, some country-wise grouping can be detected, as for example at the border between
Belgium and France, Spain and France, Finland to Sweden, possibly due to the different measurement
techniques and protocols among EU countruies (e.g. Solazzo and Galmarini, 2015). Figures S8 in the
Supplementary report the same analysis repeated for the 'zero Emi' and 'Const BC' runs. The difference
between the time shift of the base case and the zeroed emission scenario reveals the effects of the timing of
the anthropogenic emissions on the covariance error. The effect is null over EU (median value of the difference
of zero) and is very limited in NA (median value of zero during summer and of -1 during winter).
While errors in emission profiles obviously can be one cause of the phase shift and thus the covariance error of
the modelled ozone signal, the representation of boundary layer processes clearly can be a factor as well. As
discussed in e.g. Herwehe et al. (2011), the parameterisation of vertical mixing during transitional periods of
the day can cause a time shift in the modelled ozone concentrations due to its effects on the near-surface
concentrations of $NO_x$ and ozone, which in turn affect the chemical regime and balance between ozone
formation and removal.
To quantify the importance of the covariance error caused by a phase shift relative to other sources of error,
**Figure 16** shows the curves of normalised MSE as the observed ozone time series is shifted with respect to
itself between -10 and 10 hours. The MSE curve equals zero for a zero-hour lag and is symmetric with respect
to the sign of the lag. Since this analysis compares the observed signal to itself (with varying degrees of time
lags), the MSE fraction of bias and variance is zero while all of the MSE is due to the covariance.
The curves in **Figure 16** shows that a phase lag in the diurnal cycle of $\pm6$ hour causes a MSE error in the diurnal
component of magnitude ~$var(obs)$ (in both EU and NA), where $var(obs)$ is the variance of the measured
diurnal cycle (top panel). The effect on the full (undecomposed) time series is that a phase lag of $\pm4$ (EU) and
$\pm5$-6 (NA) hour in the diurnal cycle causes a MSE error of magnitude ~$var(obs)$, where in this case the variance
is that of the undecomposed time series of ozone (lower panel).
Therefore, a modelled ozone peak that occurs 4 to 5 hours too early (a feature that is detected at some EU3
and Canadian stations) corresponds to a covariance error of 9.0 ppb (i.e. the standard deviation of the
network-average ozone observations in summer in both EU and NA). This result also helps explain the large
covariance error in EU3, which can be at least partially attributed to the large phase shift of the daily cycle.
4.4 EXPLAINING THE ERROR OF OZONE
In this section a simple linear regression model for the error of ozone $\Delta O_3$ is applied with the goal of detecting
the causes of model errors on the daily and longer term scales identified in the previous section. Although a
linear model is overly simplistic and other methods are available (e.g kernel smoothers), we employed the
simpler approach since *i)* it is not the aim of this study to build a statistically accurate model for the model
error , and *ii)* by pursuing simple reasoning we hope to identify the time scale of the error and the most likely
fields causing it at that time scale. More advanced techniques are likely to overcomplicate the results and their
interpretations but could be pursued in future studies.
The available regressors (explanatory variables) are the errors of the variables for which measurements have
been collected within AQMEII, i.e. NO (EU only), $NO_2$, Temp, and WS:

$$\Delta O_3 = \beta_1 \Delta NO + \beta_2 \Delta NO_2 + \beta_3 \Delta Temp + \beta_4 \Delta WS + k \qquad \textbf{Eq 3}$$


where $\beta_i$ are the coefficients of the multiple linear regression, and the intercept *k* is the portion of the ozone
error not explainable by any of the regressors. A bootstrap analysis (Mudelsee, 2014; Groemping, 2006) is used
to calculate the relative importance of each error field in explaining the variance of $\Delta O_3$ (**Figure 17** and **Figure
18**) with an uncertainty of ~5%. The analysis is restricted to stations of ozone, $NO_x$, WS and Temp that are
The errors of temperature and wind speed explain about a third of the daylight winter ozone error of CMAQ,
while ~20% of the ozone error variability during daylight summer ozone is associated with the error in
temperature and, to a lesser extent, wind speed (**Figure 17**). In contrast, in Chimere the NO and $NO_2$ error over
EU during winter is correlated with the error of ozone, especially during night-time. (**Figure 18**). Overall, there
is no instance where the variance explained by the available variables (quantified through the coefficient of
determination $R^2$) exceeds 0.45 (corresponding to a linear correlation coefficient of ~0.67). The ACF of the
residuals of the regression show that there is an overwhelming daily memory of the error that can only
partially be attributed to errors of the available regressor variables, pointing to the need to include additional
variables in future applications of this regression analysis.
A straightforward limitation of **Eq 3** is that it assumes that successive values of the error terms are independent
while in practice this is not the case. Table 2 reports the correlation coefficient of the diurnal fluctuations of
the residuals, obtained by filtering out fluctuations faster than ~1.5 days from the measured and observed
time series (for the analysis of Table 2 the co-location restriction on the rural receptors is removed to allow
spatial considerations, the only constraint is on the of the vertical displacement among stations to be less than
250m). Several significant collinearities can be detected (e.g between ΔWS and ΔTemp; $ΔNO_2$ and ΔTemp,
especially in winter).
In addition to the collinearity issue, there are other endogenous variables that are not part of the regression
analysis but whose error contributes to total $ΔO_3$, as revealed by the ACF and PACF of the first-order
differentiated residuals of the regression, reported in the last panels of each plot. Such missing variables are
likely to correlate with both the dependent ($ΔO_3$) and the explanatory variables. For instance, errors in the
cloud cover and/or radiation scheme, land use masking, etc. are shared by the chemical species (ozone and its
precursors) as well as by the meteorological fields. The ACF and PACF suggest that the common, omitted error
of the fit propagates with daily recurrence and is not explained by the available variables, stressing the findings
of the previous section and again pointing to PBL-related errors.
However, since we are not in a position to estimate the errors associated with PBL variables (radiation,
temperature, turbulence) an alternate approach is to filter out the diurnal process from the modelled and
observed time series and repeat the analysis based on Eq 3 (Figure S11 and Figure S12). The correlation
coefficients of the residuals with the diurnal component filtered-out. The collinearity has been largely
removed, especially for NA, while for EU some strong correlation persists ($ΔNO_2$ and ΔNO, and between ΔWS
and ΔTemp in winter):
The $R^2$ of the regression for the 'no-DU' case drops drastically in NA, while keeping approximately the same
values in EU (but in EU3 $R^2$ does not exceed 0.10, not shown) as shown in Figures S16 and S17. Moreover, this
analysis and its comparison to the results presented in earlier sections lead to the following conclusions:
• A strong daily error component is common to all variables investigated here.
• This error manifests itself in the correlation coefficient, thus is due to a variance/covariance type of
error (otherwise, if it was a bias-type error, the $R^2$ would have been similar between the analysis of
the signal with and without the diurnal component);
• By inspecting the 'no-DU' case, at least in NA (Fig S16), the bias error discussed in section 3 cannot be
explained simply in terms of the fields $NO_2$, Temp, and WS. Hence, the bias of the CMAQ model over
the NA continent appears to be associated with processes with longer time scales (i.e. longer than
daily), such as boundary conditions (inducing mostly bias error, as discussed in section 3), deposition,
and/or transport (potential systematic errors in wind direction, for example, would likely produce a
bias-type error);
•
•   The impact of $\Delta NO_2$ and $\Delta NO$ in EU (all sub-regions, mostly daylight) and of $\Delta WS$ in EU1 (and partially
EU2) on the error of ozone (not shown) is similar with and without the diurnal fluctuations, indicating
cross-correlation of these error fields for periods longer than one day.

## 625 5 Discussions

The application of several diagnostic techniques in conjunction with sensitivity scenarios has allowed analysing
in depth the time scale properties of the ozone error of CMAQ and Chimere, two widely applied modelling
systems. The main results, as stemming from various aspects of the investigation, are that the largest share of
MSE (~70-85%) is associated with fluctuations longer than the daily scale, and mostly due to offsetting error in
NA and due to covariance error in EU, while the remaining MSE is due to processes with daily variation. The
causes of the long term error need to be sought in the fields that produce (mainly) a bias type of error such as
emissions, boundary conditions, and deposition for NA, while the time shift of the slow fluctuations in EU is
possibly due to timing error of the synoptic drivers or other synoptic processes.
By excluding other plausible causes, and assuming that observational data are 'correct' (not affected by
systematic errors), we can conclude based on multiple indicators that the dynamics of the boundary layer
(which in turn depend on the representation of radiation, surface characteristics, surface energy balance, heat
exchange processes, development or suppression of convection, shear generated turbulence, and entrainment
and detrainment processes at the boundary layer top for heat and any other scalars) is responsible for the
recursive daily error. The most revealing indicator is the analysis of the ACF and PACF of the time series of
ozone residuals that shows a daily periodicity: the 24-hour errors are highly associated throughout the year,
i.e. the error repeats itself with daily regularity. This could be caused by multiple processes occurring on a daily
time scale, such as chemical transformations, the timing of the emissions, and PBL dynamics. However,
analyses of the error periodicity of primary species (to exclude the role of chemical transformations) and of the
scenario with zeroed anthropogenic emissions (to exclude the role of emissions) have shown the same error
structure, pointing to PBL processes as the main cause of daily error.
Due to the spatial aggregation of these analyses and the non-linearity of the models' components, it is possible
that the periodicity of the error could be due to a combination of multiple processes at specific sites. However,
the absence of a spatial or emission dependence and the persistence of the daily periodicity indicate that the
main cause of the daily error stems from PBL dynamics. Furthermore, the analogies of the time shift of the
diurnal component of the base and zeroed emission cases suggest that the timing error (pure covariance error)
is not caused by anthropogenic emissions (with the possible exception of winter in NA where some small
differences are present).

## 656 6. Conclusions

This study is part of the goal of AQMEII to promote innovative insights into the evaluation of regional air
quality models. This study is primarily meant to introduce evaluation methods that are innovative and that
move towards diagnosing the causes of model error. It focuses on the diagnostic of the error produced by
CMAQ and Chimere applied to calculate hourly surface ozone mixing ratios over North America and Europe.
We argue that the current, widespread practice (although with several exceptions) of using time-aggregate
metrics to merely quantify the average distance (in a metric space) between models and observations has
clear limitations and does not help target the causes of model error. We therefore propose to move towards
the qualification of the error components (bias, variance, covariance) and to assess each of them with relevant
diagnostic methods. At the core of the diagnostic methods we have devised over the years within AQMEII is
the quality of the information that can be extracted from model and measurements to aid understanding of
the causes of model error, thus providing more useful information to model developers and users than can be
gained from more aggregate metric. Applying such approaches on a routine basis would help boost the
confidence in using models prediction for various applications. At the current stage, the methods we propose
help identify the time scale of the error and its periodicity. The step to link the error to specific processes can
only be reached by integrating the analysis with sensitivity model runs. For instance, we can infer that the
timing error of the diurnal component is (at least partially) associated to the dynamics of the PBL, but further
analyses are necessary to isolate the components of the PBL responsible for that error.
While remarking that the analyses carried out are not meant to compare the two models but are rather meant
to show how the two models, applied to different areas and using different emissions, respond to changes, the
main conclusions of this study are:
- While the zeroing/modification of input of ozone from the lateral boundaries causes a shift of the
ozone diurnal cycle in both CMAQ and Chimere, the response of the two models to a modification of
anthropogenic emission and deposition fluxes is very different. For CMAQ, the effect of removing
anthropogenic emissions causes a shift and a flattening of the diurnal curve (bias and variance error),
while for Chimere the effect is restricted to a shift. In contrast, setting the ozone dry deposition
velocity to zero causes a shift (bias error) for CMAQ, while a profound change of the error structure
occurs for Chimere with significant impacts not only on the bias but also the variance and covariance
terms.
- The response of the models to variations in anthropogenic emissions and boundary conditions show a
pronounced spatial heterogeneity, while the seasonal variability of this response is found to be less
marked. Only during the winter season the zeroing of boundary values for North America produces a
spatially uniform deterioration of the model accuracy across the majority of the continent.
- Fluctuations slower than ~1.5 days account for 70-85% of the total ozone quadratic error. The
partition of this error into bias, variance and covariance depends on season and region. In general,
the CMAQ model suffers mostly from bias error (model overestimation during summer and
underestimation during winter), while the Chimere model is rather 'centred' (i.e. almost unbiased)
but suffers high covariance error (associated with the timing of the signal, thus likely to synoptic
drivers)
- A recursive, systematic error with daily periodicity is detected in both models, responsible for 10-20%
of the quadratic total error, possibly associated with the dynamics of the PBL;
- The modelled ozone daily peak accurately reproduces the observed one, although with significant
exceptions in France, Italy and Austria for Chimere and with the exceptions of Canada and some areas
in the eastern US for CMAQ. Assuming the accurateness of the observational data in these regions,
the modelled peak is anticipated by up to 6 hours, causing a covariance error as large as 9 ppb. The
analysis suggests that the timing of the anthropogenic emissions is not responsible for the phasing
error of the ozone peaks, but rather indicates that it might be caused by the dynamics of the PBL
(although the role of biogenic emissions and chemistry cannot be ruled out);
- The ozone error in CMAQ has a weak/negligible dependence on the error of $NO_2$ and wind speed,
while the error of $NO_2$ impacts significantly the ozone error produced by Chimere. On time scales
longer than 1.5 days, the Chimere ozone error is significantly associated with the error of wind speed
and temperature.
Although having exploited several evaluation frameworks over the past ten years within AQMEII (operational,
diagnostic, and probabilistic) the goal of clearly associating errors to processes has not yet been achieved. As
already suggested in the conclusions of the collective analysis of the AQMEII3 suite of model runs summarised
by Solazzo et al. (2017), future model evaluation activities would benefit from incorporating sensitivity
simulations and process specific analyses that help to disentangle the non-linearity of the many model
variables, possibly by focusing on smaller modelling communities. The 'theory of evaluation' being put forward
by the hydrology modelling community (Nearing et al., 2016 and references therein) may provide a template
for the air quality community to further advance their model evaluation approaches.

ACKNOWLEDGMENTS
We gratefully acknowledge the contribution of various groups to the third air Quality Model Evaluation
International Initiative (AQMEII) activity.  The following agencies have prepared the data sets used in this
study: U.S. EPA (North American emissions processing and gridded meteorology); U.S. EPA, Environment
Canada, Mexican Secretariat of the Environment and Natural Resources (Secretaría de Medio Ambiente y
Recursos Naturales-SEMARNAT) and National Institute of Ecology (Instituto Nacional de Ecología-INE) (North
American national emissions inventories); TNO (European emissions processing); ECMWF/MACC (Chemical
boundary conditions).   Ambient North American concentration measurements were extracted from
Environment Canada's National Atmospheric Chemistry Database (NAtChem) PM database and provided by
several U.S. and Canadian agencies (AQS, CAPMoN, CASTNet, IMPROVE, NAPS, SEARCH and STN networks);
North American precipitation-chemistry measurements were extracted from NAtChem's precipitation-
chemistry data base and were provided by several U.S. and Canadian agencies (CAPMoN, NADP, NBPMN,
NSPSN, and REPQ networks); the WMO World Ozone and Ultraviolet Data Centre (WOUDC) and its data-
contributing agencies provided North American and European ozonesonde profiles; NASA's AErosol RObotic
NETwork (AeroNet) and its data-contributing agencies provided North American and European AOD
measurements; the MOZAIC Data Centre and its contributing airlines provided North American and European
aircraft takeoff and landing vertical profiles; for European air quality data the folowing data centers were used:
EMEP/EBAS and European Environment Agency/European Topic Center on Air and Climate Change/Air Quality
e-reporting provided European air- and precipitation-chemistry data. The Finnish Meteorological Institute for
providing biomass burning emission data for Europe. Data from meteorological station monitoring networks
were provided by NOAA and Environment Canada (for the US and Canadian meteorological network data) and
the National Center for Atmospheric Research (NCAR) data support section. Joint Research Center
Ispra/Institute for Environment and Sustainability provided its ENSEMBLE system for model output
harmonisation and analyses and evaluation. Although this work has been reviewed and approved for
publication by the U.S. Environmental Protection Agency, it does not necessarily reflect the views and policies
of the agency.
APPENDIX 1
The autocorrelation function (ACF) is derived by the autocovariance (ACV) and expresses the correlation of a
time series with its lagged version (e.g. Chatfield, 2004):
$$ACV(k) = E\{[X(t) - \mu][X(t+k) - \mu]\} = Cov[X(t), X(t+k)];$$
$$ACF(k) = ACV(k)/ACV(0)$$
At any lag $k$, the autocovariance coefficients $c_k$ are given by:

$$c_k = \frac{1}{N} \sum_{t=1}^{N-k} (x_t - \overline{x})(x_{t+k} - \overline{x})$$

And, as usual, the autocorrelation coefficients are given by normalizing $c_k$ with $c_0$.
The partial autocorrelation function (PACF) measures the excess of correlation between two elements of $X(t)$
lagged by $s$ elements not accounted for by the autocorrelation of the intermediate $s-1$ elements. In other
words, the ACF of $X(t)$ and $X(t+s)$ includes all the linear dependence between the intermediate $s-1$ lags. The
PACF allows to investigate the direct effect of lag $t$ on the lag $t+s$.
The advantage of using ACF and PACF is that are function of the lag $k$ only (and not of the specific time $t$). This
condition holds only if $X(t)$ is stationary (i.e. its mean and variance do not change over time). Several tests are
available to check $X(t)$ for stationarity (e.g. Chatfield, 2004). Differencing the time series is typically a way to
achieve stationarity.

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

TABLES
**TABLE 1.** MSE (ppb$^2$) of the full, undecomposed ozone time series (FT) and relative fraction of MSE of the time series derived by filtering
out the diurnal fluctuations (noDU) and of the time series derived by keeping only the diurnal fluctuations (DU). The diurnal signal has
been isolated by applying a filter kz(13,5). The relative fraction of noDU and of DU not adding up to 100% is because the filter allows some
leakage to the nearest frequencies (see Hogrefe et al. (2000) and Solazzo and Galmarini (2016) for details). *a)* NA; *b)*EU
a)

| NA1 | | | NA2 | | | NA3 | | | Continent | | |
|---|---|---|---|---|---|---|---|---|---|---|---|
| CMAQ MSE- Summer | | | | | | | | | | | |
| FT (ppb$^2$) | noDU | DU | FT (ppb$^2$) | noDU | DU | FT (ppb$^2$) | noDU | DU | FT (ppb$^2$) | noDU | DU |
| 28.65 | 40% | 41% | 49.12 | 70% | 23% | 79.35 | 84% | 13% | 28.25 | 56% | 29% |
| CAMQ MSE- Winter | | | | | | | | | | | |
| 86.08 | 94% | 5% | 19.27 | 75% | 21% | 61.67 | 74% | 21% | 22.38 | 85% | 9% |


b)

| EU1 | | | EU2 | | | EU3 | | | Continent | | |
|---|---|---|---|---|---|---|---|---|---|---|---|
| CHIMERE MSE- Summer | | | | | | | | | | | |
| FT (ppb$^2$) | noDU | DU | FT (ppb$^2$) | noDU | DU | FT (ppb$^2$) | noDU | DU | FT (ppb$^2$) | noDU | DU |
| 20.91 | 85% | 10% | 46.19 | 78% | 15% | 125.86 | 26% | 67% | 26.95 | 76% | 18% |

**CHIMERE MSE- Winter**

| 20.87 | 85% | 12% | 19.95 | 85% | 10% | 39.91 | 38% | 59% | 11.34 | 73% | 16% |
|---|---|---|---|---|---|---|---|---|---|---|---|


**TABLE 2.** Linear correlation coefficient between the diurnal residuals of the regressors of Eq 3. The residuals are calculated by removing
from the measured and modelled time series fluctuations faster the ~1.5 days. All the correlation values are significant up to 1%
significance threshold. *a)* NA; *b)* EU. For each set of variables, the regression analysis inlcudes the rural stations within a differential
altitude of maximum 250m.
a)

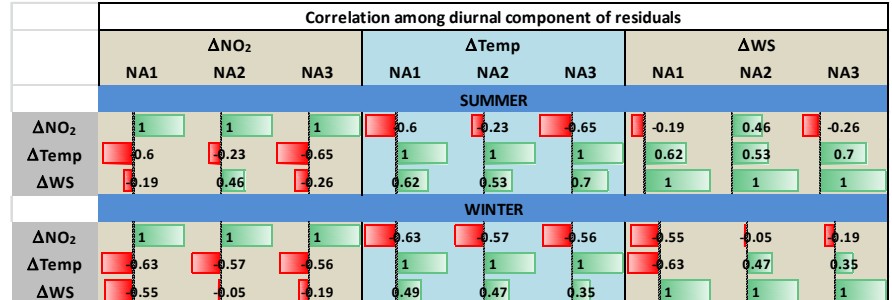

| | Correlation among diurnal component of residuals | | | | | | | | |
|---|---|---|---|---|---|---|---|---|---|
| | ΔNO₂ | | | ΔTemp | | | ΔWS | | |
| | NA1 | NA2 | NA3 | NA1 | NA2 | NA3 | NA1 | NA2 | NA3 |
| | **SUMMER** | | | | | | | | |
| ΔNO₂ | 1 | 1 | 1 | -0.6 | -0.23 | -0.65 | -0.19 | 0.46 | -0.26 |
| ΔTemp | -0.6 | -0.23 | -0.65 | 1 | 1 | 1 | 0.62 | 0.53 | 0.7 |
| ΔWS | -0.19 | 0.46 | -0.26 | 0.62 | 0.53 | 0.7 | 1 | 1 | 1 |
| | **WINTER** | | | | | | | | |
| ΔNO₂ | 1 | 1 | 1 | -0.63 | -0.57 | -0.56 | -0.55 | -0.05 | -0.19 |
| ΔTemp | -0.63 | -0.57 | -0.56 | 1 | 1 | 1 | -0.63 | 0.47 | 0.35 |
| ΔWS | -0.55 | -0.05 | -0.19 | 0.49 | 0.47 | 0.35 | 1 | 1 | 1 |


b)

| | Correlation among diurnal component of residuals | | | | | | | | | | | |
|---|---|---|---|---|---|---|---|---|---|---|---|---|
| | ΔNO | | | ΔNO₂ | | | ΔTemp | | | ΔWS | | |
| | EU1 | EU2 | EU3 | EU1 | EU2 | EU3 | EU1 | EU2 | EU3 | EU1 | EU2 | EU3 |
| | **SUMMER** | | | | | | | | | | | |
| ΔNO | 1 | 1 | 1 | 0.05 | 0.68 | 0.48 | -0.08 | -0.05 | -0.27 | -0.07 | 0.11 | -0.02 |
| ΔNO₂ | 0.05 | 0.68 | 0.48 | 1 | 1 | 1 | 0.57 | 0.18 | -0.27 | 0.51 | 0.38 | 0.26 |
| ΔTemp | -0.08 | -0.05 | -0.27 | 0.57 | 0.18 | -0.27 | 1 | 1 | 1 | 0.81 | 0.63 | 0.21 |
| ΔWS | -0.07 | 0.11 | -0.02 | 0.51 | 0.38 | 0.26 | 0.81 | 0.63 | 0.21 | 1 | 1 | 1 |
| | **WINTER** | | | | | | | | | | | |
| ΔNO | 1 | 1 | 1 | 0.31 | 0.6 | 0.73 | 0.02 | -0.52 | -0.62 | 0.03 | 0.12 | 0.06 |
| ΔNO₂ | 0.31 | 0.6 | 0.73 | 1 | 1 | 1 | -0.13 | 0.7 | 0.7 | -0.01 | 0.09 | 0.11 |
| ΔTemp | 0.02 | -0.52 | -0.62 | -0.13 | 0.7 | 0.7 | 1 | 1 | 1 | 0.48 | 0.02 | -0.01 |
| ΔWS | 0.03 | 0.12 | 0.06 | -0.01 | 0.09 | 0.11 | 0.48 | 0.02 | -0.01 | 1 | 1 | 1 |



**TABLE 3.** Linear correlation coefficient between the residuals of the regressors of Eq 3, when the diurnal fluctuations are filtered out. The
residuals are calculated by removing from the measured and modelled time series fluctuations faster the ~1.5 days. All the correlation
values are significant up to 1% significance threshold. *a)* NA; *b)* EU. For each set of variables, the regression analysis inlcudes the rural
stations within a differential altitude of maximum 250m.
a)

| | Correlation among residuals (diurnal fluctuations removed) | | | | | | | | |
|---|---|---|---|---|---|---|---|---|---|
| | ΔNO₂ | | | ΔTemp | | | ΔWS | | |
| | NA1 | NA2 | NA3 | NA1 | NA2 | NA3 | NA1 | NA2 | NA3 |
| | **SUMMER** | | | | | | | | |
| ΔNO₂ | 1 | 1 | 1 | -0.2 | -0.02 | -0.26 | -0.06 | -0.05 | -0.19 |
| ΔTemp | -0.2 | -0.02 | -0.26 | 1 | 1 | 1 | 0.28 | 0.09 | 0.42 |
| ΔWS | -0.06 | -0.05 | -0.19 | 0.28 | 0.09 | 0.42 | 1 | 1 | 1 |
| | **WINTER** | | | | | | | | |
| ΔNO₂ | 1 | 1 | 1 | -0.12 | -0.42 | -0.03 | -0.02 | -0.16 | -0.11 |
| ΔTemp | -0.12 | -0.42 | -0.03 | 1 | 1 | 1 | 0.54 | 0.34 | 0.13 |
| ΔWS | -0.02 | -0.16 | -0.11 | 0.54 | 0.34 | 0.13 | 1 | 1 | 1 |


b)

| | Correlation among residuals (diurnal fluctuations removed) | | | | | | | | | | | |
|---|---|---|---|---|---|---|---|---|---|---|---|---|
| | ΔNO | | | ΔNO₂ | | | ΔTemp | | | ΔWS | | |
| | EU1 | EU2 | EU3 | EU1 | EU2 | EU3 | EU1 | EU2 | EU3 | EU1 | EU2 | EU3 |
| **SUMMER** | | | | | | | | | | | | |
| ΔNO | 1 | 1 | 1 | 0.22 | 0.71 | 0.69 | 0.12 | -0.23 | -0.03 | 0.06 | -0.23 | -0.08 |
| ΔNO₂ | 0.22 | 0.71 | 0.69 | 1 | 1 | 1 | -0.27 | -0.41 | -0.11 | -0.54 | -0.43 | -0.01 |
| ΔTemp | 0.12 | -0.23 | -0.03 | -0.27 | -0.41 | -0.11 | 1 | 1 | 1 | 0.44 | 0.22 | 0.36 |
| ΔWS | 0.06 | -0.23 | -0.08 | -0.54 | -0.43 | -0.01 | 0.44 | 0.22 | 0.36 | 1 | 1 | 1 |
| **WINTER** | | | | | | | | | | | | |
| ΔNO | 1 | 1 | 1 | 0.21 | 0.64 | 0.46 | -0.22 | -0.19 | -0.02 | -0.15 | -0.14 | -0.01 |
| ΔNO₂ | 0.21 | 0.64 | 0.46 | 1 | 1 | 1 | -0.09 | -0.38 | -0.35 | -0.07 | -0.2 | -0.08 |
| ΔTemp | -0.22 | -0.19 | -0.02 | -0.09 | -0.38 | -0.35 | 1 | 1 | 1 | 0.37 | -0.1 | 0.38 |
| ΔWS | -0.15 | -0.14 | -0.01 | -0.07 | -0.2 | -0.08 | 0.37 | -0.1 | 0.38 | 1 | 1 | 1 |


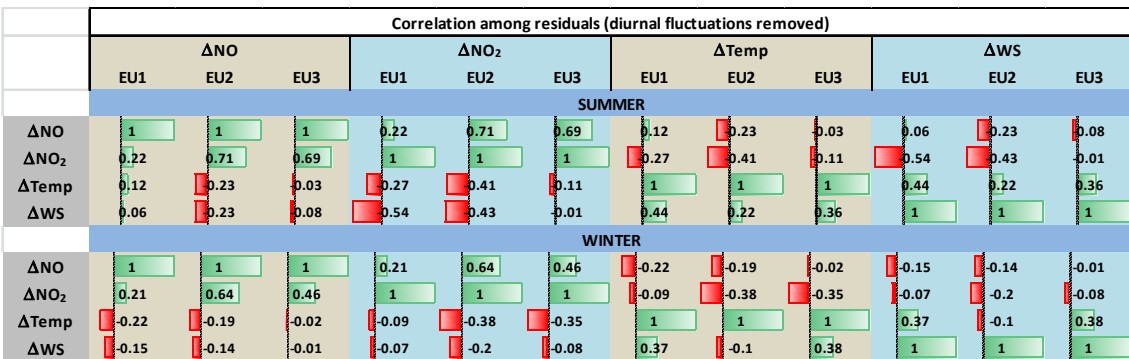

## FIGURES CAPTIONS

**Figure 1** Continental domains and sub-regions used for analysis. The networks of ozone receptors are also shown.

**Figure 2**. Average monthly (right column of panels) and diurnal curves (left column of panels) constructed from January – December 2010 time series of hourly ozone observations and model simulations for three North American sub-regions

**Figure 3**. Average monthly (right column of panels) and diurnal curves (left column of panels) constructed from January – December 2010 time series of hourly ozone observations and model simulations for three European sub-regions.

**Figure 4** MSE decomposition for June – August hourly ozone into bias$^2$, variance and covariance for the three NA sub-regions. Results are presented separately for daylight hours (left) and night-time hours (right).

**Figure 5** MSE decomposition for June – August hourly ozone into bias$^2$, variance and covariance for the three EU sub-regions (the zero_Dep data refers to the month of July only). Results are presented separately for daylight hours (left) and nighttime hours (right)

**Figure 6** CMAQ MSE breakdown for summer and winter for the base case and sensitivity simulations over NA. The error coeffcients $F_b,F_v,F_c$ are reported on the left axis, the total MSE (ppb$^2$) on the right axis (red triangles). The '+' and '-' signs within the bias and variance portions of the errors indicate model over- or under-prediction of mean concentration or variance, respectively. The values in the covariance portion indicate the correlation coeffcient between modelled and observed time series. *a)* hourly time series of ozone (base case); *b)* hourly time series of '20% reduction' scenario; *c)* hourly time series of 'zero boundary conditions' scenario; *d)* hourly time series of the 'zeroed anthropogenic emissions' scenario; *e)* base case rolling average daily maximum 8-hour ozone time series. For the analysis of hourly time series in panels a) – d), results are provided separately for daytime and nighttime.

**Figure 7**. Chimere MSE breakdown for summer and winter for the base case and sensitivity simulations over EU. The error coeffcients $F_b,F_v,F_c$ are reported on the left axis, the total MSE (ppb$^2$) on the right axis (red triangles). The '+' and '-' signs within the bias and variance portions of the errors indicate model over- or under-prediction of mean concentration or variance, respectively. The values in the covariance portion indicate the correlation coeffcient between modelled and observed time series. *a)* hourly time series of ozone (base case); *b)* hourly time series of '20% reduction' scenario; *c)* hourly time series of 'constant boundary conditions' scenario; *d)* hourly time series of the 'zeroed anthropogenic emissions' scenario; *e)* base case rolling average daily maximum 8-hour ozone time series. For the analysis of hourly time series in panels a) – d), results are provided separately for daytime and nighttime.

**Figure 8**. Top row: Spatial maps of RMSE (in ppb) for the base case. Middle row: Percentage RMSE changes for the zeroed emissions case with respect to the base case. Lower row: Percentage RMSE changes for the zeroed boundary condition case with respect to the base case. Left column: Winter months (DJF); Right column: summer months (JJA).

**Figure 9** Top row: Spatial maps of RMSE (in ppb) for the base case. Middle row: Percentage RMSE changes for the zeroed emissions case with respect to the base case. Lower row: Percentage RMSE changes for the constant boundary condition case with respect to the base case.. Left column: Winter months (DJF); Right column: summer months (JJA).

**Figure 10**. Annual time series of differences between CMAQ and observed $O_3$ ($\Delta O_3$, top panel) and Morlet wavelet analysis of the periodogram of $\Delta O_3$ (lower panel) for the three NA subdomains. Black contours lines identify the 95% confidence interval. The period (in days) is reported in the vertical axis, while the quantiles of the power spectral density are measured in $ppb^2$. (the scale reports the quantiles of the power spectrum).

**Figure 11.** Same as in FIGURE 10 for Chimere over the three EU subdomains

**Figure 12.** CMAQ model: autocorrelation (ACF) and partial autocorrelation (PACF) function for *a)* the differenced time series of residuals of ozone (mod-obs) and *b)* the differenced time series of residual of ozone obtained by filtering out the diurnal fluctuations from the modelled and observed time series. The differentiation is necessary to remove non-stationarity and thus to make the ACF and PACF values depending on lag only.

**Figure 13**. Chimere model: autocorrelation (ACF) and partial autocorrelation (PACF) function for *a)* the differenced time series of residuals of ozone (mod-obs) and *b)* the differenced time series of residual of ozone obtained by filtering out the diurnal fluctuations from the modelled and observed time series. The differentiation is necessary to remove non-stationarity and thus to make the ACF and PACF values depending on lag only.

**Figure 14**. Phase shift of the diurnal cycle (in hours). A positive phase shift indicates that the model peak is 'late', while a negative phase shift indicates that the modelled peak precedes the observed peak. This analysis includes urban and suburban stations in addition to rural stations.

**Figure 15.** As in Figure 14 for EU.

**Figure 16**. Normalised MSE produced by lagging the observed diurnal cycle with respect to itself. The MSE due to such a shift is entirely due to covariance error. The plots are presented for EU2 (left) and NA2 (right) for the months of JJA. The top panel shows the impact of the phase shift on the DU component, and the lower panels show results for the undecomposed time series (FT). For EU2, a shift of $\pm 3$ hours causes an MSE of ~0.5 times the variance of the observations.

**Figure 17**. Percentage of variance explained by the regressors (the total $R^2$ for the regression is reported in the title of each panel). The relative importance of each variable is assessed by using a bootstrap resampling. The plots at the bottom show the ACF and PACF of the yearly time series of residual of the fit, i.e. the portion of the ozone time series that was not captured by the linear regressions on the available variables. **The analysis encompasses 47 co-located stations (the NA stations for ozone, $NO_2$, WS, and Temp that fall in a radius of 1000 m and vertical displacement less than 250m).**

**Figure 18**. Same as **Figure 17** for EU. **The analysis encompasses 61 co-located stations (the EU stations for ozone, NO, $NO_2$, WS, and Temp that fall in a radius of 1000 m and vertical displacement less than 250m).**

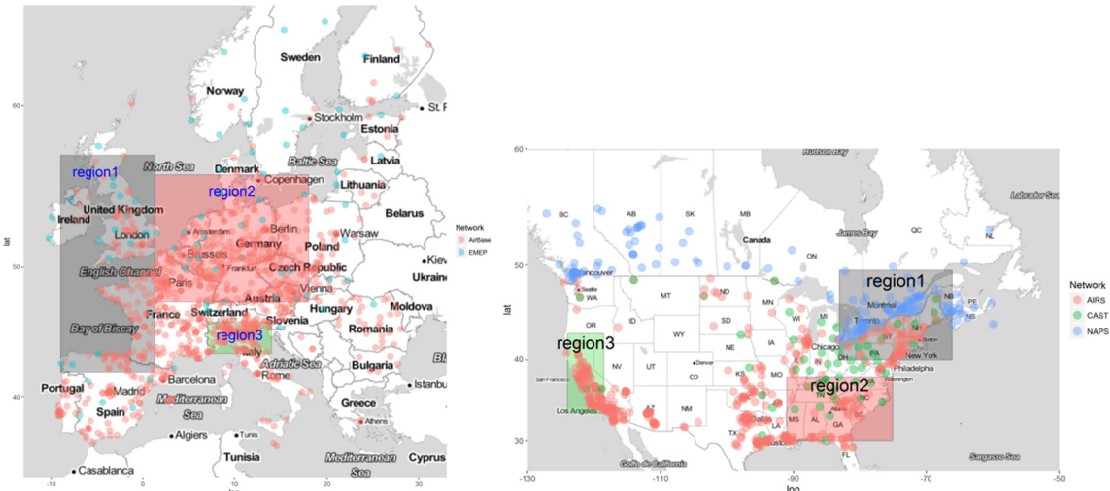

**FIGURE 1.** Continental domains and sub-regions used for analysis. The networks of ozone receptors are also shown

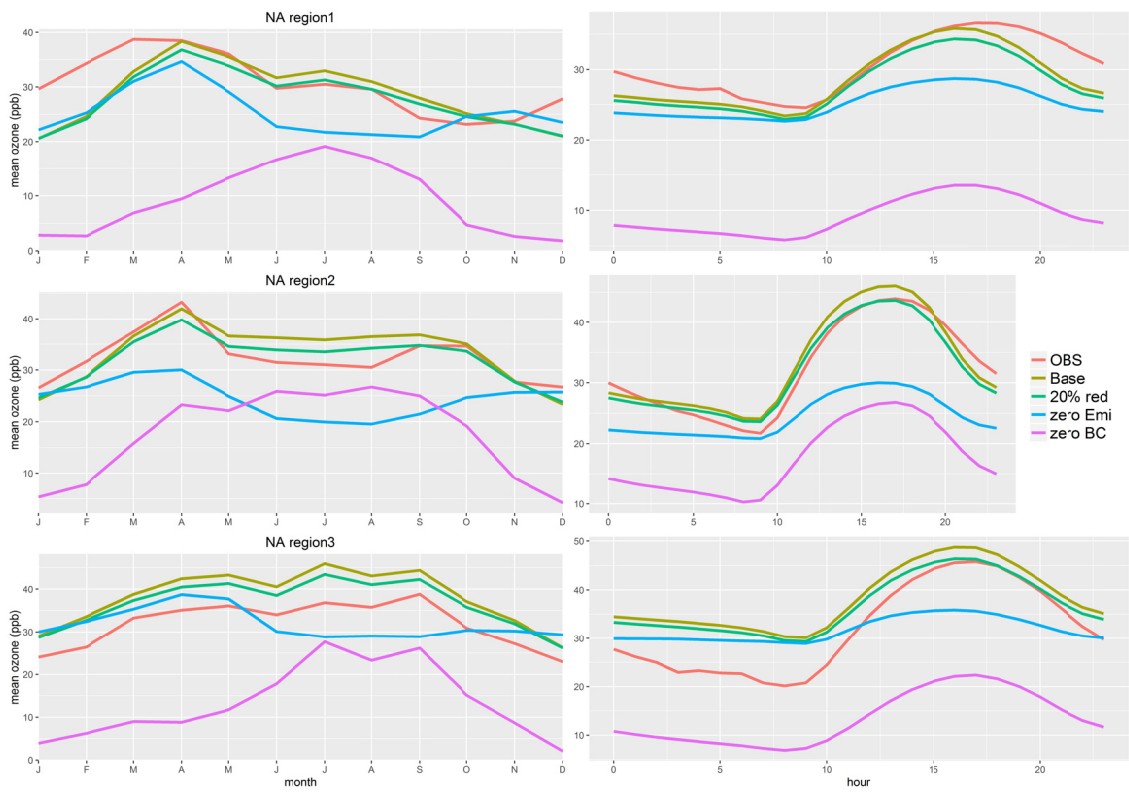

**FIGURE 2.** Average monthly (right column of panels) and diurnal curves (left column of panels) constructed from
January – December 2010 time series of hourly ozone observations and model simulations for three North
1037 American sub-regions

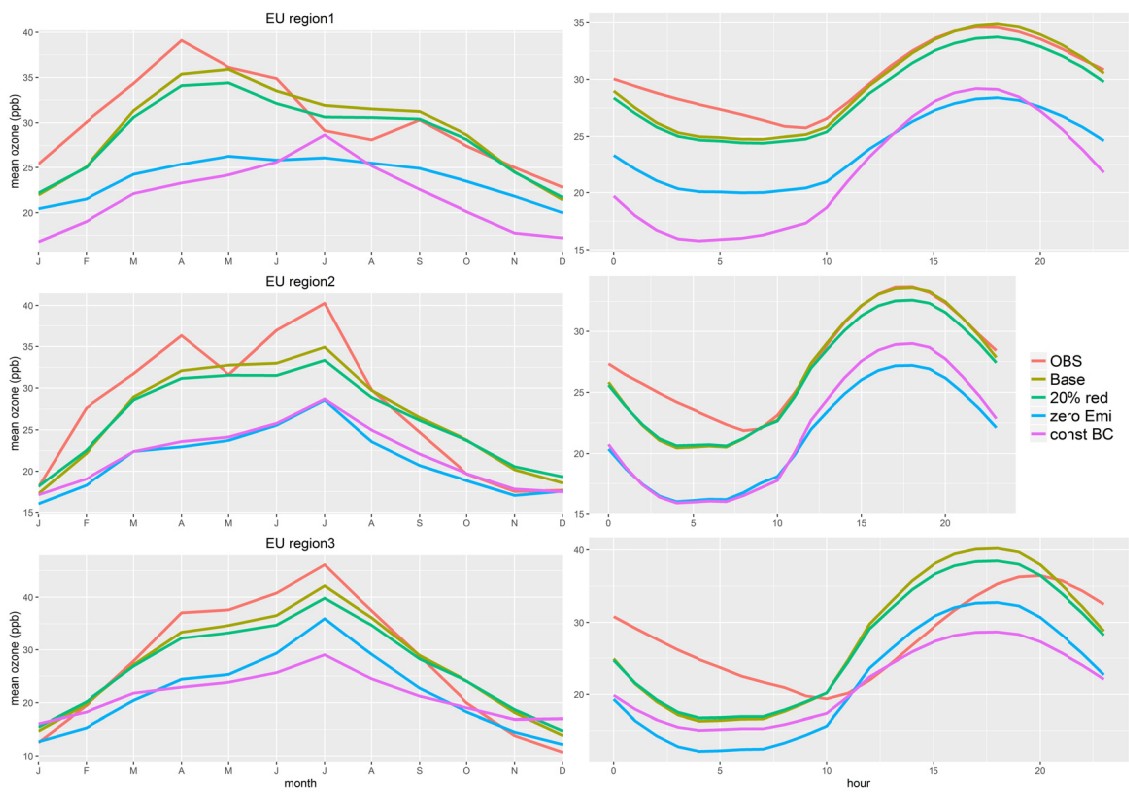

1035
1036

**FIGURE 3**. Average monthly (right column of panels) and diurnal curves (left column of panels) constructed from January – December 2010
time series of hourly ozone observations and model simulations for three European sub-regions.









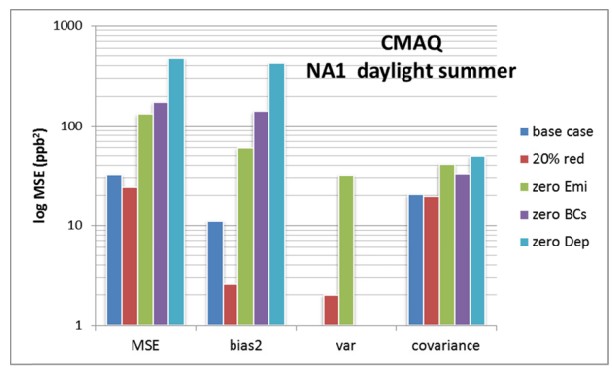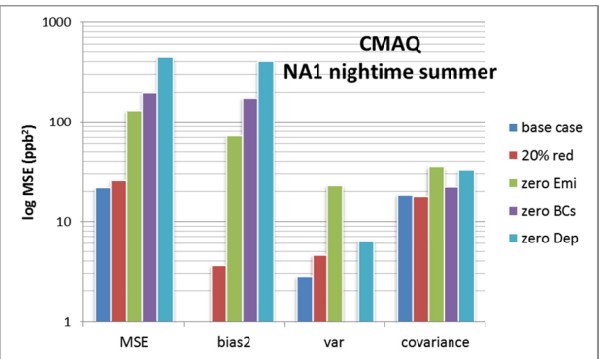
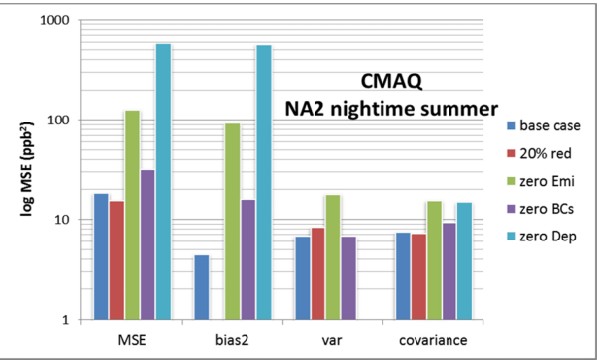
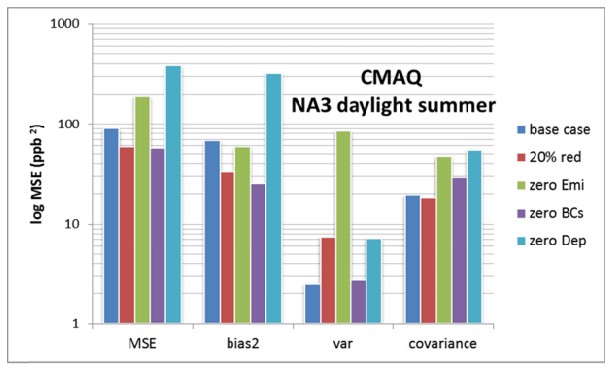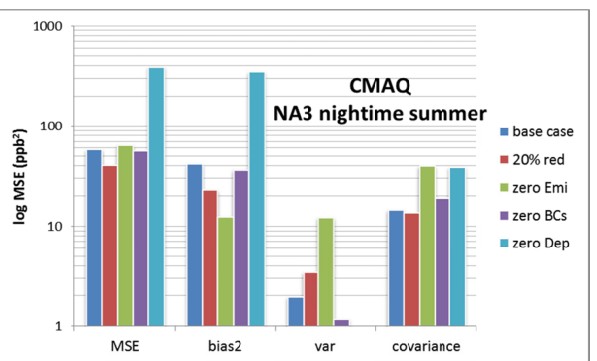

**FIGURE 4**. MSE decomposition for June – August hourly ozone into bias[2], variance and covariance for the three NA sub-regions. Results are
presented separately for daylight hours (left) and night-time hours (right)




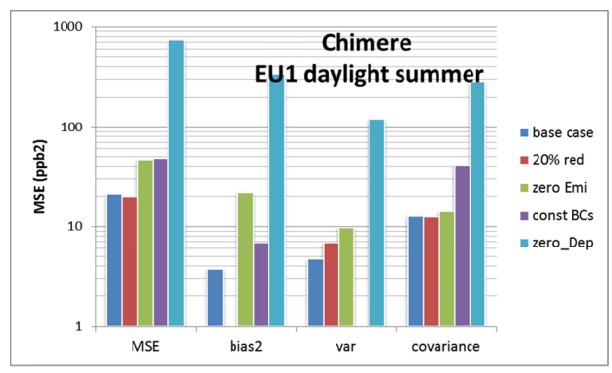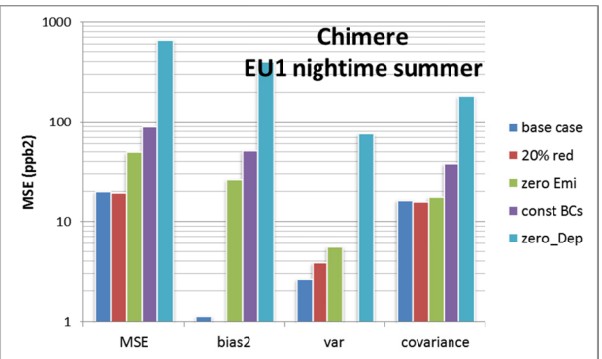
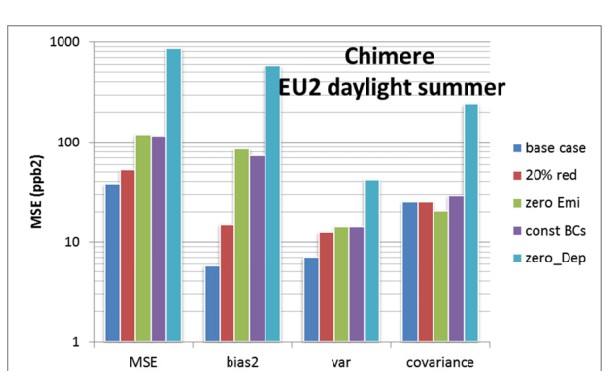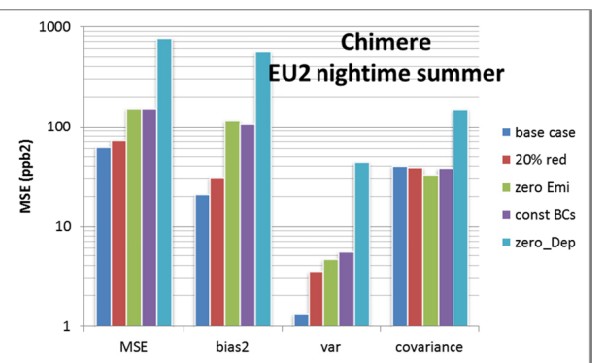
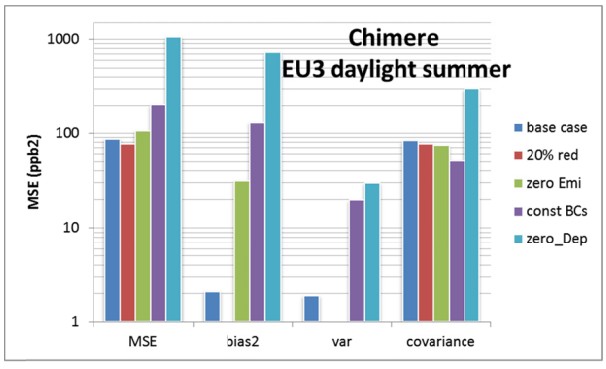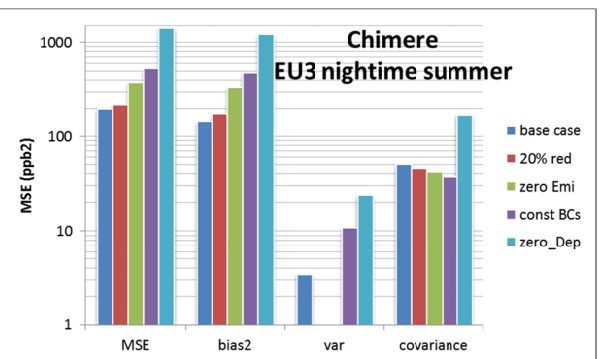
**FIGURE 5.** MSE decomposition for June – August hourly ozone into bias$^2$, variance and covariance for the three EU sub-regions (the
'zero_Dep' data refers to the month of July only). Results are presented separately for daylight hours (left) and night-time hours (right)

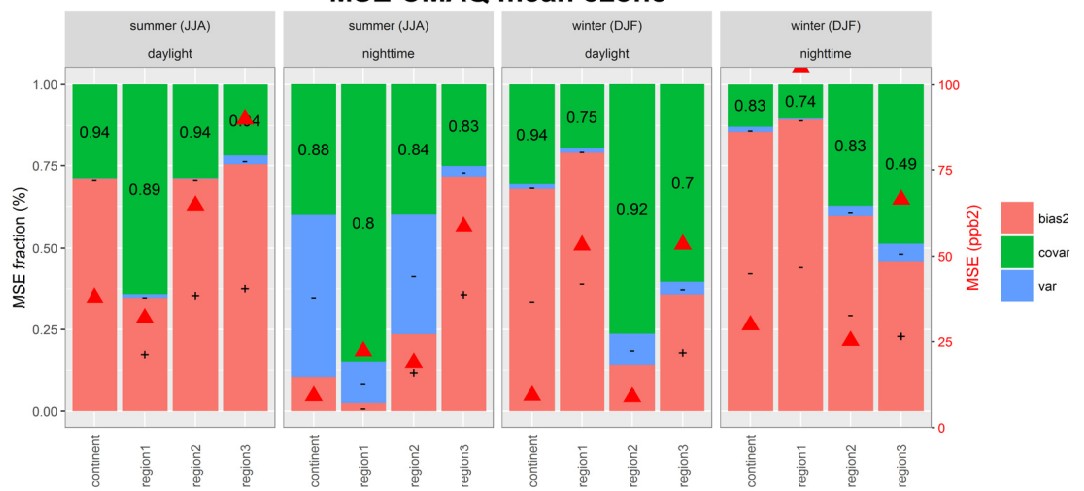


a)

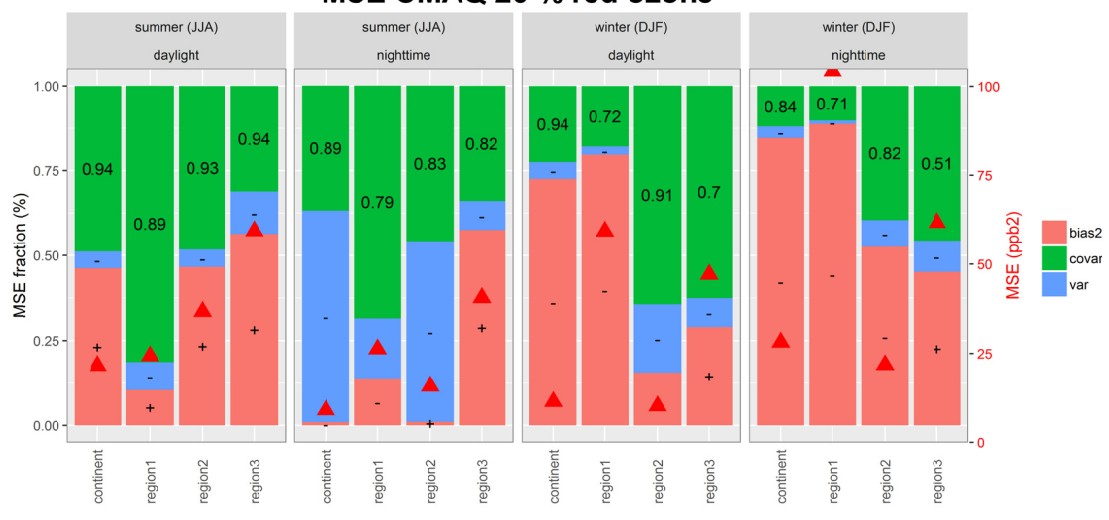

b)

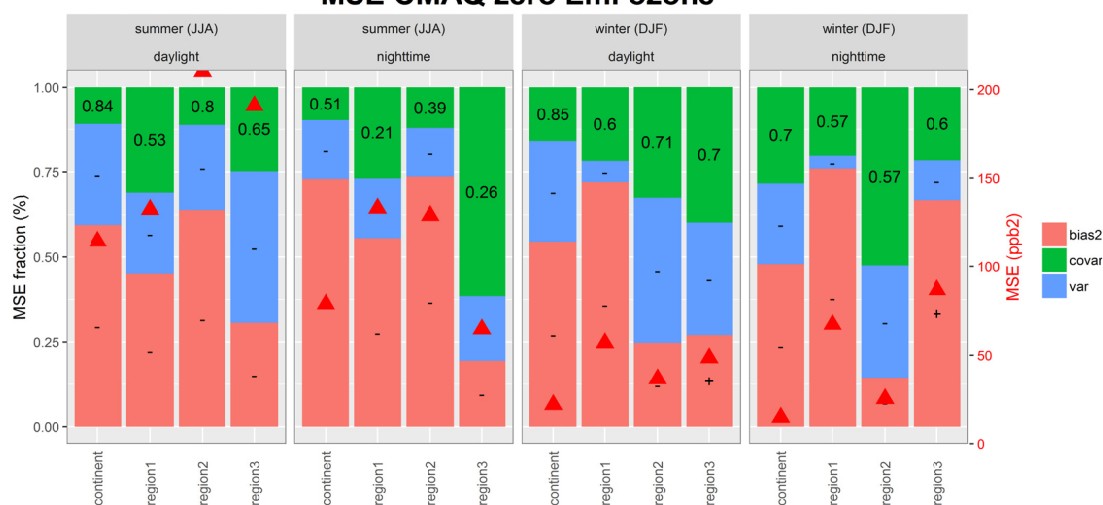

c)

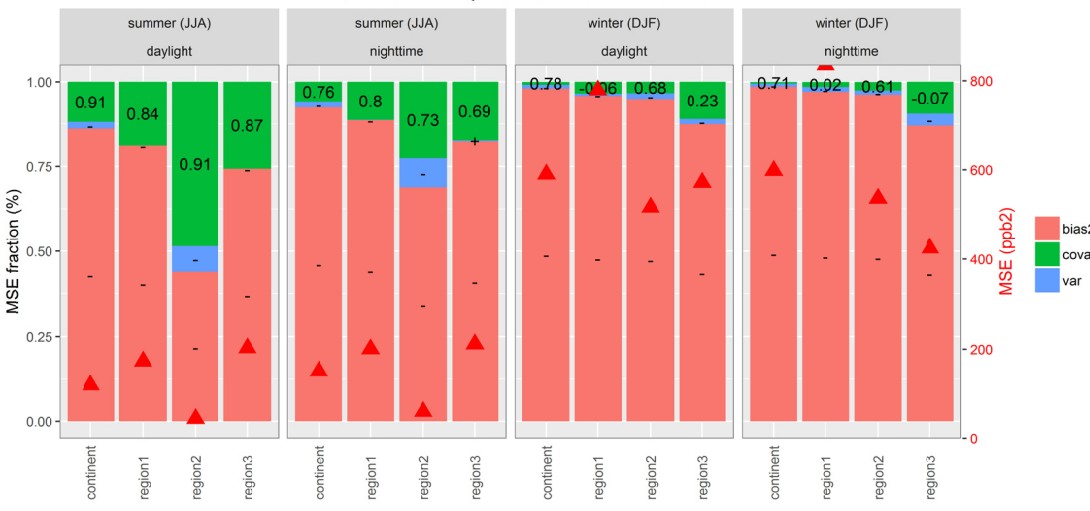

d)

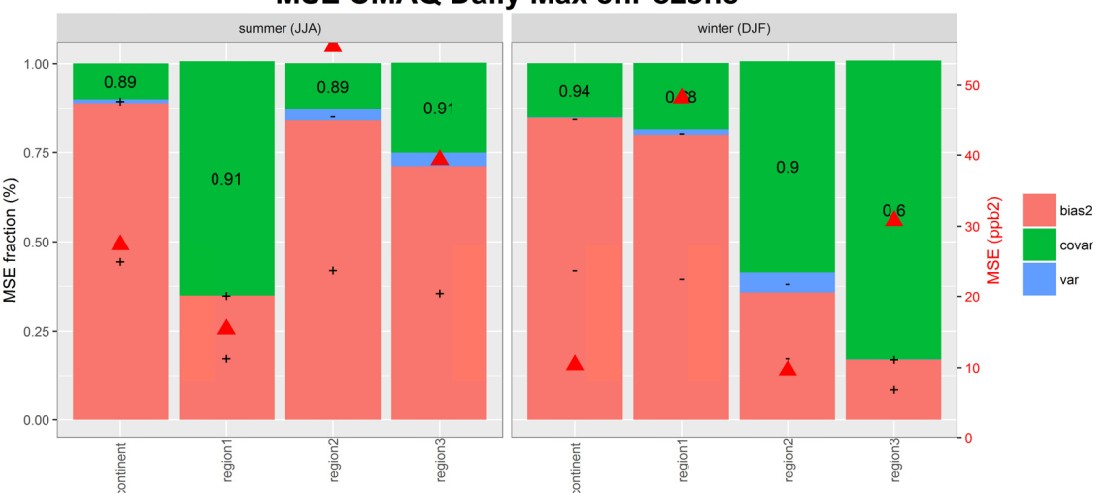

**FIGURE 6.** CMAQ MSE breakdown for summer and winter for the base case and sensitivity simulations over NA. The error coeffcients
$F_b$,$F_v$,$F_c$ are reported on the left axis, the total MSE (ppb$^2$) on the right axis (red triangles). The '+' and '-' signs within the bias and variance
portions of the errors indicate model over- or under-prediction of mean concentration or variance, respectively. The values in the
covariance portion indicate the correlation coeffcient between modelled and observed time series. *a)* hourly time series of ozone (base
case); *b)* hourly time series of '20% reduction' scenario; *c)* hourly time series of 'zero boundary conditions' scenario; *d)* hourly time series
of the 'zeroed anthropogenic emissions' scenario; *e)* base case rolling average daily maximum 8-hour ozone time series. For the analysis of
hourly time series in panels a) – d), results are provided separately for daytime and nighttime.  .

## MSE Chimere mean ozone

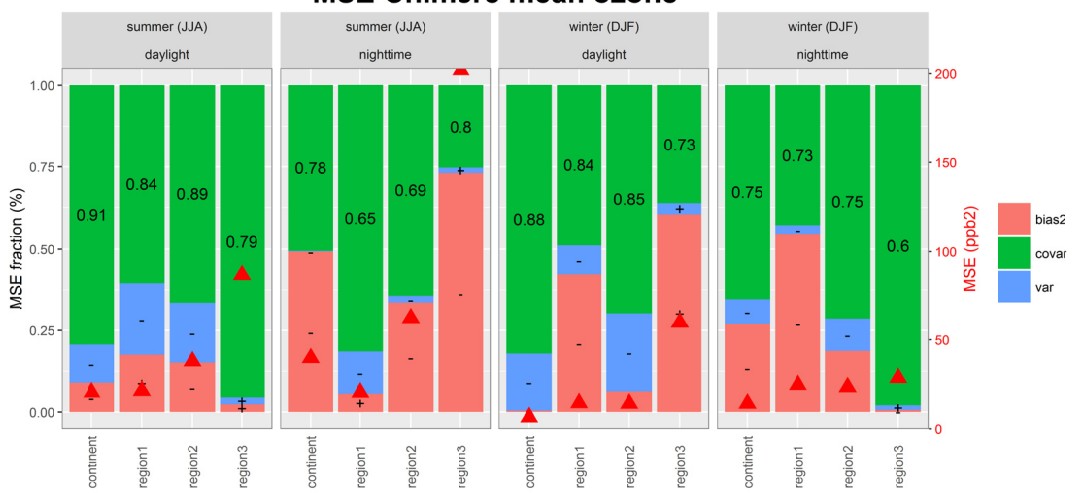

a)

## MSE Chimere 20 % red ozone

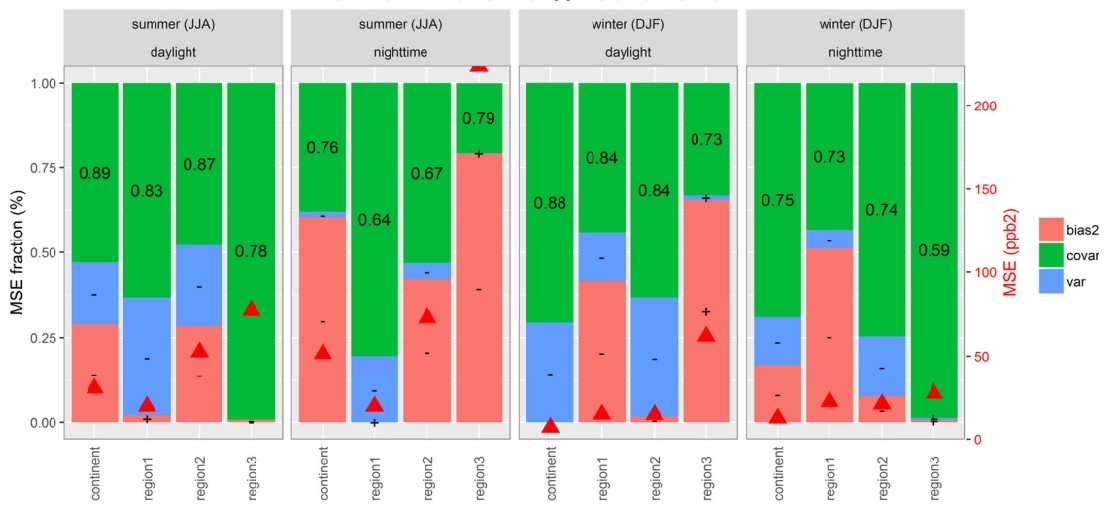

b)

## MSE Chimere zero Emi ozone

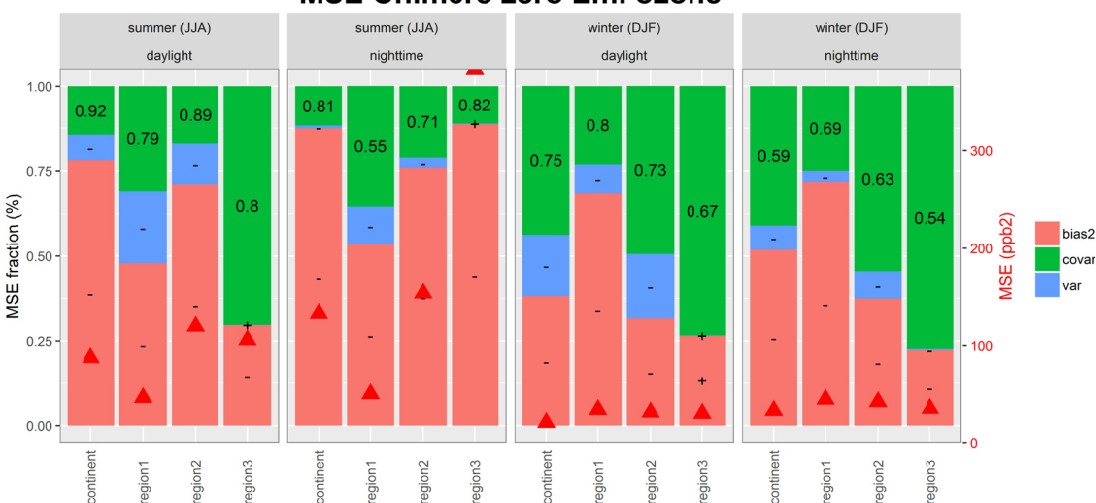

c)

1080

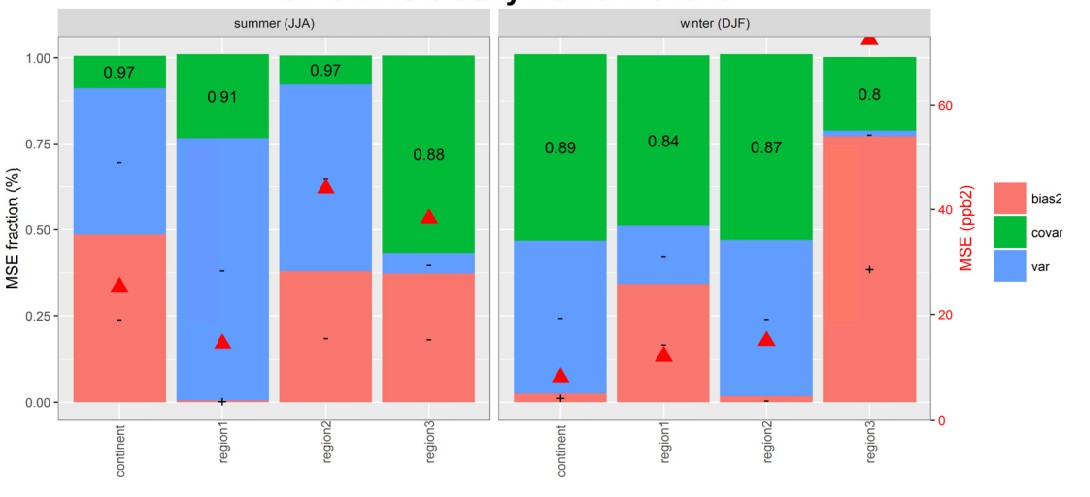

1084
1085    d)

e)

**FIGURE 7.** Chimere MSE breakdown for summer and winter for the base case and sensitivity simulations over EU. The error coeffcients
$F_b, F_v, F_c$ are reported on the left axis, the total MSE (ppb$^2$) on the right axis (red triangles). The '+' and '-' signs within the bias and variance
portions of the errors indicate model over- or under-prediction of mean concentration or variance, respectively. The values in the
covariance portion indicate the correlation coeffcient between modelled and observed time series. *a)* hourly time series of ozone (base
case); *b)* hourly time series of '20% reduction' scenario; *c)* hourly time series of 'constant boundary conditions' scenario; *d)* hourly time
series of the 'zeroed anthropogenic emissions' scenario; *e)* base case rolling average daily maximum 8-hour ozone time series. For the
analysis of hourly time series in panels a) – d), results are provided separately for daytime and nighttime.  .

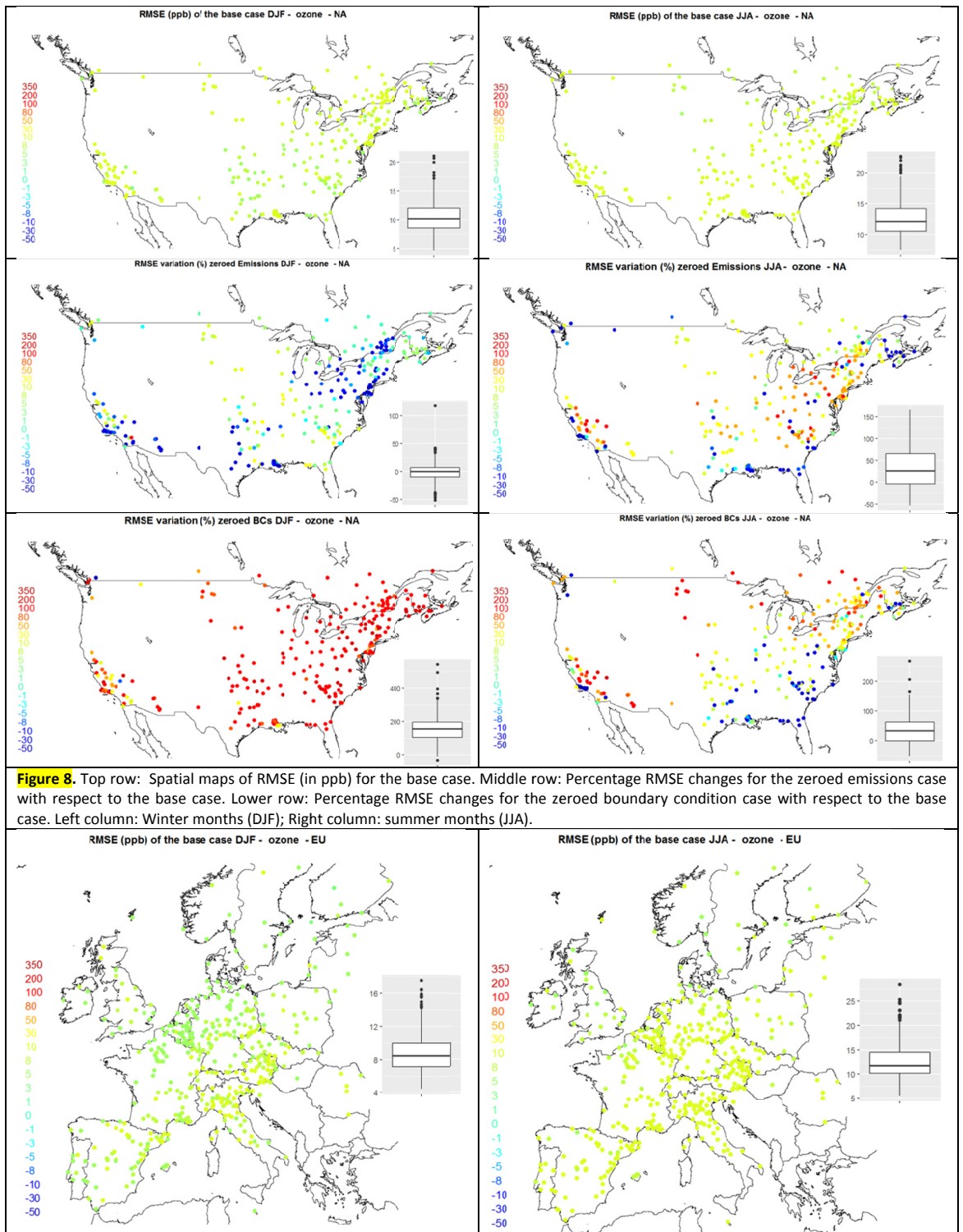

**Figure 8.** Top row: Spatial maps of RMSE (in ppb) for the base case. Middle row: Percentage RMSE changes for the zeroed emissions case with respect to the base case. Lower row: Percentage RMSE changes for the zeroed boundary condition case with respect to the base case. Left column: Winter months (DJF); Right column: summer months (JJA).

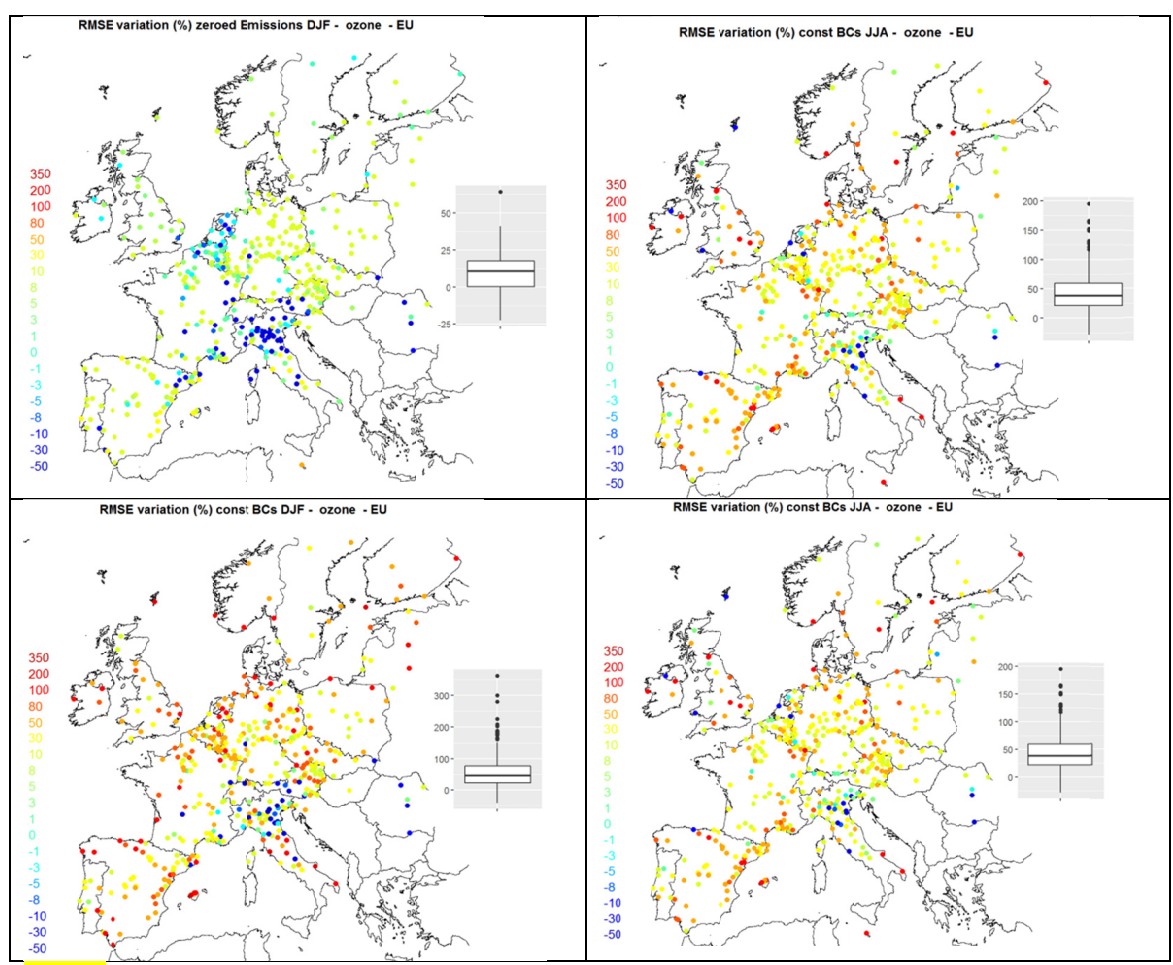

 **FIGURE 9.** Top row:  Spatial maps of RMSE (in ppb) for the base case. Middle row: Percentage RMSE changes for the zeroed emissions case
with respect to the base case. Lower row: Percentage RMSE changes for the constant boundary condition case with respect to the base
case. Left column: Winter months (DJF); Right column: summer months (JJA).



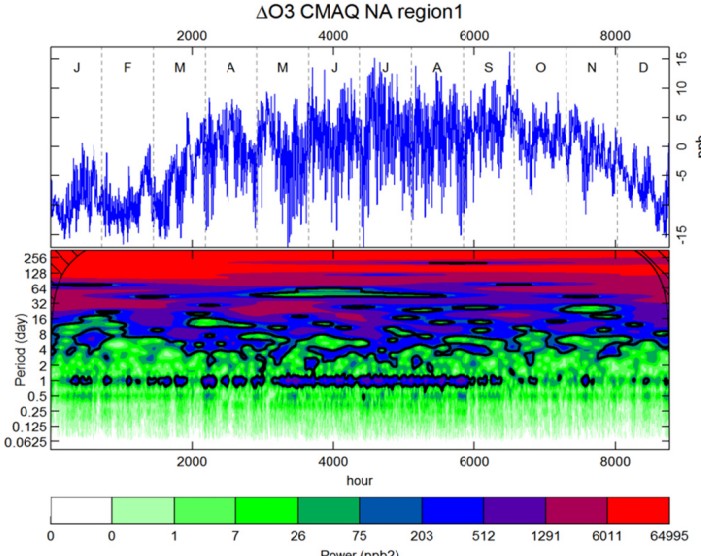


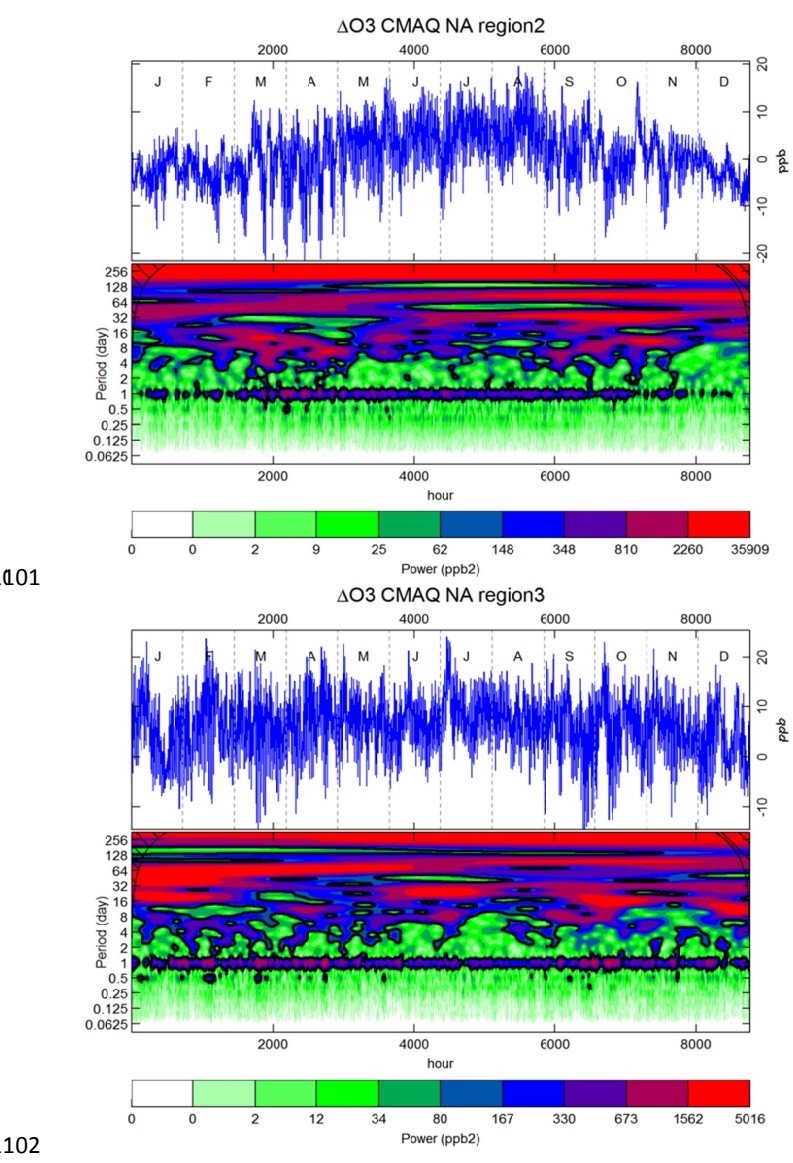



**FIGURE 10.** Annual time series of differences between CMAQ and observed O$_3$ ($\Delta$O$_3$, top panel) and Morlet wavelet analysis of the
periodogram of $\Delta$O$_3$ (lower panel) for the three NA subdomains. Black contours lines identify the 95% confidence interval. The period (in
days) is reported in the vertical axis, while the quantiles of the power spectral density are measured in ppb$^2$.

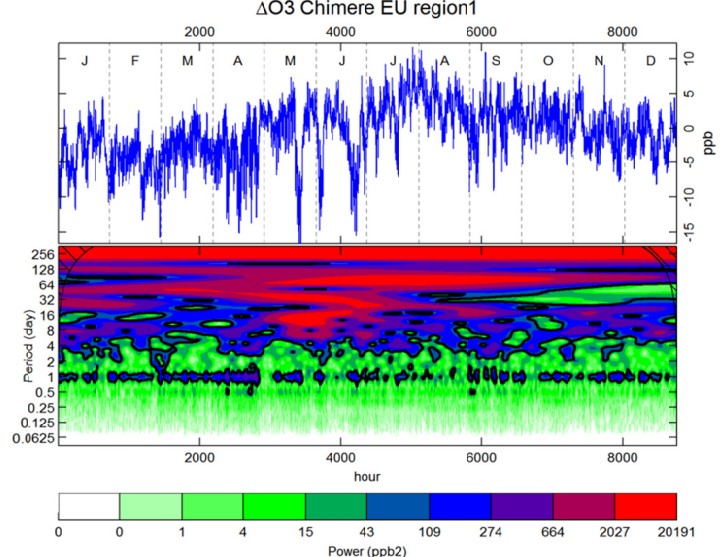

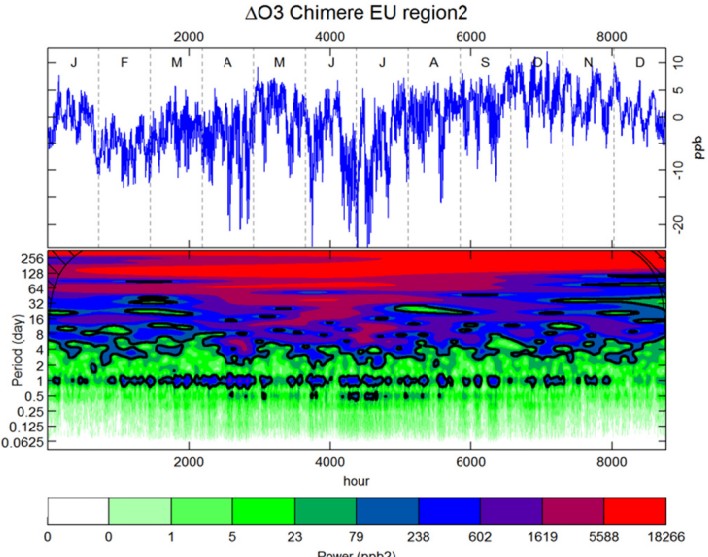

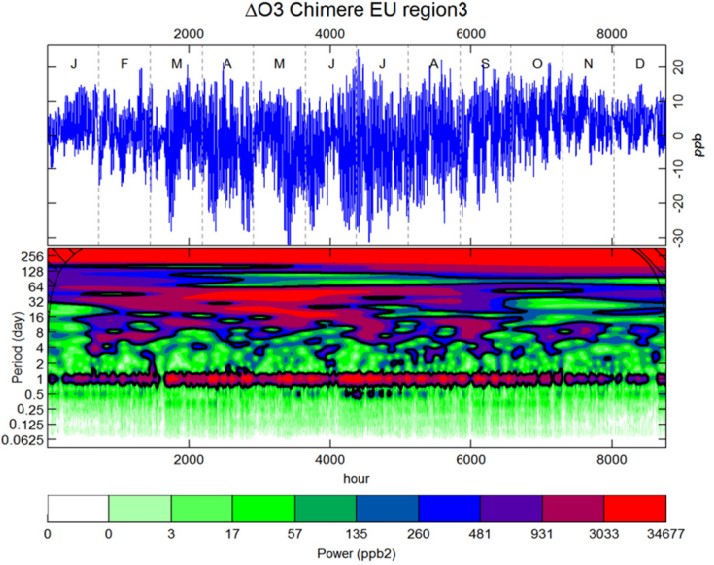

**FIGURE 11.** As in Figure 10 for Chimere over the three EU subdomains.

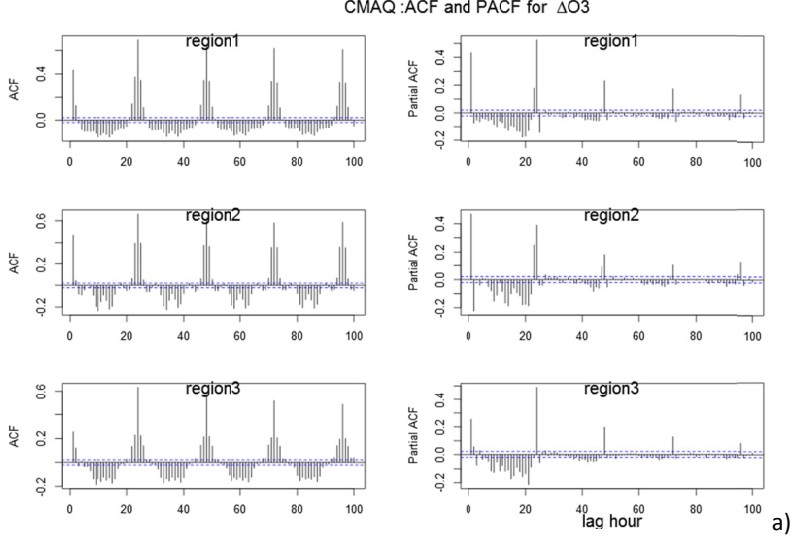

a)

CMAQ :ACF and PACF for ΔO3 (no diurnal fluctuations)

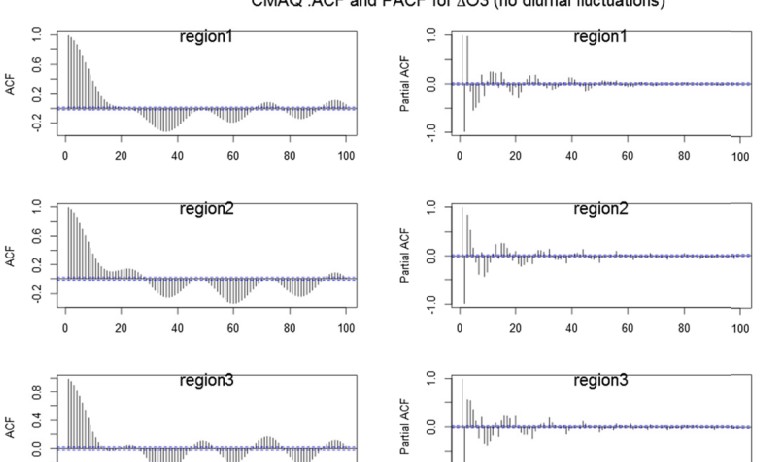

b)

**FIGURE 12.** CMAQ model: autocorrelation (ACF) and partial autocorrelation (PACF) function for *a)* the differenced time series of residuals
of ozone (mod-obs) and *b)* the differenced time series of residual of ozone obtained by filtering out the diurnal fluctuations from the
modelled and observed time series. The differentiation is necessary to remove non-stationarity and thus to make the ACF and PACF values
depending on lag only.

Chimere ACF and PACF for ΔO3

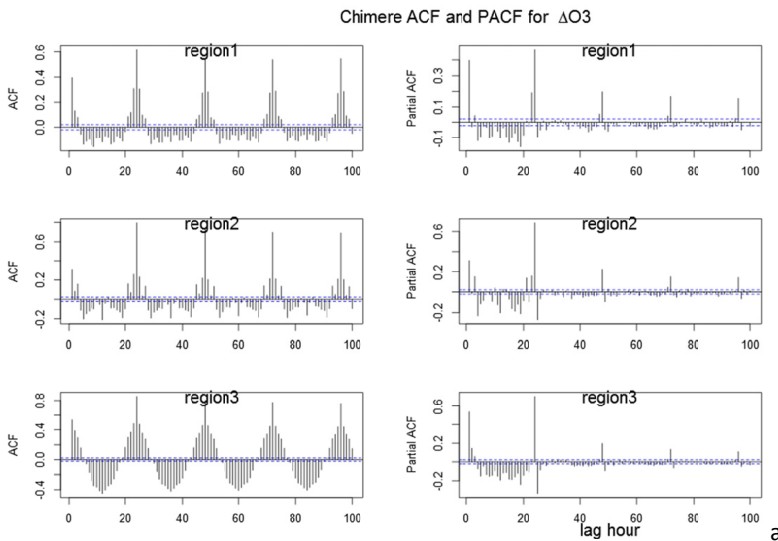

a)

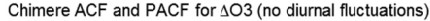

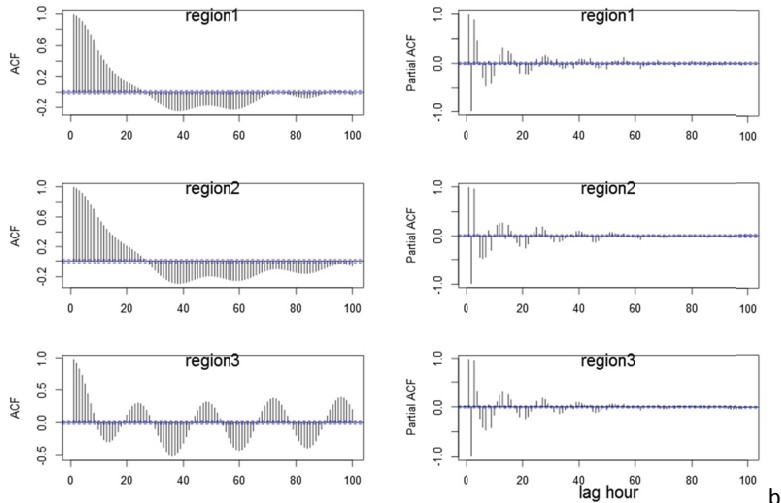

**FIGURE 13.** Chimere model: autocorrelation (ACF) and partial autocorrelation (PACF) function for *a)* the differenced time series of
residuals of ozone (mod-obs) and *b)* the differenced time series of residual of ozone obtained by filtering out the diurnal fluctuations from
the modelled and observed time series. The differentiation is necessary to remove non-stationarity and thus to make the ACF and PACF
values depending on lag only.

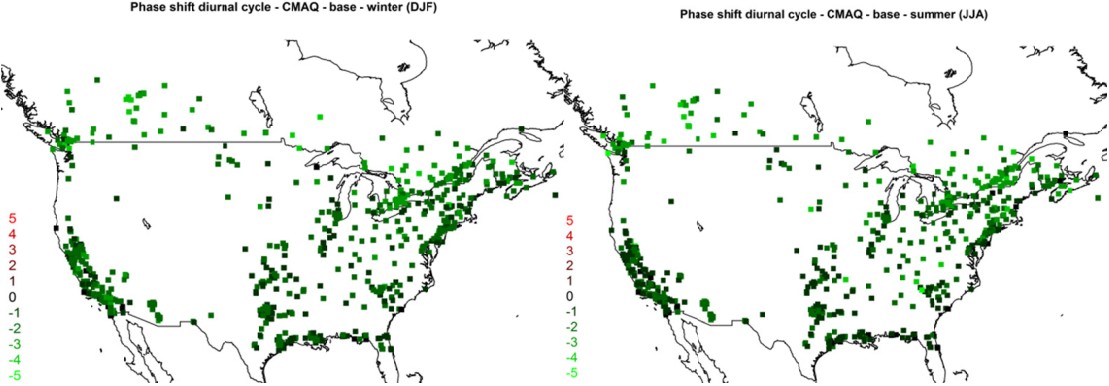

**Figure 14.** Phase shift of the diurnal cycle (in hours). A positive phase shift indicates that the model peak is 'late', while a negative phase
shift indicates that the modelled peak precedes the observed peak. This analysis includes urban and suburban stations in addition to rural
stations.

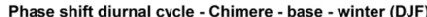

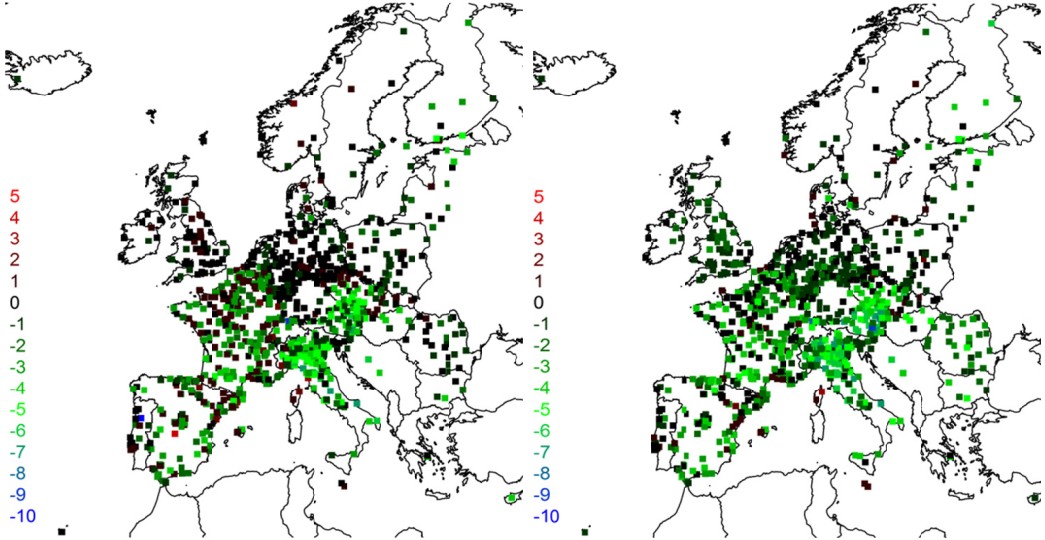

**FIGURE 15.** As in Figure 14 for EU.

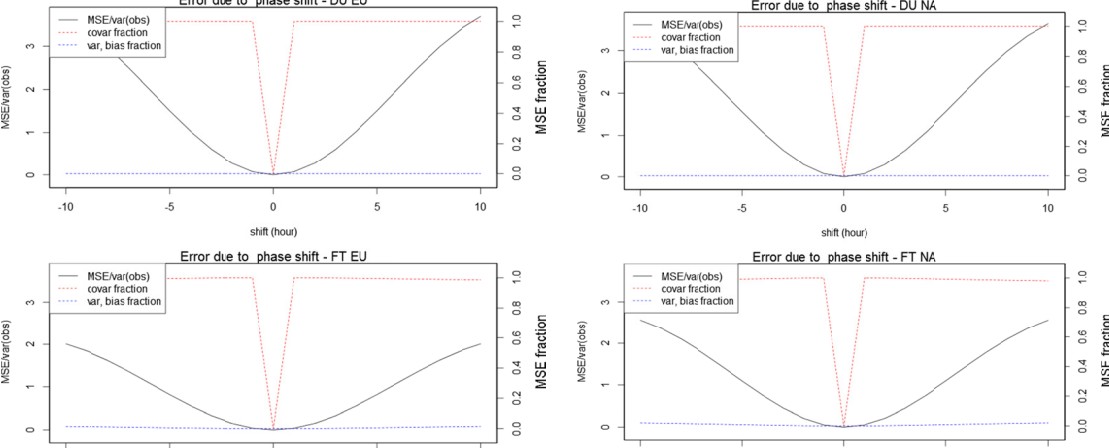

**FIGURE 16.** Normalised MSE produced by lagging the observed diurnal cycle with respect to itself. The MSE due to such a shift is entirely
due to covariance error. The plots are presented for EU2 (left) and NA2 (right) for the months of JJA. The top panel shows the impact of
the phase shift on the DU component, and the lower panels show results for the undecomposed time series (FT). For EU2, a shift of ±3
hours causes an MSE of ~0.5 times the variance of the observations.

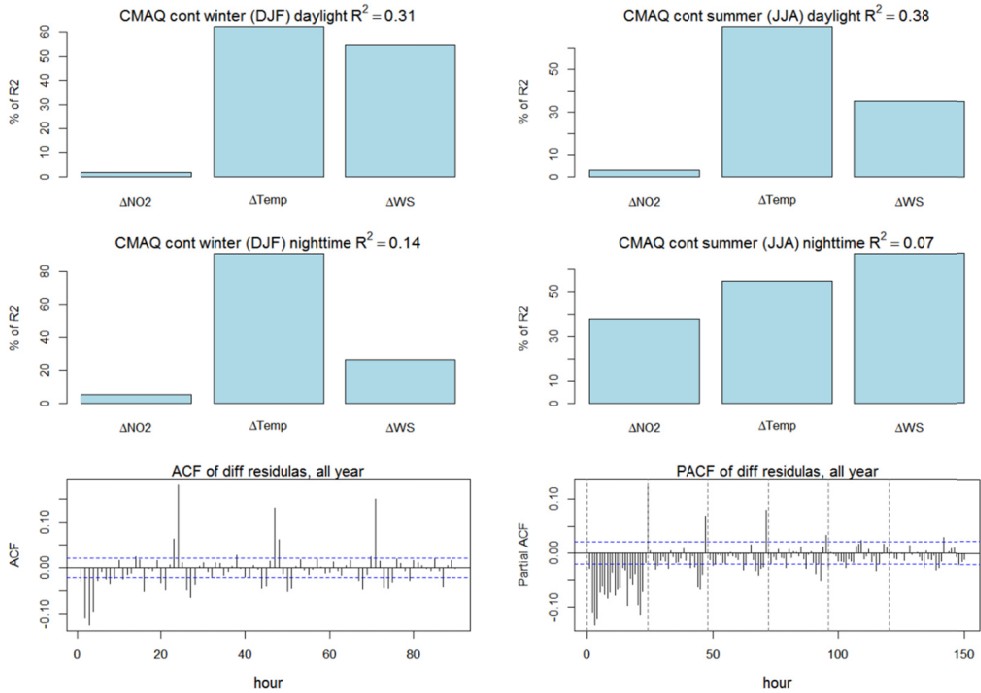

**FIGURE 17.** Percentage of variance explained by the regressors (the total $R^2$ for the regression is reported in the title of each panel). The
relative importance of each variable is assessed by using a bootstrap resampling. The plots at the bottom show the ACF and PACF of the
yearly time series of residual of the fit, i.e. the portion of the ozone time series that was not captured by the linear regressions on the
available variables. The analysis encompasses 47 co-located stations (the NA stations for ozone, $NO_2$, WS, and Temp that fall in a radius of
1000 m and vertical displacement less than 250m).


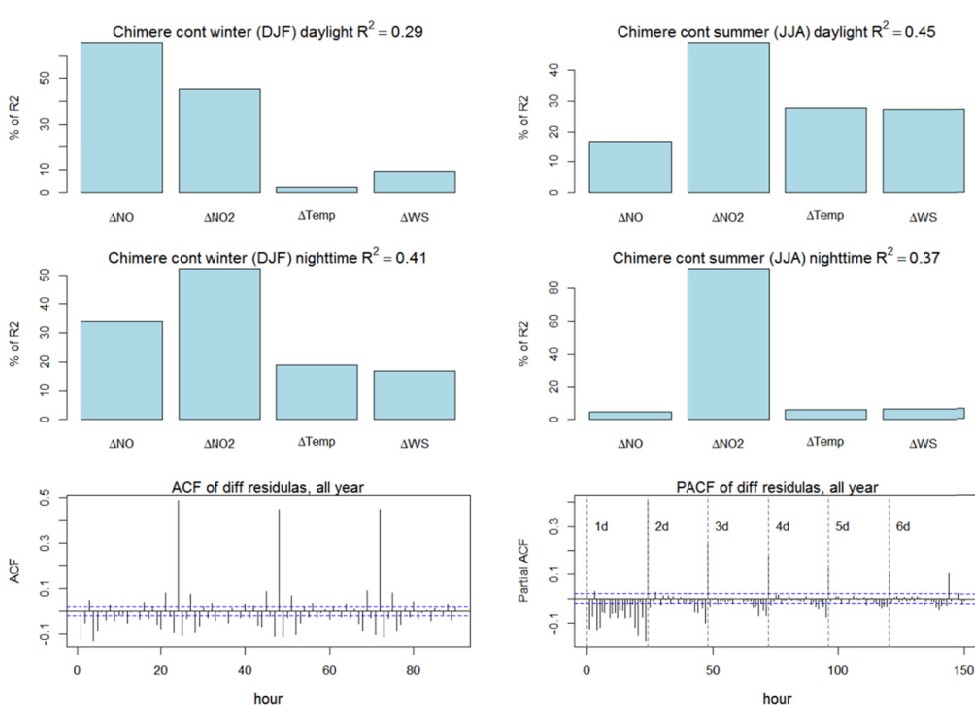



# SUPPLEMENTARY FIGURES

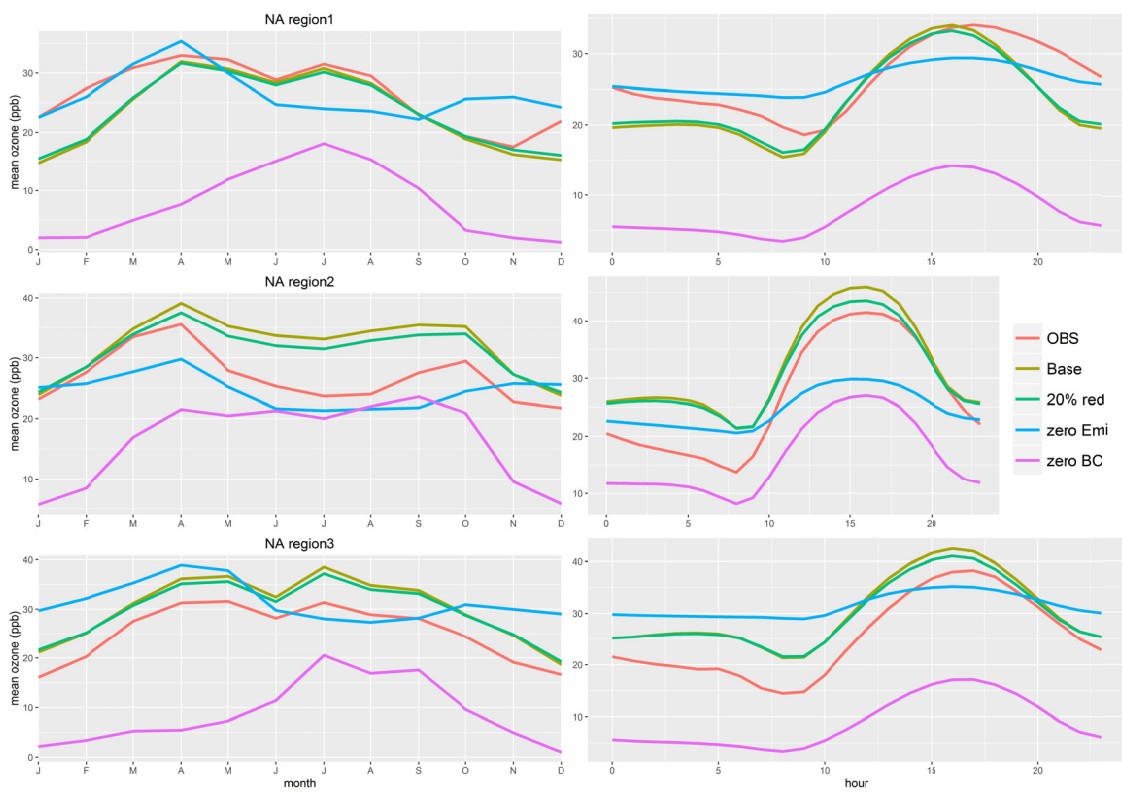


**FIGURE S1** Annual aggregated ozone time series for the North American sub-regions (Urban stations only).
Average monthly and diurnal curves



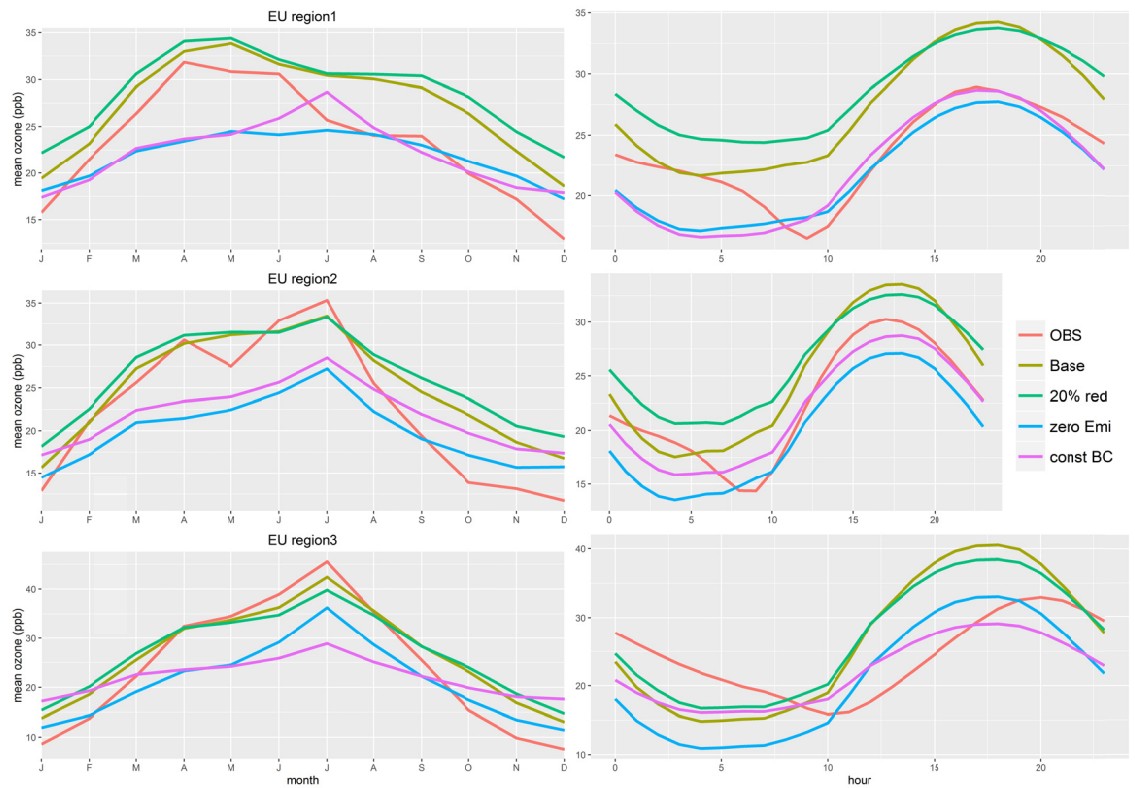


**FIGURE S2.** Annual aggregated ozone time series for the European sub-regions (urban stations only). Average
monthly and diurnal curves

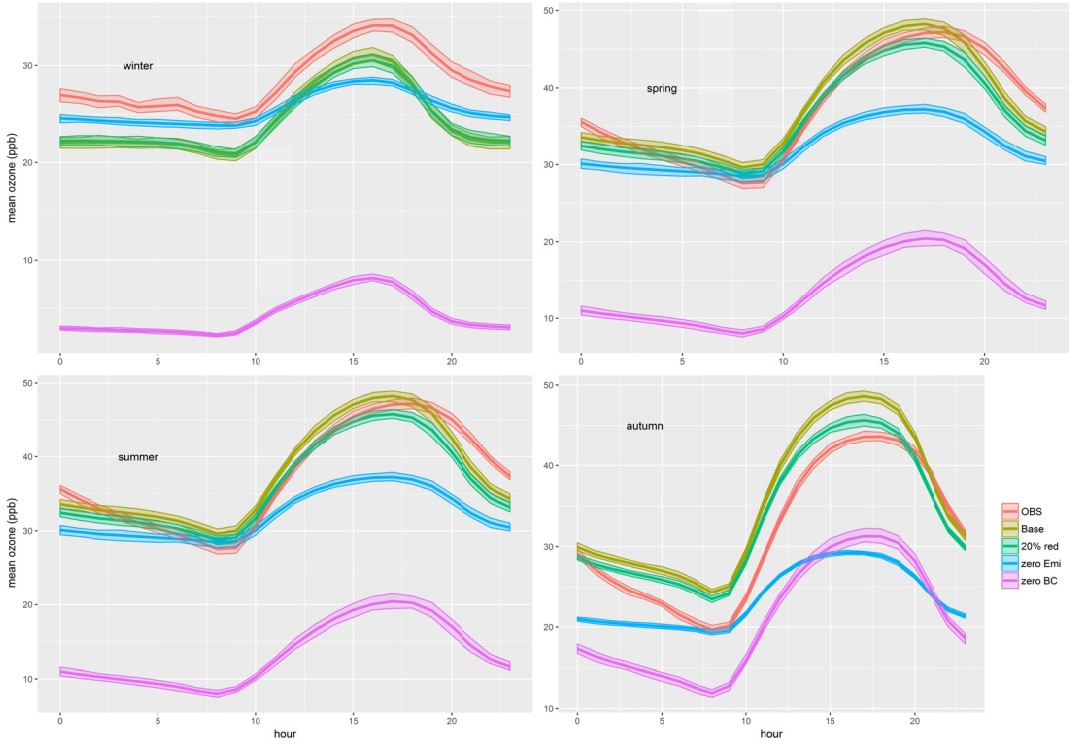

**FIGURE S3** Network average seasonal daily ozone profiles for North America. The confidence bands indicate the 95% confidence interval of
the mean

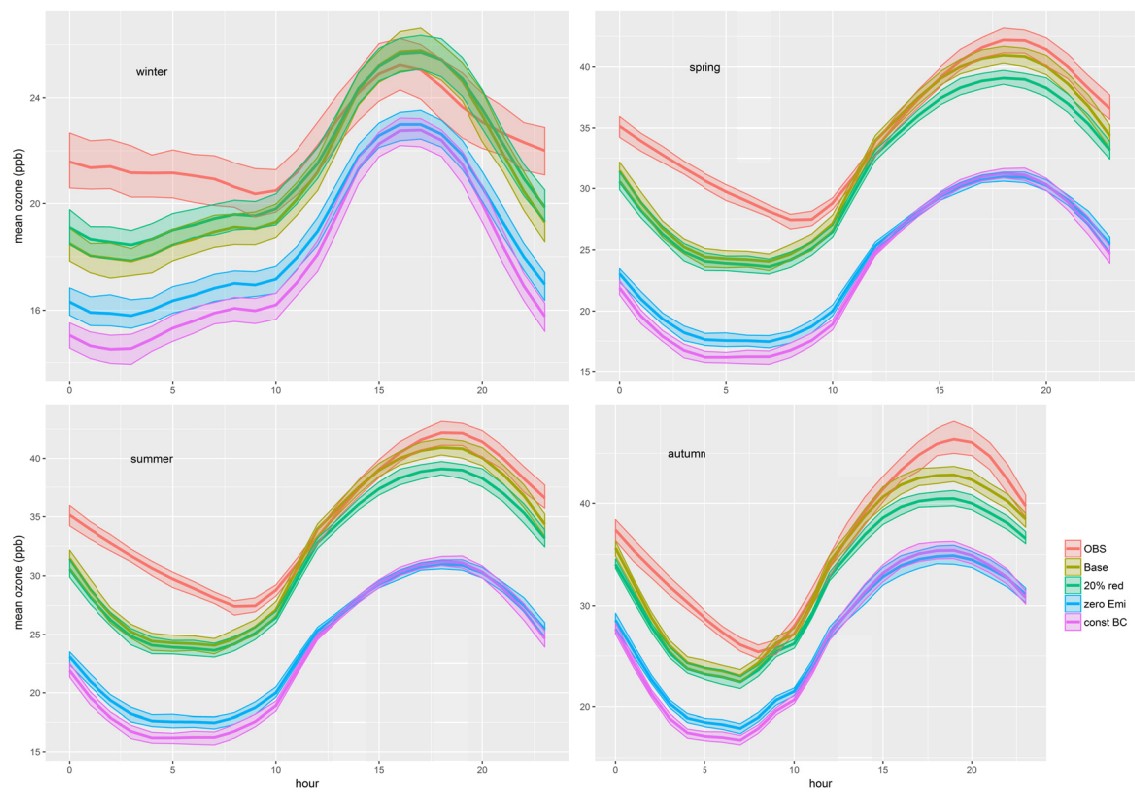

1161

**FIGURE S4.** Network average seasonal daily ozone profiles for Europe. The confidence bands indicate the 95%
confidence interval of the mean

1164

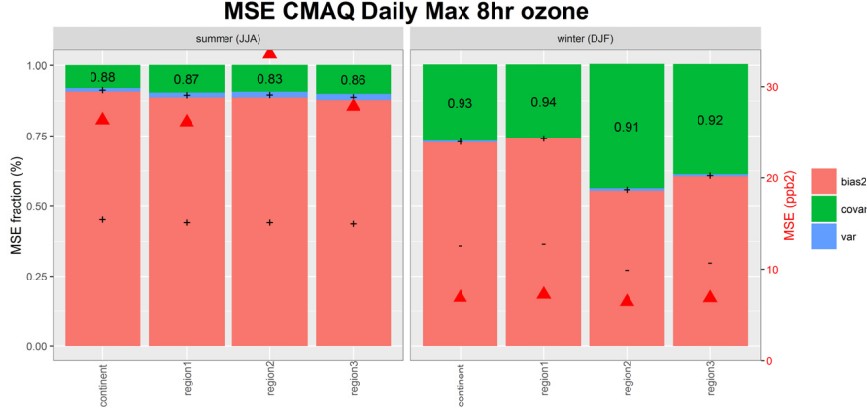

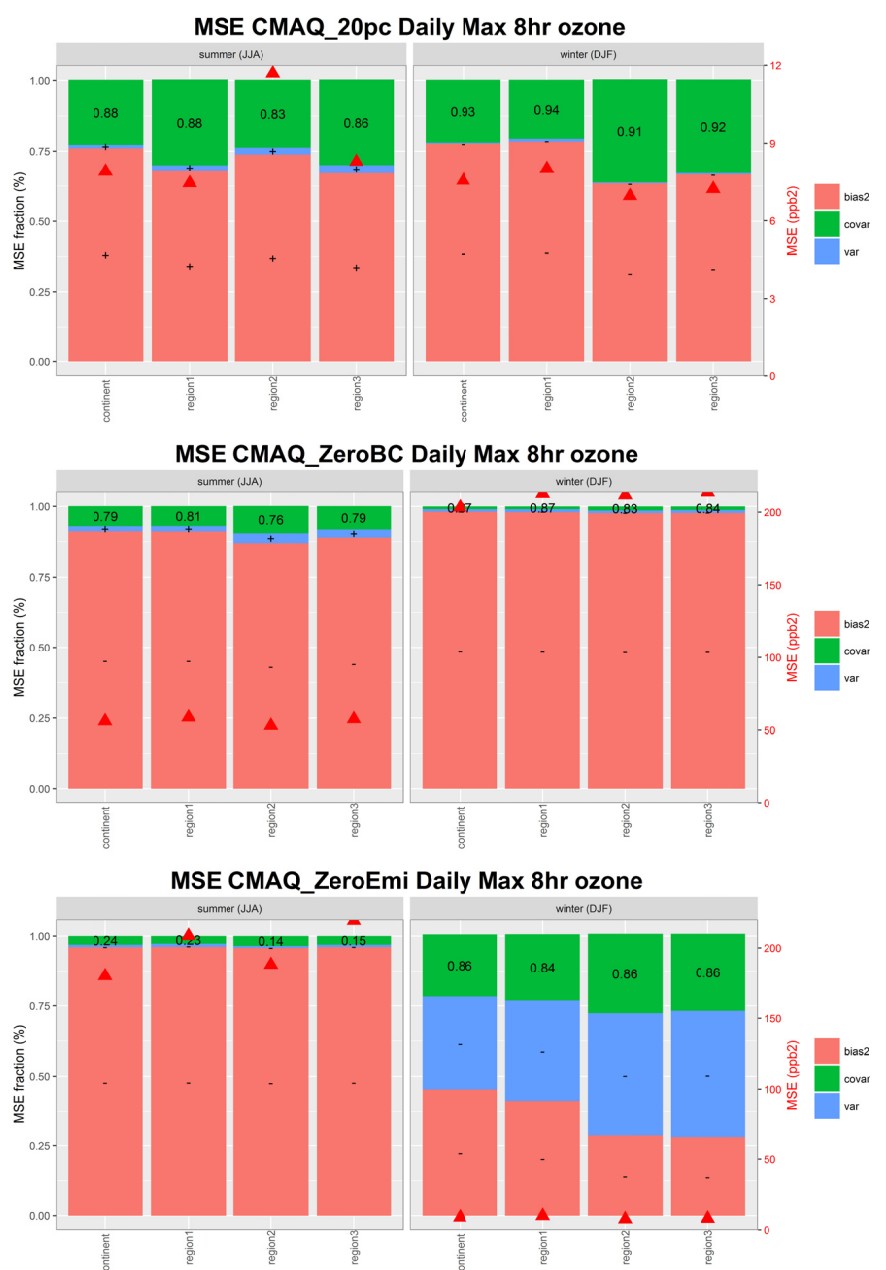

1165    **FIGURE S5.** Same as Figure 6 of the main text for the daily maximum 8-hour rolling average ozone

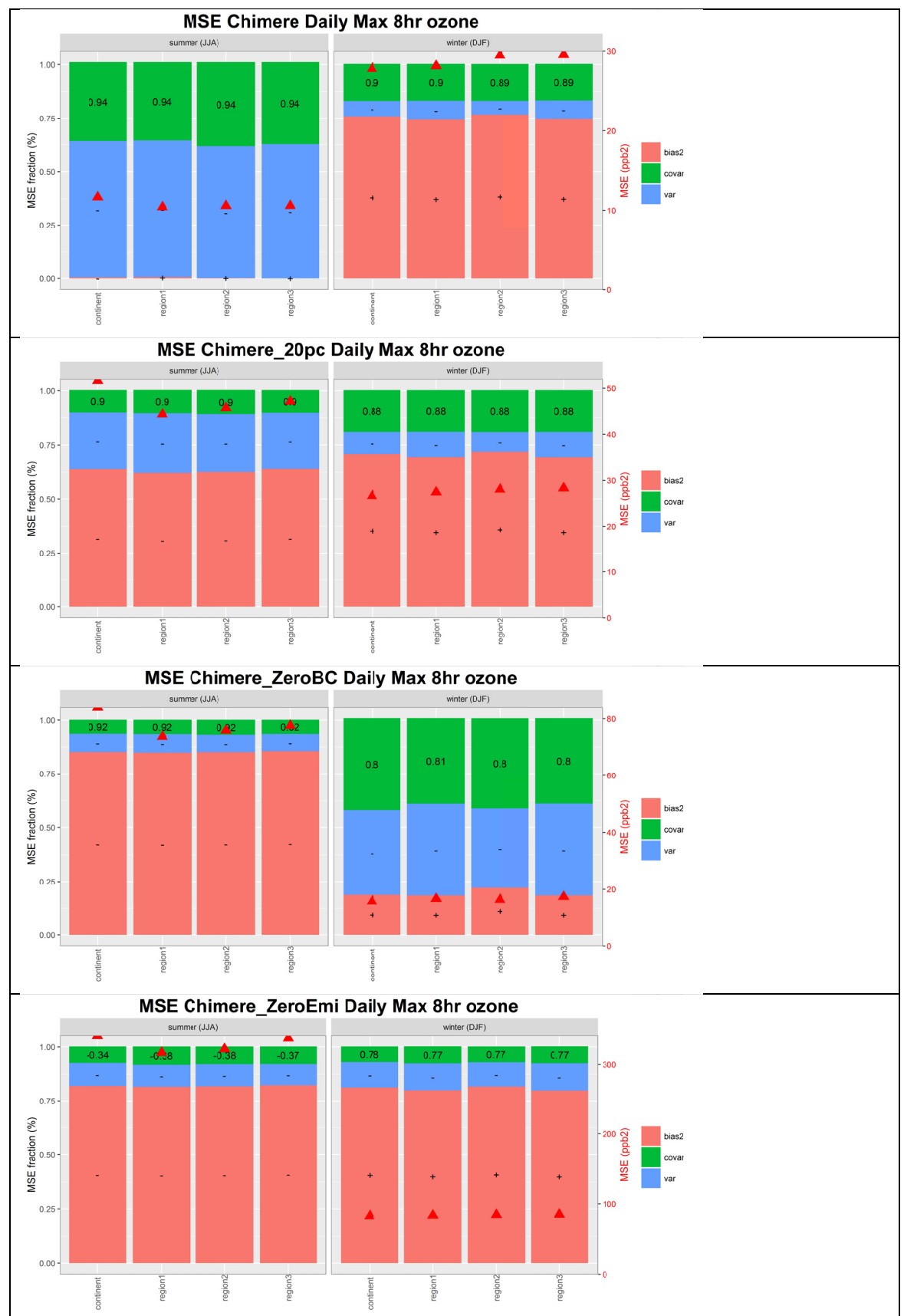

1166 **FIGURE S6.** Same as Figure 7 of the main text for the daily maximum 8-hour rolling average ozone

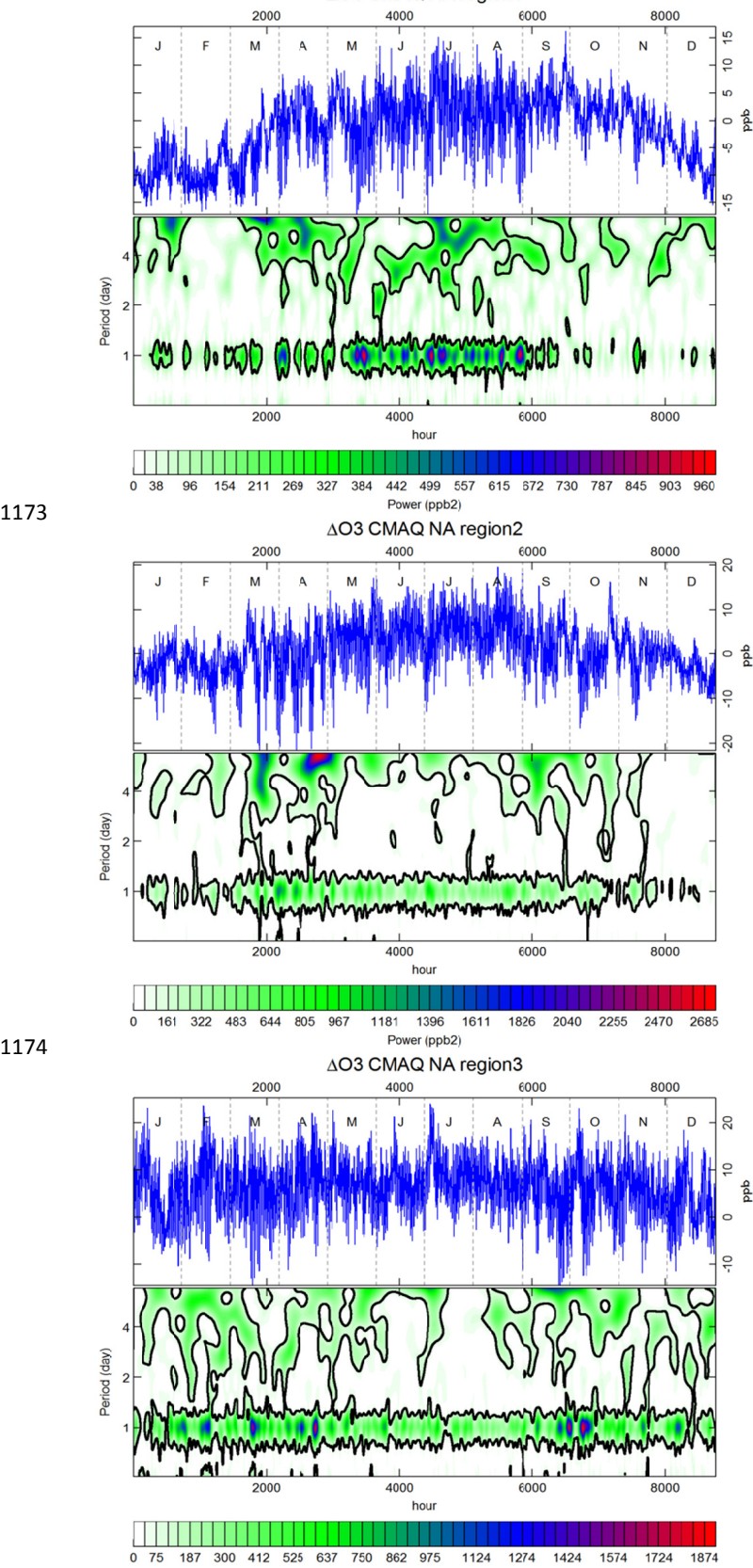




**FIGURE S7**. Annual time series of differences between CMAQ and observed O$_3$ ($\Delta$O$_3$, top panel) and Morlet
wavelet analysis of the periodogram of $\Delta$O$_3$ (lower panel) for the three NA subdomains, for periods between
0.5 and 6 days. Black contours lines identify the 95% confidence interval. The period (in days) is reported in the
vertical axis, while the quantiles of the power spectral density are measured in ppb$^2$.

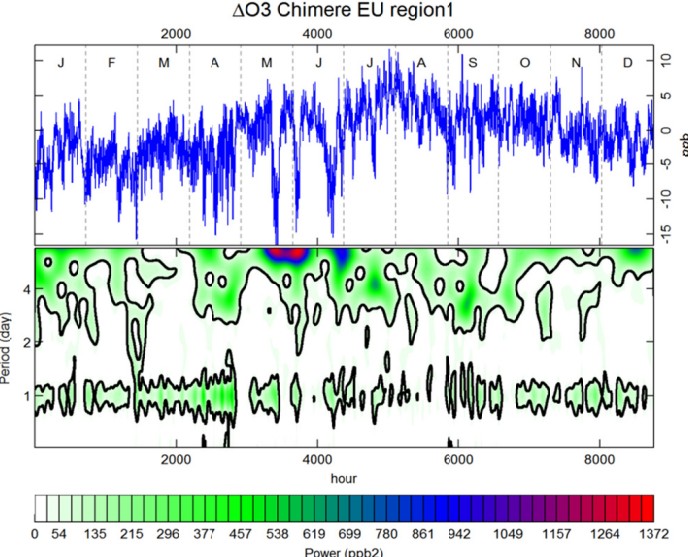

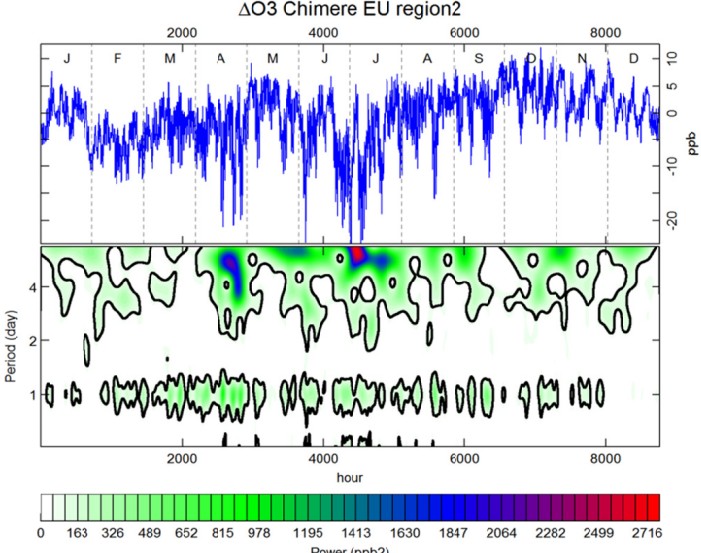

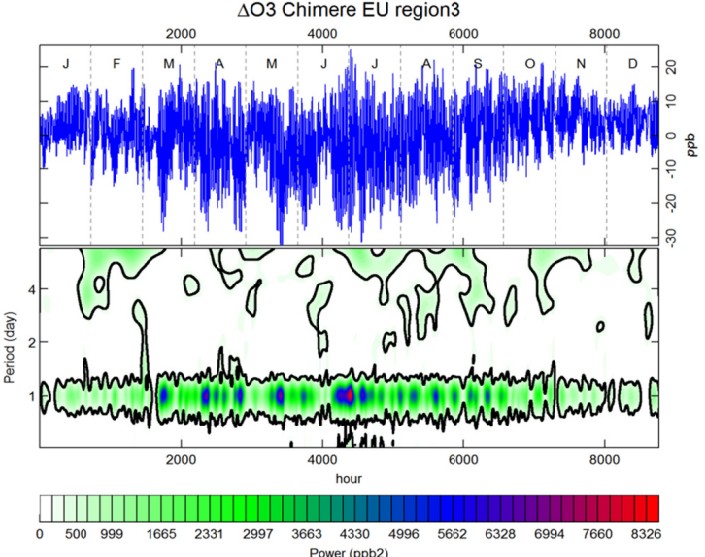

**FIGURE S8**. Annual time series of differences between Chimere and observed O$_3$ (ΔO$_3$, top panel) and Morlet
wavelet analysis of the periodogram of ΔO$_3$ (lower panel) for the three EU subdomains, for periods between
0.5 and 6 days. Black contours lines identify the 95% confidence interval. The period (in days) is reported in the
vertical axis, while the quantiles of the power spectral density are measured in ppb$^2$.

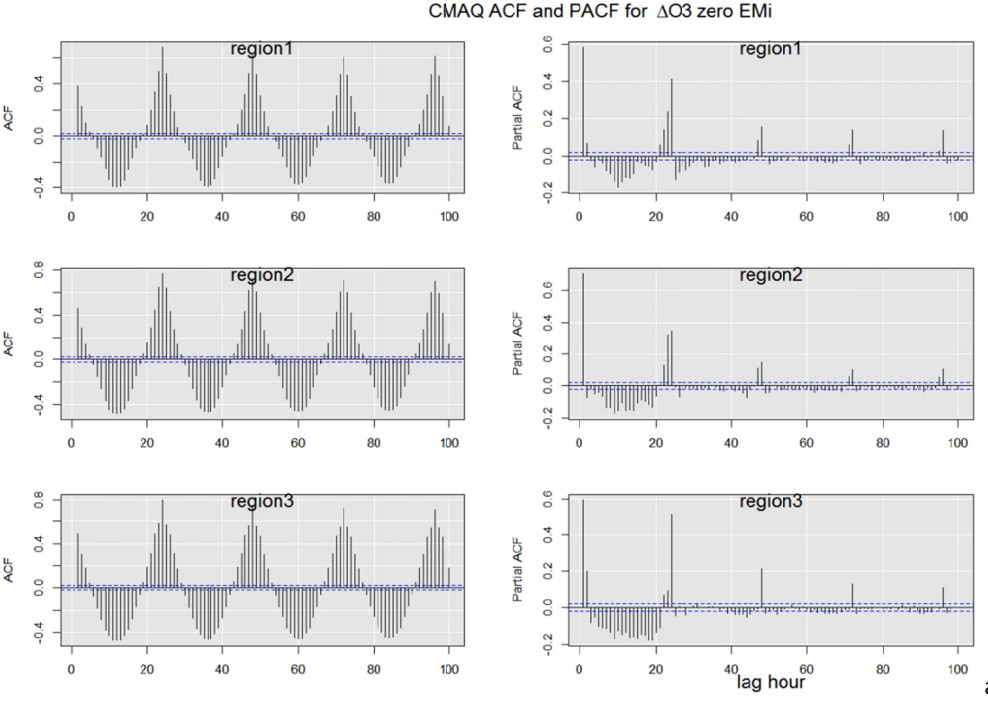


a)

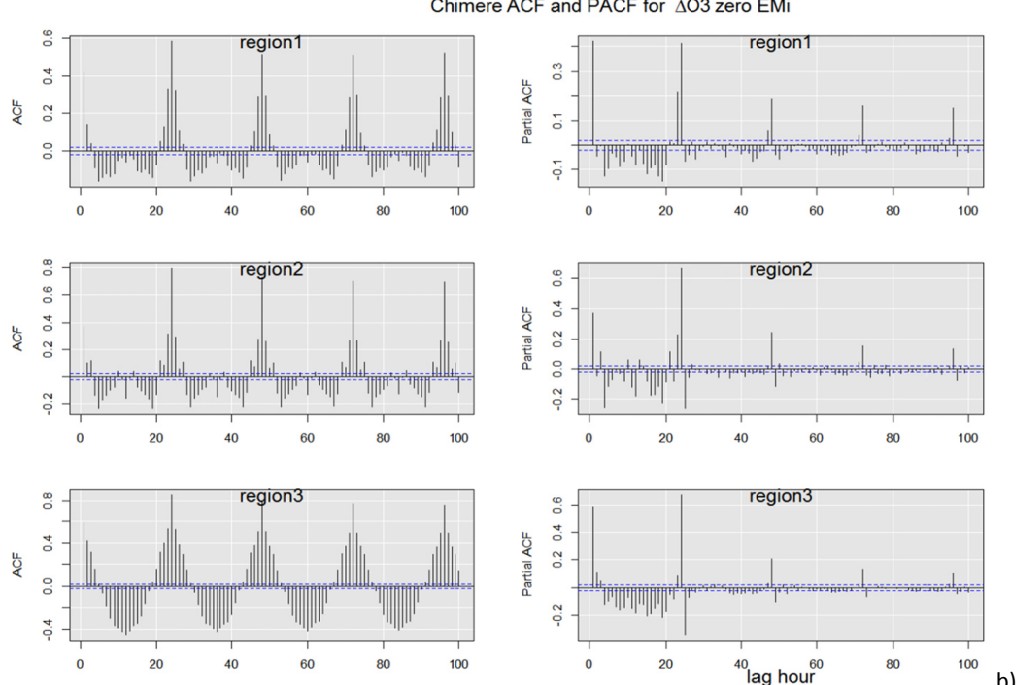


**FIGURE S9.** Autocorrelation (ACF) and partial autocorrelation (PACF) function for the differenced time series of
residuals of ozone (mod-obs) for the 'zero Emi' scenario for a): CMAQ and b): Chimere.

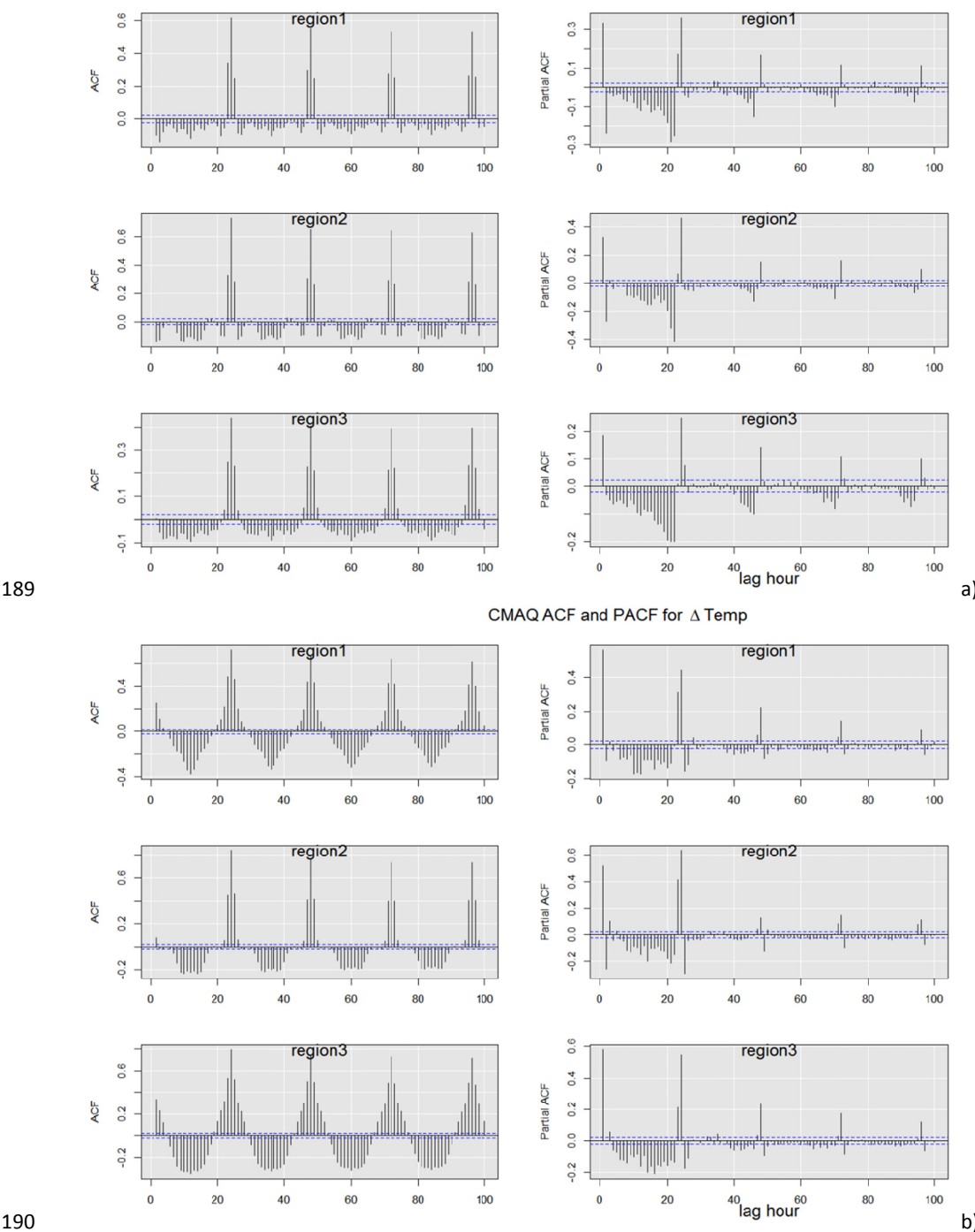



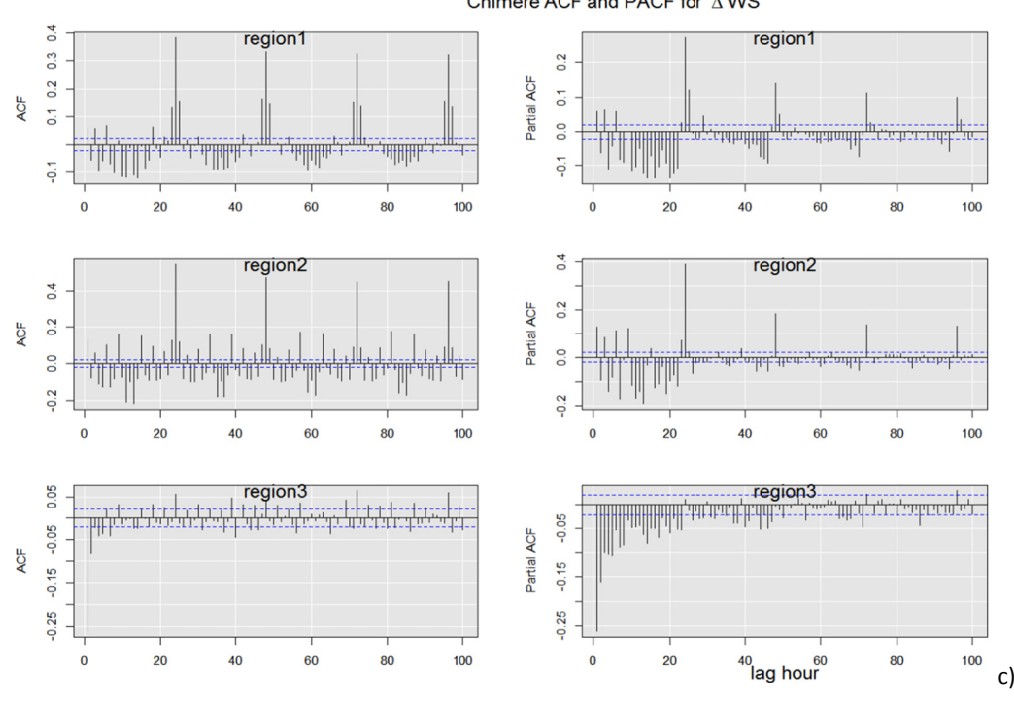

c)

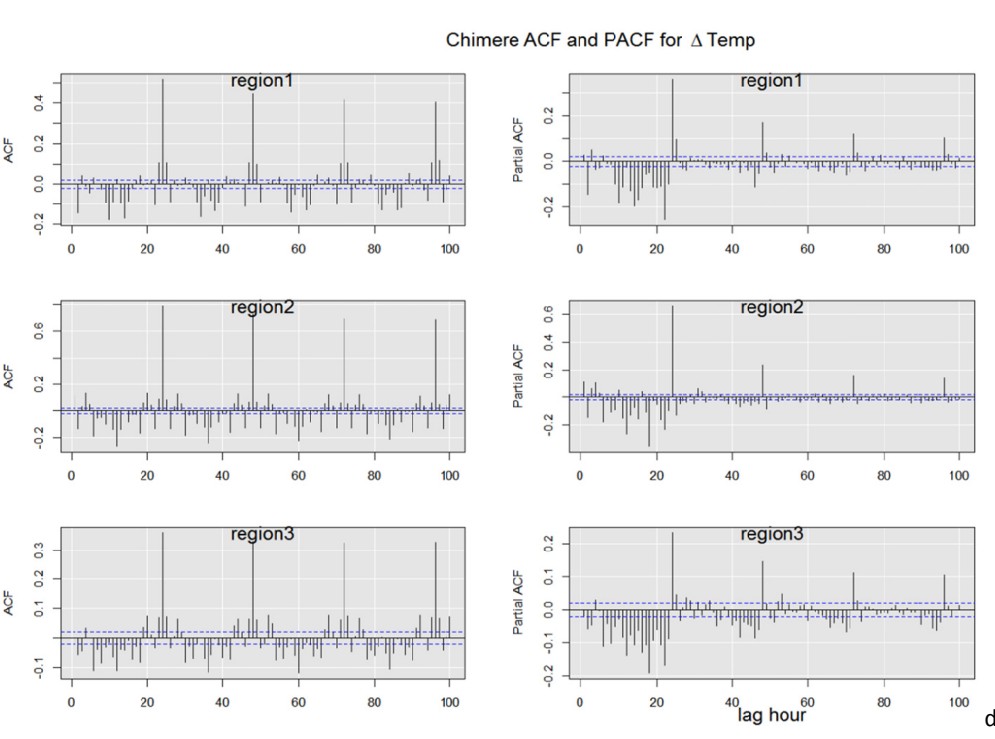

1189                  d)

**FIGURE S10.** Autocorrelation (ACF) and partial autocorrelation (PACF) function for *a)* the differenced time series
of residuals of WS (mod-obs) (a: CMAQ, c: Chimere) and Temp (b: CMAQ, d: Chimere).

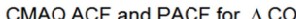

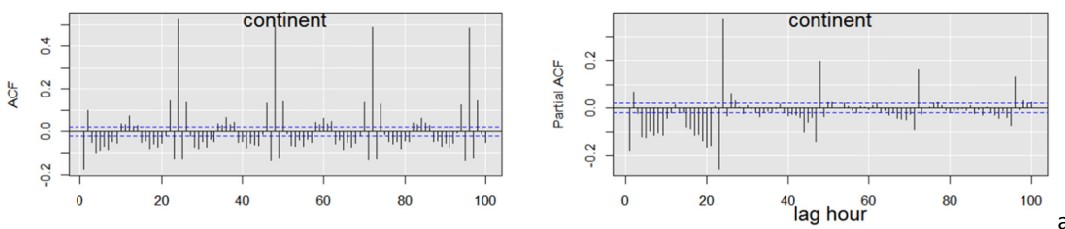

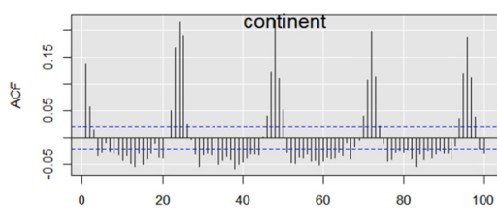

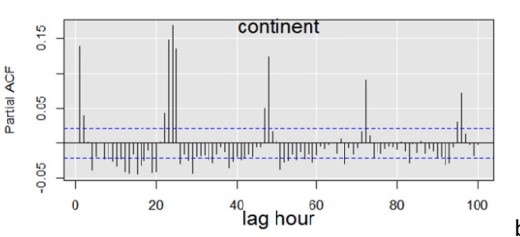

**FIGURE S11** Autocorrelation (ACF) and partial autocorrelation (PACF) function for the differenced time series of
residuals (mod-obs) of a) CO in NA and b) $PM_{10}$ in EU.

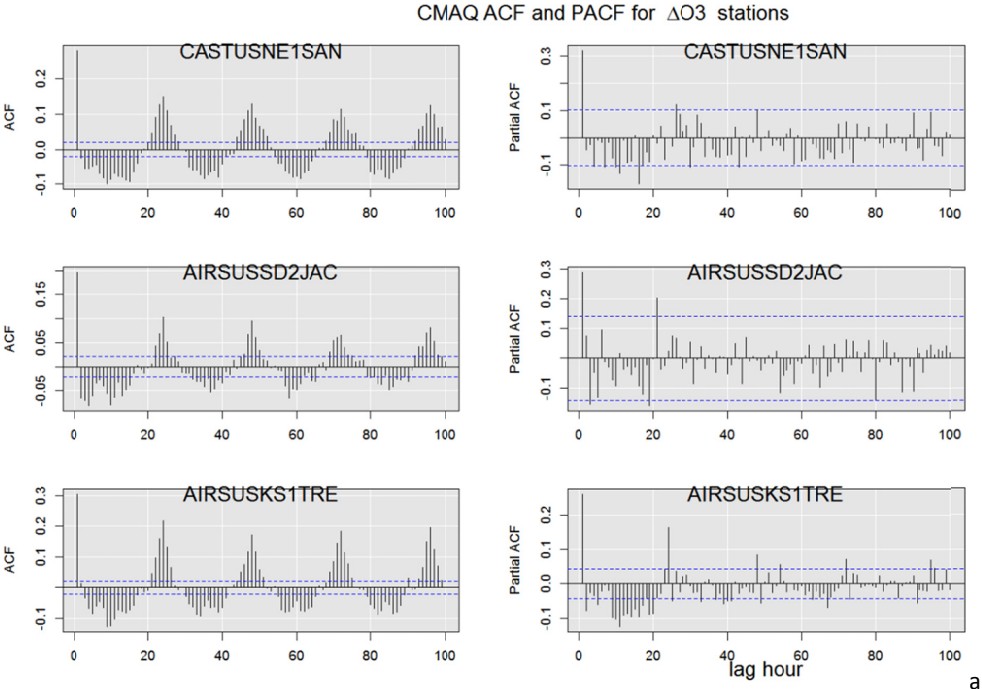

1197

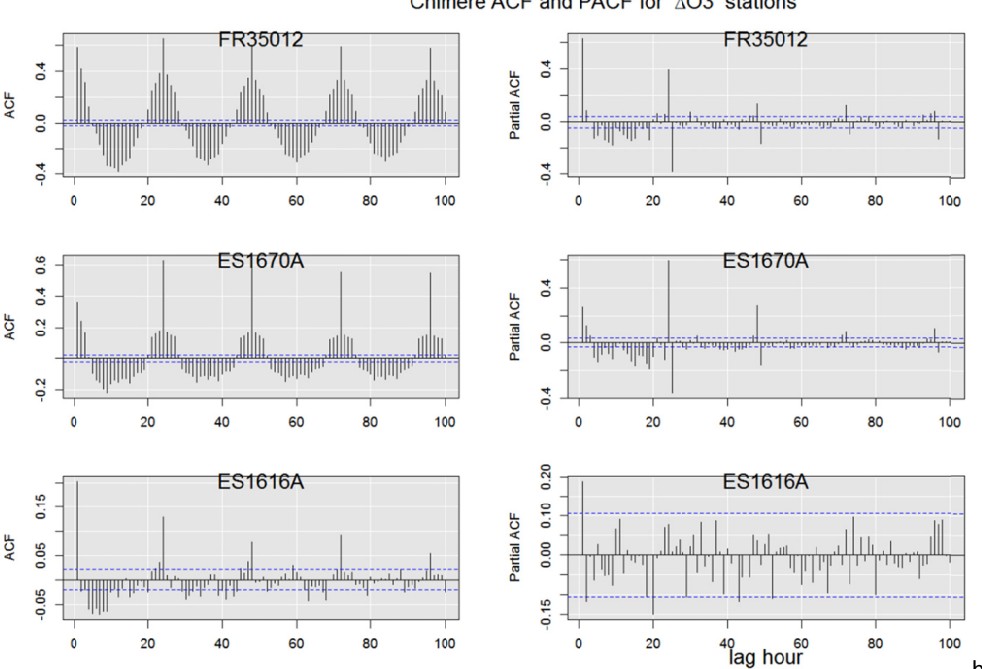

1198

b)

**FIGURE S12.** Autocorrelation (ACF) and partial autocorrelation (PACF) function for the differenced time series of
residuals of ozone (mod-obs) for the 'zero Emi' scenario calculated at three stations where the cumulated
isoprene concentration over the months of June-July-august is minimum (compatibly with the availability of
and completeness of monitoring data). a) CMAQ and b) Chimere

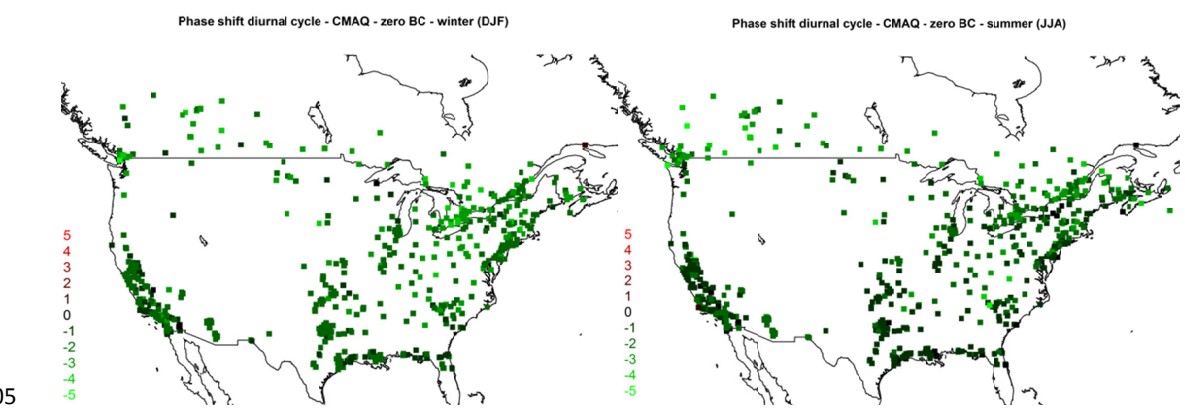

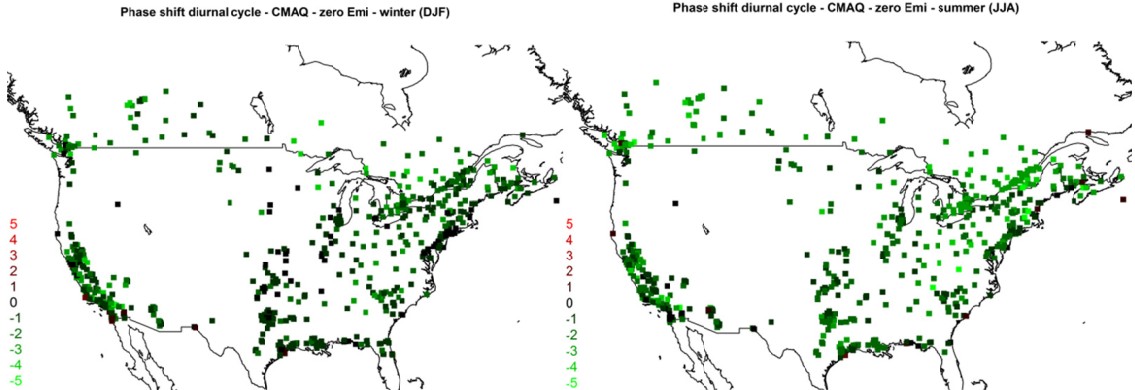

**FIGURE S13.** Phase shift of the diurnal cycle (in hour) for the network of receptors in NA (includes all stations
types). A positive phase shift indicates that the model peak is 'late', vice-versa if the phase shift is negative, the
model peak occurs earlier than the observed peak. Top panels 'Zero BC' case (winter and summer); lower
panels: 'Zero EMI' case (winter and summer)

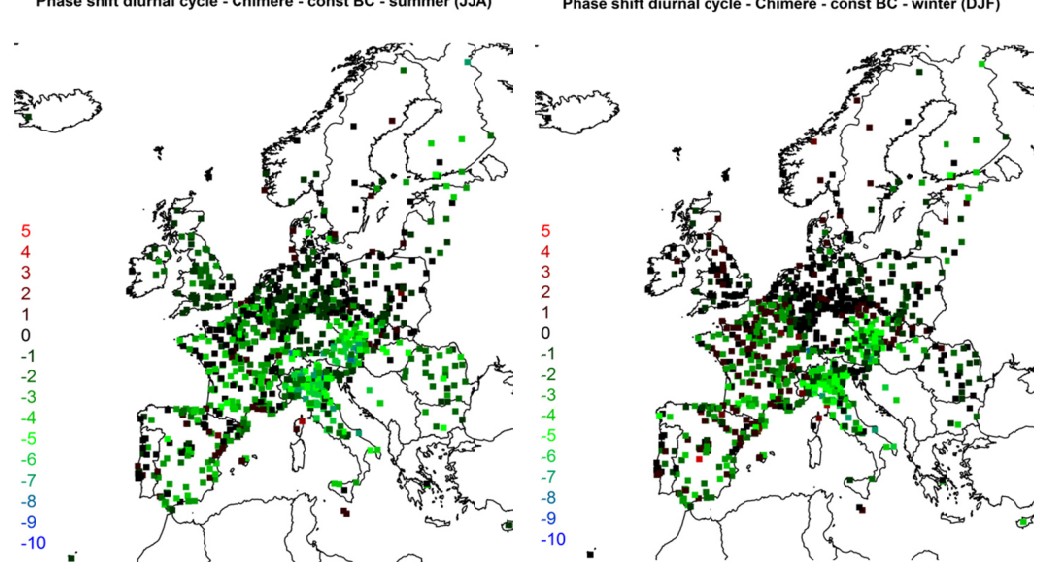

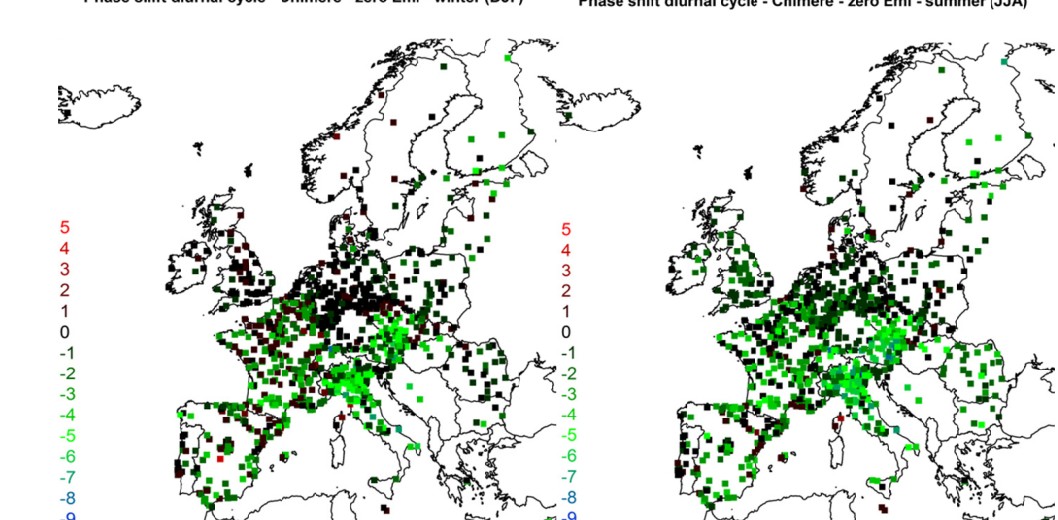

**Figure S14**. As in Figure S13 for Europe

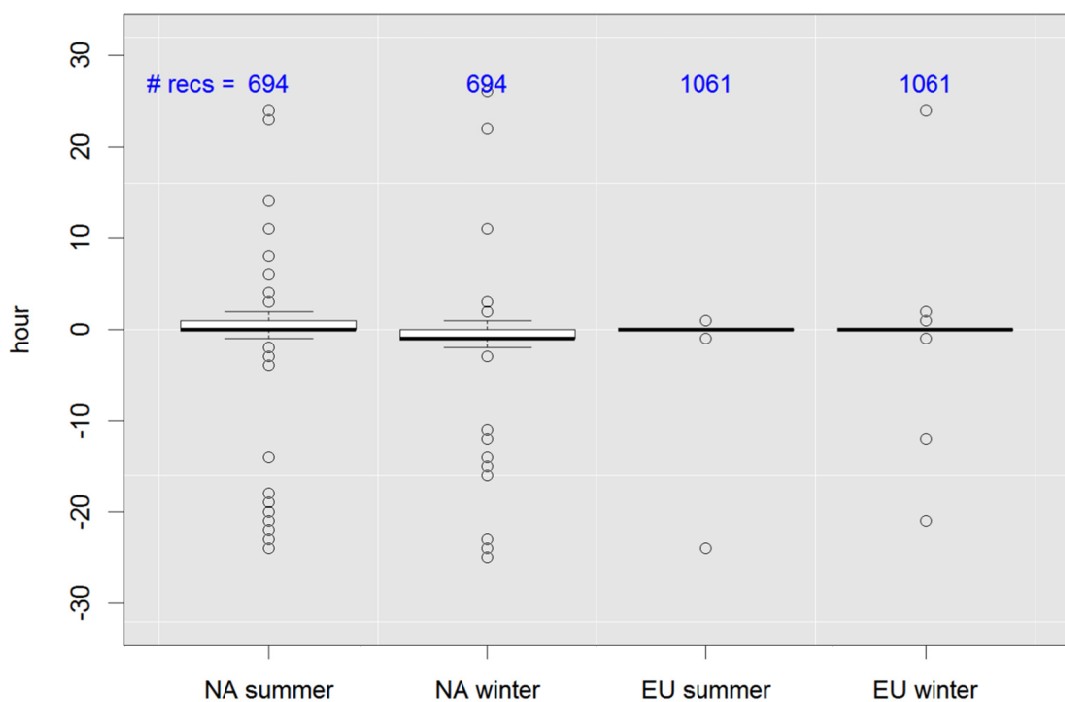

**FIGURE S15.** Percentile distribution of the difference (in hour) at the receptors (the number of receptors is
reported at the top) between the time shift of the base and the 'zero Emi' case

1217

1218

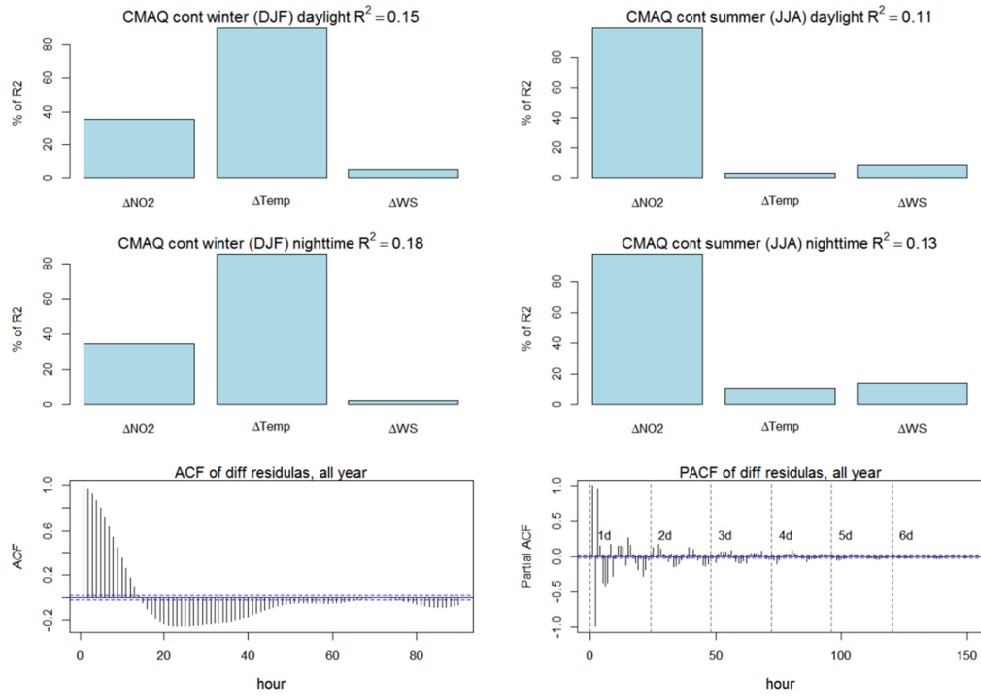

1219

**FIGURE S16.** Percentage of variance explained by the regressors (the total $R^2$ for the regression is reported in the
title of each panel) when the diurnal fluctuations are removed. The relative importance of each variable is
assessed by using a bootstrap resampling. The plots at the bottom show the ACF and PACF of the yearly time
series of residual of the fit, that is of what is not captured by the linear regressions on the available variables. .
The analysis encompasses 47 coolocated stations (the NA stations for ozone, $NO_2$, WS, and Temp that fall in a
radius of 1000 m and vertical displacement less than 250m).

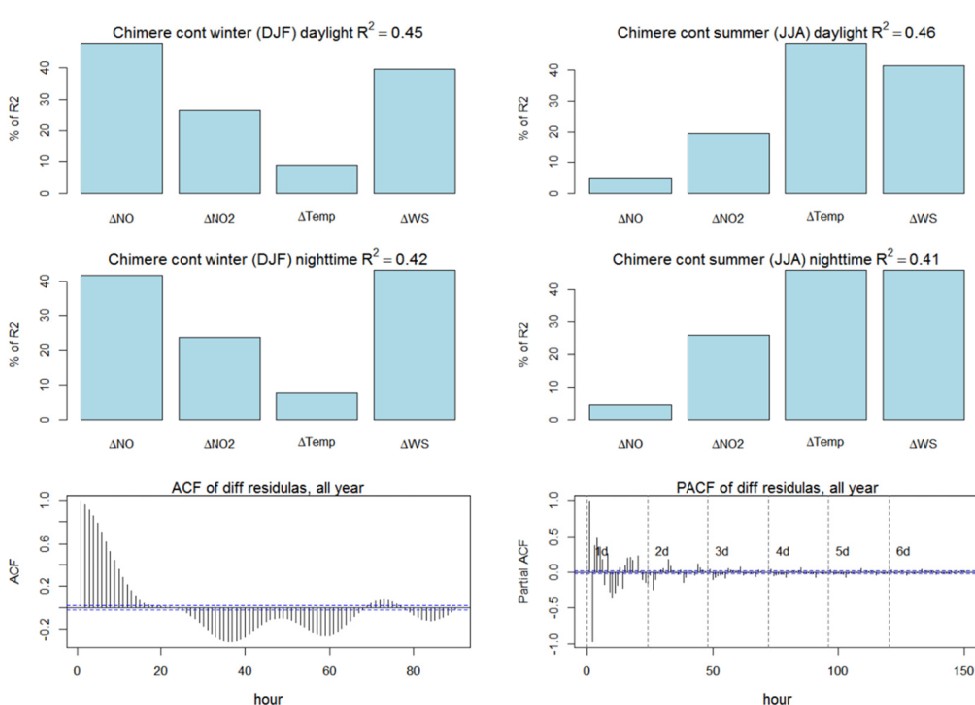

1226

**FIGURE S17.** Same as for **FIGURE S166** EU. The analysis encompasses 61 coolocated stations (the EU stations for
ozone, NO, NO$_2$, WS, and Temp that fall in a radius of 1000 m and vertical displacement less than 250m).

1226

1227