# Peer review of "Advanced error diagnostics of the CMAQ and Chimere modelling"

_Atmospheric Chemistry and Physics, 2017_

## Referee Comment (RC1) · Anonymous Referee #1 · 25 Apr 2017

In this work authors pursue their efforts in devising more sophisticated methods for error analysis in air quality models. In the light of experience gained during the AQMEIIs phases, they analyze results for ozone simulation for North America (CMAQ) and Europe (CHIMERE).

Given that a statistical analysis of model results can only be the first step toward a more-in-depth diagnosis of model deficiencies (it is difficult at this level to disentangle the impact of NOx/VOC chemistry, radiation, boundary layer dynamics and biogenic emissions, dry deposition, etc.), they report some interesting results which would be help in orienting modelers during model development or gaining confidence in using model predictions.

They suggest a combination of spectral (wavelet), time series (ACF, PACF and Kolmogorov-Zurberko filter) statistical tools and simple linear regression analysis to apportion model errors, applied to the decomposition of mean square error into its components (square bias, variance and covariance).

In my opinion, an intelligent approach is to subdivide the AQMEII domains into sub-regions and highlights the differences in model performances for each domain. For example a striking feature is the CHIMERE bust for the Po valley (EU3), associated with the diurnal component, opposed to the better behavior for EU1 and EU2 domains. Why CHIMERE performs better for north Europe? In a certain sense this work opens more questions than answers, but (I think) this is exactly the aim of authors. Though their analysis does not provide a solution to the problems raised during their evaluation, the combination of these statistical tools allows a better understanding of model deficiencies.

My major remark is that authors condense a great mass of information, difficult to assimilate without rereading back and back again. Moreover, they present results for different sensitivity scenarios, e.g. zero BC, const BC, 20% red, ... Given a so large mass of information and different statistical analysis, I suggest, if possible, to re-organize the paper. I prefer a more in depth discussion of what may be the physical reasons for model deficiencies. A suggestion may be to try to highlight the role of physical mechanisms (this has already be done here and there, throughout the paper, but I prefer more emphasis). The logical course could be to start from model components (dry deposition, PBL dynamics, etc.), show what are model deficiencies and how your analysis is able to highlight these deficiencies. In this manner your ideas could are introduced as a "proof of concept" applied to a concrete example.

What is the role of dry deposition? Could the PBL dynamics better analyzed? The mean square error decomposition into its component and spectral components could identify where model need a deeper analysis? A plainer analysis of these aspects, moving secondary results to the supplementary section, would help the reader to track

the pieces of information better.

Overall, I recommend the publication of this work, since it summarize the efforts made during AQMEII3 phase and suggest useful statistical analysis, well beyond the 'standard' statistical metrics, often used to qualify model results.

---

## Referee Comment (RC2) · Anonymous Referee #2 · 25 Apr 2017

General Comment: This study pursues to outline the evaluation methods applied during the three phases of the AQMEII activity in view of building an evaluation strategy. Annual simulations of the WRF-CMAQ and IFS-Chimere are evaluated, as well as runs with perturbed emissions or boundary conditions or ozone deposition. The analysis focuses on ozone simulations over selected European and North American subregions. The validation is based on (a) the decomposition of the mean square error into bias, variance and covariance (to isolate the sources of error into systematic, variability and phasing) and (b) the decomposition of the time series into spectral components (to investigate the temporal characteristics of the error and elicit the associated misrepresented processes). The authors conclude that both decompositions can aid understanding of the causes of model error, when sensitivity simulations or process analysis simulations are not available. The study is addressing an interesting problem and I recommend it should be published. Few comments may be helpful to improve the paper presentation.

Specific Comments: (a) [Eq 1, L174] Please include some explanations for Eq 1. Is it an identity? Are there any assumptions behind it? (b) [ACF & PACF discussion, paragraph 4.2] Please include in the text only arguments arising from significant correlations. In the same manner, significant correlations should be distinguishable in Figures 15 and 16. (c) [kz filter, paragraph 4.3] Please provide an estimate for the error leakage from the use of non-independent spectral components and discuss its impact on the results. (d) [regression, paragraph 4.4] The attempt to explain the ozone error using the error of selected variables through linear regression analysis achieves very low R2. I would suggest to remove any inference built from this linear model (as it fails to explain the variability of ozone error) and remove figures 17 and 18. Alternatively, the authors can augment the statistical model with the errors from more explanatory variables.

Technical Comments: (a) [L175] Replace 'Where' with 'where' (b) [L370-372] Please expand or rephrase (c) Figures: please increase the font size in the majority of figures.

---

## Referee Comment (RC3) · Anonymous Referee #3 · 10 May 2017

Review of "Advanced error diagnostics of the CMAQ and Chimere modelling systems within the AQMEII3 model evaluation framework, but Solazzo et al. Section 4.3:

General comments: My overall rating for this paper is "major revisions" – the paper is valuable from the standpoint of showing how the errors in air pollution models may be linked to particular time scales, but some of the methodology appears to be circular, and they attribute some of the errors to specific processes without sufficient evidence. Further model simulations or rewriting portions of the paper to provide more caveats on their conclusions is necessary. There were issues with regards to formatting of the paper (ordering of the figures not matching the text) and inconsistent terminology, which should have been addressed prior to submission.

[Figure]

My main concerns are given a "**" symbol in the following detailed comments

Abstract: ** The abstract makes the conclusion that the representation of the PBL is pivotal, but I see little direct evidence in the paper that this is the case – rather, that the PBL representation, based on its diurnal variability and a version of CMAQ which was not tested by the authors analysis, may have a role in the errors. That role has not been established in the authors work – which suggests that the errors have time scales similar to the PBL variation. The potential for any other sources of error with a diurnal timescale to have this effect is insufficiently addressed in the main body of the text and the conclusions. I'll return to this point here and there throughout my review.

** Point (i) made on line 27 suggests that much of the total ozone quadratic error occurs in the component with time scales > 1.5 days – this makes the contribution of identifying a time scale which contains much of the error. However, I'm concerned that this conclusion is the result of a circular / potentially bias-inducing process in selecting the stations for comparison. Later in the paper, the regions for comparison have been selected based on hierarchical clustering which the authors mention select for longer time scales, and spatial averaging is also used, which will tend to minimize the shorter time scales. The finding that much of the error lies in the larger time scales may thus be a result of the selection of stations for analysis. If the entire dataset is used, within NA and within EU respectively, will the same finding still hold?

Point (iv): CMAQ ozone error has a negligible dependence on errors in NO2, while NO2 errors are important in Chimere – a sentence on why this might be the case would be useful (e.g. are the NOx emissions in the European domain much more poorly characterized than in the North American emissions data, hence the European values are more influenced by that error? Point (v): did zeroing CMAQ's winter anthropogenic emissions have no impact on the ozone concentrations (this is how I'm interpreting the word "null" on line 34) – that's a surprising result; there should have been some response due to the removal of NOx via the NOx titration path, at the least. Can the authors explain why one domain is much more sensitive to anthropogenic emissions

than the other, or can these differences be ascribed to differences in the two modelling frameworks?

Comments on the main part of the manuscript: **Page 3, lines 109 to 115: the first of the five enumerated results in the abstract is that 70 to 85% of the total ozone quadratic error is may be attributed to fluctuations associated with time scales > 1.5 days in duration. Here, the sections chosen for analysis are described as being those which cluster closely for time scales of > 1.5 days in duration. The regions for analysis have thus been selected based on the level of similarity in that time scale range, so perhaps the result that most of the error lies within that time scale is not surprising – it does however raise the issue of whether the finding is broadly applicable, or is the result of this selection procedure. If all stations are included in the analysis, rather than the subregions indicated in Figure 1, is finding (i) from the abstract still true? Can the authors provide an argument regarding why the finding is more broadly significant beyond a possible selection bias in how points were selected for comparisons?

Line 128: I think the writing might be a bit unclear – it sounds like the model values being subjected to a two-hour average when the obs are a one-hour average? Might be better to use "The model concentrations were assumed to be linear between the instantaneous on-the-hour reporting times; the integration (average) between those times was used to construct hour starting (or ending) values in order to more directly compare to the averaging used in the observations." I suspect that what is meant in the sentence, please confirm and rewrite for clarity.

**Lines 132 to 133: Why were the data spatially averaged? This needs to be discussed: how does this aid in or improve the analysis? This will tend to smooth out local sources of short term variations, which will reinforce the tendency of the resulting data to have more error associated with the longer time scales (point (i) issue noted above) – are there other reasons why this is done? Line 136: "Missing values have not been imputed" – please clarify: if you are spatially averaging data at each hour, does that not mean that missing data are being imputed spatially?

Line 145: shouldn't that be "BEIS"? Lines 142 to 152: somewhere, mention the extent of the model domain in each case (or add that to Figure 1 as an outline, perhaps?).

**Lines 161 to 165: why was the same lateral boundary condition not used for both domains? Please include an explanation. A zero or non-zero value may have a large impact on the model results; the response of either model O3 may not be linear with respect to the magnitude of the boundary ozone value chosen. For example, a model initialized with 35 ppbv O3 would have more OH on the model inflow boundaries than a 0 ppbv boundary condition. Were any tests conducted showing the potential impact of the choice of boundary condition level on the simulations?

Line 173: Ah – I had been wondering what had been meant by "total ozone quadratic error" in the abstract. Please change the abstract to use either "mean square error" or "total ozone quadratic error (mean square error)"; more people are familiar with that term than total ozone quadratic error, and the terms have not been introduced in the abstract.

Section 3.1: The difference between North American and EU responses to zero emissions is very interesting. A few words describing the flow field in the relevant months in each case would be worthwhile (e.g. I'm assuming that the spring maximum in North America is likely due to mixing events over the Rockies; trop-strat folds and the like – are the EU O3 highs for zero emissions similarly a reflection of transport, or are this local formation)?

Figures 2 and 3: modify the captions: "Average monthly (right column of panels) and diurnal (left column of panels) curves constructed…"

**Lines 208 – 225: The authors mention the work of Appel here as a possible explanation for the more rapid decrease in ozone concentrations in NA1 and NA2. The timing seems less off in NA3, EU1 and EU2, but again the model leads the observations in EU3. So, the question: why is this effect more prominent in some areas rather than others? For example, one possibility is that the stomatal conductance and heat capacity values are more incorrect in regions NA1, NA2 and EU3 than elsewhere, if one assumes that the main cause of the problem is related to those two terms. Is there evidence to suggest that this might be the case (e.g. did Appel describe vegetation types where these differences would be the most noticeable, and did those correspond to the same regions with the largest differences)?

Section 3.1: How does this rule out other possible causes of diurnal error in the simulations?

Section 3.2: This section was very interesting – I'm wondering if the authors can suggest possible reasons for their results. For example, the NA3 subregion had little impact of emissions on night-time model performance for total MSE, but daytime performance showed a more significant impact. I'm wondering, for example, if there are sufficient natural emissions of NO in the San Joaquin valley to titrate the available ozone, hence little impact?

Lines 260 – 272: Implications, please? The anthropogenic emissions affect the bias, variance and covariance of ozone formation. This seems to have taken a back seat to PBL as a source of error elsewhere in the paper.

**Line 267-268: The absence of a variance error in the base case tells us that the standard deviation magnitude is the same for both base case and observations – the authors ascribe this to the emissions being of the correct intensity, but the reasoning for this conclusion is unclear – please explain. For example, if two time series are identical aside from a bias offset at every time, they will have the same net standard deviation from the mean, and much of the error in the zero emissions case is in the bias term – which is what Figure 4 seems to show. That to me implies that the temporal / spatial distribution of the emissions, potentially coupled with the variability due to the meteorology, results in the two time series having the same standard deviation – but not that the emissions have the right intensity (I'm interpreting intensity here as magnitude). Maybe I should ask, "What do the authors mean by 'intensity', in this

context, and how specifically does this explain the absence of a variance error?"

Lines 273 to 276: yes, this makes sense; the boundary condition being a constant in this case so only the bias should be affected. However, you'd want to be careful about assuming that the behavior of the error would be the same with a time-varying boundary condition, which would presumably have variance and covariance errors associated with how well that boundary condition actually captures the inflow conditions. Lines 277 to 285: Interesting: the bias is the dominant part of the deposition effect, which is what I would expect. The covariance term contributes at a few orders of magnitude lower – why? Spatial variability of the deposition field due to changes in ground cover, responses to meteorology of the resistance terms, e.g. temporal variation of the stomatal size? One implication is that getting the deposition magnitude "right" is more important than, e.g., the time dependence of surface resistance. Worth mentioning in the abstract/conclusions?

Line 299: Please clarify this a bit, viz: "resemble those of the daylight base case (Figure 6a, left column), but reduced in magnitude during winter. . ."

**Figure 8 and 9 and similar figures: a general comment: These figures would benefit from a common (perhaps logarithmic) colour scale being used, a scale based on something other than an even division from maximum to minimum in each figure alone (e.g. -80, -50, -30, -10, -8, -5, -3, -1, 0, 1, 3, 5, 8, 10, 30, 50, 80, 100, 300), The graphics quality is poor (images are very fuzzy in the version provided to reviewers). It's difficult to cross compare the different panels to get a feeling for the relative magnitude of the changes in each case, due to these combined issues.

** There are two places where the figure ordering doesn't follow the standard convention of numbering the figures and presenting them in the order in which they are discussed in the text, this being the first. This should have been addressed prior to submission. Given that Figures 4, 6 and 8 are described together in the text, followed by Figures 5, 7, and 9 – they should be reordered and numbered in the order in which

they are discussed.

Lines 308 to 310: what is the (potential) physical significance of the differences between the different regions? Much more of the error seems associated with the covariance than with the NA simulation. Assuming that the two meteorological models are approximately the same in accuracy and timing of events, would this imply that the Chimere model has more issues with the timing of emissions than with the magnitudes?

**Line 311: "Removing the anthropogenic emissions had almost no effect on the covariance share of the MSE". This might be suggested by Figure 5, though since the scale is logarithmic the differences may not appear large. But Figure 7 (a) compared to Figure 7(c) implies that the share of the covariance has decreased significantly for the zero emissions case, going from less than 10% for the continent, for the base case mean ozone, to over 75% for the zero emissions mean ozone. I agree that the variance portion is unchanged, though. Please correct or clarify the statement starting on line 311.

Line 327-328: Another reason to wonder why choose 35 ppbv as the boundary condition for one model simulation and 0 for the other. It's unlikely that a model would use 0 as its boundary condition... it would have been better to use a constant value for CMAQ, perhaps chosen based on upwind observations. Why was 35 ppbv chosen for Chimere, and zero for CMAQ?

Line 333-334: Interesting. This implies that the variability of the boundary conditions themselves become more important in winter, as well.

Line 335: RMSE or MSE? Please use consistent terminology throughout the paper. I think this should be MSE in this section. Line 336: "... and bias error during the night-time (Figure 5e)."

Line 341: Clarify with an additional reference to the figure demonstrating the point

being made, viz. "a drastically reduced covariance error compared to the mean ozone (Figure 7a); the timing error is now shifted..."

Figure 6e versus Figure 7e. Why is a much larger portion of the daily maximum 8hr ozone MSE for Chimere in variance, while it's almost non-existent in CMAQ? I suspect that this may represent issues with the timing of the emissions in the European domain. It would be useful for the authors to state the values of sigma-m and sigma-o for both models for 8hr daily max O3. My guess here is that the Chimere value of ïAşo is larger than the European value of sigma-m, which would indicate a larger variability of conditions in the observations than in the model (given that I'm assuming the meteorological variability is likely to be the same, this makes the emissions/AQ model variability the likely culprit). Actually stating the values of sigma-o and sigma-m would help. It would be good to compare Figure 6 e and Figure 7e in this regard, too, in the text, since they are so very different, and to provide some possible explanation for these differences.

Figure 9: all figures look alike to casual inspection, since the colour scales differ in extent. See the comment above regarding Figures 8 and 9 and the figures of this type in the text.

Lines 348 to 352: Would the authors be able to explain why the impact of deposition might be so different between the two models? E.g. perhaps a paragraph in which how deposition is implemented at the code level might explain this. Are both models using deposition as a flux boundary condition on the vertical diffusion equation, for example, or is one electing to remove the mass from the lowest model layer? Are the magnitude of the deposition fluxes larger in Chimere than in CMAQ (e.g. compared the total O3 deposition from the two models)?

**Section 4.1 and Figures 10,11: This is interesting, but I'm not sure of the relevance to one of the main issues with regards to correctly predicting air pollution: accurate prediction of high concentration ozone episodes, which are usually of a few days duration. I think this may need a bit of rewriting to focus on the implications of the analysis

towards these episodes. For example, having more energy in the differences in the longer time scales implies that boundary conditions are important again (line 377); the analysis shows that much of the energy in the differences between model and obs is associated with longer time periods; whole year, 1 to 2 months, etc. However, the analysis elsewhere in the paper suggests that boundary conditions are more important only during the winter. I'm not sure, based on the discussion, what this adds to the overall analysis. A suggestion: can you expand the y-axis to focus on time scales of hourly to synoptic (e.g. 5 days), perhaps as a second column of panels in Figures 10 and 11, modify the colour scale accordingly, and focus the discussion on that part of the power spectrum? It also might help if the x-axes of Figures 10 and 11 were done in months of the year. The key question to be discussed in this section should be the extent to which this analysis can suggest deficiencies in the models on the time scales associated with ozone episodes. E.g. an expansion of lines 386 to 391, with less of a focus on the longer time scales, unless the authors can make an argument that the longer time scales are in some way influencing high ozone days, and add more to the issue of boundary conditions than has already been discussed.

Line 393: How well can a model with the relatively low resolution of Chimere in these simulations (25x18 km) be able to capture the daily variation in upslope/downslope winds in the Alps/Po valley – and have there been any higher resolution simulations in the vicinity of EU3 which have better comparisons to observations than attempted here?

Line 398: yes, fold events might be an issue as well – or their combination with upslope/downslope winds. The ozone high in the springtime in North America has been linked to Troposphere/Stratosphere fold events as well, and the events tend to be better captured at higher resolution (though some low resolution models include parameterizations for the events). Transport of fold events to the surface sometimes requires coupling with smaller scale processes such as convection or upslope/downslope winds; the resolution may thus play a factor in this issue.

**The second place where the figure ordering does not match the order of appearance in the text occurs at this point in the analysis. This is disturbing in a multi-author paper which has seen one level of pre-peer-review at the ACPD stage and presumably peer review in some of the writers' institutions; the figures should be ordered and numbered according to their appearance in the text. Figures 12, 13 and 14 appear after Figures 15 and 16 in the text – please renumber the figures to appear in the sequence they appear in the text, and reorder and re-caption the figures accordingly. Perhaps these were last minute changes to the manuscript?

Section 4.2, 4.3:

**Line 413: "explains" is too generous here. "The majority of the variance is accounted for in the 24 hour lag autocorrelation" might be a better way to phrase this. This sort of analysis identifies the time scales associated with the model error – which is a very useful thing to do – but does not describe the causes of those errors. There are a number of variables listed (lines 414-417) which have been known to affect ozone formation for several decades, many of which are interlinked; e.g. the incoming shortwave radiation may be affected by the cloud cover in turn affected by the relative humidity, etc.. The variables are not independent. The key finding of these sections is that a 24 hour (or multiple of 24 hour) periodicity exists within the ozone errors. A similar periodicity is found for the wind speed and temperature errors – but the authors imply a causal link to the meteorological modules (line 433-434), which I don't think is justified based on their analysis ("correlation /= causation"). Here's two alternative possibilities (neither of which may be the "actual" cause, my point here is that unless there's direct evidence, the linkage suggested is hypothesis rather than the result of analysis):

(1) Suppose the diurnal temporal allocation of the emissions data is incorrect, so that the model emissions peak at the wrong time of day – this would result in a diurnal error variation which has nothing to do with the meteorology. The authors later acknowledge this possibility later in section 4.3, line 483 – but in the Conclusions (lines 605 to 609) focus only on the PBL possibility – where really, it could be either, or something else

altogether. The only solid conclusion is that the error does have that periodicity – which is very valuable information for model developers! However, the evidence for PBL being the main culprit is not clear from the analysis presented here.

(2) Suppose that the actual radiation reaching the surface was attenuated more than in the model, in a fashion where the amount of attenuation depends on the solar zenith angle? This would also result in a diurnal signal to the error (and possibly influence some of the other meteorological processes noted as well). And again have little to do with the PBL.

The authors go on to link the analysis (which only shows the temporal nature of the errors) with the PBL dynamics in both models, and present that link as one of the conclusions of the paper (line 605-609) and in the abstract. This link is not justified by their analysis, which identifies time scales only, not specific processes. The suggestion that the issue observed in the current work has been addressed in a more recent version of CMAQ (line 436) is unfounded – unless the authors make use of that specific version of CMAQ and repeat their analysis, and find that the error is completely removed. The conclusions and the 4.2/4.3/4.4 analysis need the text modified to make it clear that there are, based on the analysis carried out thus far, multiple possibilities for the time offset noted in the errors (e.g. modifying their text to remove phrases like "likely" with regards to the cause of the errors, using instead, "possibly, subject to further investigation"). Alternatively, they should redo the analysis (if not for the whole period, for a shorter test period in order to better make their case) with the new version of CMAQ mentioned, and/or modify Chimere in order to test out the hypothesis that the PBL parameterization.

\*\* The authors mention that the European measurement data themselves may have timing errors, referencing an earlier paper (lines 480 to 481), as a possible explanation for the grouping by country of the model errors which can be seen in Figure 13(a). Explain why timing issues in the observations doesn't invalidate or weaken the analyses in this section. This is a serious issue – the authors have made an effort to ensure that

the model values are specific and representative of specific hours – but if the measurement data time stamps can't be trusted, that adds additional uncertainty to the analysis and its conclusions. For example, if the time series in the observations are all off by 2 hours in some parts of the domain, would that not show up as an "error" signal with a strong diurnal variation? I also wonder about time zone errors in reporting (local time, standard time, daylight savings time, etc.).

Line 480: harmonization misspelled.

Section 4.4:

**Line 518-519: The authors need to explain why the data were not limited to co-located measurements of air pollutants and meteorology, in this section's analysis. The authors are averaging different sets of locations between air pollutants and meteorology – and making the assumption that the resulting averages will both represent the same region to the same degree. Worst case scenario, for example: air pollution measurements in valley bottoms and meteorological measurements on mountain-tops.

Line 525: The authors use the phrase "overwhelming daily memory of the error" – please explain. The term implies that the results of day 1 are in some way affecting day 2 – but the analysis thus far has dealt with periodicity, which does not necessarily imply a causal link between one day and the next.

Line 531-532: At what point would these collinearities invalidate the analysis?

Line 538-540: The authors again attribute the errors to PBL issues, without sufficient cause. All that can safely be concluded by the analysis thus far is that the errors have a significant diurnal component, and that the PBL parameterization is one of many possible contributors to that diurnal component.

Lines 544-546 and lines 554-558: Filtering out the shorter time-scales removes much of the collinearity of the fields examined – which says that those fields are interrelated for the shorter time scales more than the longer time scales. This does not imply that

errors with CMAQ are associated with longer time scales, however (line 555) – the reasoning behind this statement made by the authors is not clear.

Line 566: Anything with a diurnally repeating signal in the error could be the root cause here.

Conclusions: The authors state (line 577) that other methods of analysis do not help target the causes of model error. I'm not completely sure that the work they have presented does a better job – the main contribution (and a very valuable one!) is that the time scales of the errors may be identified with the methodologies the authors present. However, the causes of those errors remain speculation at this point in time, since a causal link between specific processes and the time scale analysis has not been demonstrated. Nevertheless, the analysis methods introduced here are an excellent route by which to reach that endpoint, and worthy of publication in ACP on that basis alone. Coupled with process analysis (their suggestion for further work), it may be possible to tease out the causes of the errors: e.g. output the change in concentration due to each operator at each time step, and analyzing the rates of change of the different components.

---

## Author Comment (AC1) · 30 Jun 2017

Replies to the comments to the manuscript '*Advanced error diagnostics of the CMAQ and Chimere modelling systems within the AQMEII3 model evaluation framework*' by Solazzo et al., 2017(doi:10.5194/acp-2017-257)

Reviewer 1

In this work authors pursue their efforts in devising more sophisticated methods for error analysis in air quality models. In the light of experience gained during the AQMEIIs phases, they analyze results for ozone simulation for North America (CMAQ) and Europe (CHIMERE). Given that a statistical analysis of model results can only be the first step toward a more-in-depth diagnosis of model deficiencies (it is difficult at this level to disentangle the impact of NOx/VOC chemistry, radiation, boundary layer dynamics and biogenic emissions, dry deposition, etc.), they report some interesting results which would be help in orienting modelers during model development or gaining confidence in using model predictions.

They suggest a combination of spectral (wavelet), time series (ACF, PACF and Kolmogorov-Zurberko filter) statistical tools and simple linear regression analysis to apportion model errors, applied to the decomposition of mean square error into its components (square bias, variance and covariance). In my opinion, an intelligent approach is to subdivide the AQMEII domains into sub-regions and highlights the differences in model performances for each domain. For example a striking feature is the CHIMERE bust for the Po valley (EU3), associated with the diurnal component, opposed to the better behaviour for EU1 and EU2 domains. Why CHIMERE performs better for north Europe? In a certain sense this work opens more questions than answers, but (I think) this is exactly the aim of authors. Though their analysis does not provide a solution to the problems raised during their evaluation, the combination of these statistical tools allows a better understanding of model deficiencies.
My major remark is that authors condense a great mass of information, difficult to assimilate without rereading back and back again. Moreover, they present results for different sensitivity scenarios, e.g. zero BC, const BC, 20% red, . . . Given a so large mass of information and different statistical analysis, I suggest, if possible, to re-organize the paper. I prefer a more in depth discussion of what may be the physical reasons for model deficiencies. A suggestion may be to try to highlight the role of physical mechanisms (this has already be done here and there, throughout the paper, but I prefer more emphasis). The logical course could be to start from model components (dry deposition, PBL dynamics, etc.), show what are model deficiencies and how your analysis is able to highlight these deficiencies. In this manner your ideas could are introduced as a "proof of concept" applied to a concrete example.

We thank the reviewer for the supportive comments. We are aware that we condense a large amount of information in the paper. The paper is a follow up of an even more lengthy manuscript, where all of the models participating in AQMEII were analysed for multiple pollutants. Here we focus on two models only and only on ozone. In this respect, the work we present here needs to be considered as a complementary, deeper, analysis of what was already presented in the previous study. Also the focus is different, in that here we wish to introduce methods that can help diagnose the cause of errors. Explaining the causes of model error, on the other hand, was left as an open question from the overview analysis in the previous publication (actually since the first phase of AQMEII!) and it still is. We have managed to frame the time scale of the error and, by excluding other plausible causes, we can conclude that the dynamics of the boundary layer (which in turn depend on the representation of radiation, surface characteristics, surface energy balance, heat exchange processes, development or suppression of convection, shear generated turbulence, and entrainment and detrainment processes at the boundary layer top for heat and any other scalar) are, at least partially, responsible of the daily model error. Since '*the main aim of this study is to move towards tools devised to enable diagnostic interpretation*' as clearly stated in the introduction, we prefer to base the discussion on the methods used to analyse the processes/variables rather than on the processes/variables.
We have added a new section (section 5) where all the relevant information is summarised in support of the goals of our analysis. In section 5 we describe how the methods we have proposed, in conjunction with the sensitivity runs, can help isolate the causes of model errors.

What is the role of dry deposition? Could the PBL dynamics better analyzed? The mean square error decomposition into its component and spectral components could identify where model need a deeper analysis? A plainer analysis of these aspects, moving secondary results to the supplementary section, would help the reader to track the pieces of information better.

One of the main criticisms raised by reviewer #3 is that we do not provide enough evidence to support our claim that the PBL dynamics is responsible for (part of) the daily error. We have now deepened the discussion about PBL errors throughout the text and summarised the main findings in the new section 5. In particular, since we are not in a position to estimate *directly* the effect of the PBL error(s) on the ozone error, we try to estimate it indirectly, by excluding the role of emissions and chemistry. We have strengthened this point in the revised manuscript by including the following additional results and analyses:

- The error structure (ACF and PACF) of surface wind speed and temperature (also with diurnal fluctuations removed);

- The error structure of the ozone (ACF and PACF) for the case with zeroed anthropogenic emissions;

- The error structure of the ozone (ACF and PACF) for the case with zeroed anthropogenic emissions repeated at receptors sited in areas with negligible isoprene emissions. These stations have been selected by using the map of isoprene cumulated emissions over the months of June, July, and August as provided by the two models analysed here. We have selected three stations per continent;

- The error structure for primary species (ACF and PACF), such as $PM_{10}$ for Europe and CO for North America.

The results of these analyses (discussed in the revised manuscript) reveal a daily periodicity of the error, thus clearly pointing toward the role of PBL dynamics (in a broad sense, including e.g. radiation) in contributing to the generation of the daily error. The absence of a spatial or emission dependence and the persistence of the daily periodicity should give sufficient backing to our hypothesis.

We also note that the timing of the error of the base case discussed in section 4.3 (time shift of the diurnal component) is very similar to timing of the error of the zero emission case (reported in the Supplementary material, figure S13, and S14) regardless of the space location where this is investigated. A new figure has been added to clarify this point (Figure S15).

We are unable to isolate individual processes at this stage, given the observational and modelling data at hands. The high non-linearity of the model's components heavily complicates the interpretation of the results. The MSE decomposition has allowed us, for instance, to isolate the error due to bias from the error due to timing. We were then able to further analyse the time dependency and quantify it by cross-covariance analysis of the diurnal component, up to the conclusion that the timing accounts for an error of about the magnitude of the observed ozone variance. This example shows that the methods we have devised are complementary and that their combined use can contribute to novel insights. Again, then, our preference is to focus on how the methods can help understand the error in the processes.

Overall, I recommend the publication of this work, since it summarize the efforts made during AQMEII3 phase and suggest useful statistical analysis, well beyond the 'standard' statistical metrics, often used to qualify model results.

---

## Author Comment (AC2) · 30 Jun 2017

Replies to the comments to the manuscript '*Advanced error diagnostics of the CMAQ and Chimere modelling systems within the AQMEII3 model evaluation framework*' by Solazzo et al., 2017(doi:10.5194/acp-2017-257)

Reviewer 2

General Comment: This study pursues to outline the evaluation methods applied during the three phases of the AQMEII activity in view of building an evaluation strategy. Annual simulations of the WRF-CMAQ and IFS-Chimere are evaluated, as well as runs with perturbed emissions or boundary conditions or ozone deposition. The analysis focuses on ozone simulations over selected European and North American subregions. The validation is based on (a) the decomposition of the mean square error into bias, variance and covariance (to isolate the sources of error into systematic, variability and phasing) and (b) the decomposition of the time series into spectral components (to investigate the temporal characteristics of the error and elicit the associated misrepresented processes). The authors conclude that both decompositions can aid understanding of the causes of model error, when sensitivity simulations or process analysis simulations are not available. The study is addressing an interesting problem and I recommend it should be published.
We thank the reviewer for the supportive comments.

Specific Comments

- [Eq 1, L174] Please include some explanations for Eq 1. Is it an identity? Are there any assumptions behind it?
  The derivation of Eq.1 is standard and can be found in textbooks as well scientific publications. In the context of AQMEII, it has been extensively discussed in previous publications. It is an identity, i.e. is not an approximation. For self-consistency, a short description of the terms has been provided in the revised manuscript.

- [ACF & PACF discussion, paragraph 4.2] Please include in the text only arguments arising from significant correlations. In the same manner, significant correlations should be distinguishable in Figures 15 and 16.
  Statistically speaking, the auto and partial correlations reported in Figures 15 and 16 are significant when lying outside the horizontal blues lines centred at zero and having width proportional to the 95% significance level (i.e. threshold for the null hypothesis to be rejected of 0.05): $r_{0.95}=0 \pm 2N^{0.5}$, where N is the sample size. Therefore, the values of the auto-correlation are significant (although small sometimes) throughout the range of lags shown in Figure 15 and 16, while the partial correlation are small and non-statistically significant for lags larger than 60 hours (when the diurnal fluctuations are removed). We could therefore reduce the x-axis of the PACF plots of the 'non-diurnal fluctuations' plots, but for consistency of the discussion and presentation of the results we would rather keep the same format.

- [kz filter, paragraph 4.3] Please provide an estimate for the error leakage from the use of non-independent spectral components and discuss its impact on the results.
  The effect of the leakage among components (and its quantification) for the models participating in AQMEII is extensively discussed in Solazzo and Galmarini (2016), Hogrefe et al. (2013), Galmarini et al. (2013), Kioutsioukis and Galmarini (2016) and Solazzo et al., (2017), but it is a sensitive issue and we have added a discussion in the revised text. In this study we use the kz filtering only for the purpose of isolating the portion of the ozone time series that is faster than ~1.5 days. The effect of leakage is rather limited for this particular application (5 to 10%, Figure 1 of Solazzo and Galmarini 2016).

- [regression, paragraph 4.4] The attempt to explain the ozone error using the error of selected variables through linear regression analysis achieves very low R2. I would suggest to remove any inference built from this linear model (as it fails to explain the variability of ozone error) and remove figures 17 and 18. Alternatively, the authors can augment the statistical model with the errors from more explanatory variables.
  The main idea here is to introduce a diagnostic technique that combines the usual regression with confidence estimates and to determine the relative importance of the regressors. We believe that not only the dependence between the ozone error and that of the regressors is informative, but also the lack

of it can be instructive in some instances, especially when the variable of interest (the error of ozone) depends (and/or is strictly related) on a number of other fields which, in turn, depend on the location and time/season. For example, this analysis has allowed us to frame the error of ozone due the error of the precursors (NO, $NO_2$), and thus to reinforce our point about the daily dynamics being one of the major source of error. The suggestion of adding new regressors is indeed very valid and will be exploited in future, but we have already used all the fields for which observational data were made available in AQMEII. $R^2$ is an estimate of the explained variability (ratio of the variance explained by regressors and total variance); its square root, R, gives a sense of the overall associativity (linear dependence). Therefore, $R^2$ = 0.3-0.45 (as in most of the cases of Figures 17 and 18) corresponds to a correlation coefficient of approximately 0.5-0.7, which might not be considered as 'small' in an absolute sense.

The revised figures 17 and 18 now present the continent-wide network average (to account for co-located stations, as requested by reviewer #3), thus only one panel per continent.

Technical Comments:

- [L175] Replace 'Where' with 'where' Done
- [L370-372] Please expand or rephrase Done
- Figures: please increase the font size in the majority of figures. The majority of the figures have been revised to improve readability

---

## Author Comment (AC3) · 30 Jun 2017

Replies to the comments to the manuscript '*Advanced error diagnostics of the CMAQ and Chimere modelling systems within the AQMEII3 model evaluation framework*' by Solazzo et al., 2017(doi:10.5194/acp-2017-257)

Reviewer 3

General comments

1. Abstract: The abstract makes the conclusion that the representation of the PBL is pivotal, but I see little direct evidence in the paper that this is the case – rather, that the PBL representation, based on its diurnal variability and a version of CMAQ which was not tested by the authors analysis, may have a role in the errors. That role has not been established in the authors work – which suggests that the errors have time scales similar to the PBL variation. The potential for any other sources of error with a diurnal timescale to have this effect is insufficiently addressed in the main body of the text and the conclusions. I'll return to this point here and there throughout my review.

We find that the error of the modelled ozone, for both models, has a clear, dominant daily periodicity. The main processes having marked daily periodicity are the timing of the emissions, the chemical transformation and the PBL dynamics driven by the radiation. Perhaps we should have been clearer in stating that by 'PBL error' we meant all the chain of dynamical processes that affect the PBL variability (because we cannot refine the analysis any further in disentangling the source of error), in contrast to emissions. Since we are not in a position to estimate *directly* the effect of the PBL error(s) on the ozone error, we try to estimate it indirectly, by excluding the role of emissions and chemistry. We have strengthened this point in the revised manuscript by including the following additional results and analyses:
- The error structure (ACF and PACF) of surface wind speed and temperature (also with diurnal fluctuations removed);
- The error structure of the ozone (ACF and PACF) for the case with zeroed anthropogenic emissions;
- The error structure of the ozone (ACF and PACF) for the case with zeroed anthropogenic emissions repeated at receptors sited in areas with negligible isoprene emissions. These stations have been selected by using the map of isoprene cumulated emissions over the months of June, July, and August as provided by the two models analysed here. We have selected three stations per continent;
- The error structure for primary species (ACF and PACF), such as $PM_{10}$ for Europe and CO for North America.
The results of these analyses (discussed in the revised manuscript) reveal a daily periodicity of the error, thus clearly pointing toward the role of PBL dynamics (in a broad sense, including e.g. radiation) in contributing to the generation of the daily error. The absence of a spatial or emission dependence and the persistence of the daily periodicity should give sufficient backing to our hypothesis.

We further note that the timing error of the base case discussed in section 4.3 (time shift of the diurnal component) is very similar to timing error of the zero emission case (reported in the Supplementary material, figure S13, S14). In the reply to comment #41 further considerations and novel results are proposed in support of our point.

It is true, however, that we cannot quantify the part of daily error attributable to emissions and chemistry, but at least we can confidently state that the representation surface processes and radiation (that in turn also influence PBL dynamics) is responsible for part of it. The timing of the error of the 'zero-Emi' scenario being very similar to the error of the base case (although with some – little – exceptions during winter in NA) is a further indication that the daily error is more affected by PBL dynamics than emissions timing.
Other, more subtle and untested problems might derive from the land use masking. Any inaccuracy in the land use model directly affects the radiation balance, the surface turbulence, the biogenic emissions. In the current analysis these latter errors are not directly investigated, although the error analysis carried out for three sub-regions and the consistency of the results for the daily periodicity of the error seem to suggest that they are 'small' and/or compensating.

2. Point (i) made on line 27 suggests that much of the total ozone quadratic error occurs in the component with time scales > 1.5 days – this makes the contribution of identifying a time scale which contains much of the error. However, I'm concerned that this conclusion is the result of a circular / potentially bias-inducing process in selecting the stations for comparison. Later in the paper, the regions for comparison have been selected based on hierarchical clustering which the authors mention select for longer time scales, and spatial averaging

is also used, which will tend to minimize the shorter time scales. The finding that much of the error lies in the larger time scales may thus be a result of the selection of stations for analysis. If the entire dataset is used, within NA and within EU respectively, will the same finding still hold?

We have complemented Table 1 with the MSE for the network-averaged ozone, over the entire continental areas. The fractions of MSE due to the diurnal component are in line with those of the sub-regions, indicating that the results were not due to a station selection bias. The issue of spatial averaging is further addressed in response to comment #7, Furthermore, it is important to point out that as noted in Solazzo et al. (2017), the entire bias is apportioned to the base-line (LT) component in the spectral decomposition error apportionment approach. In other words, the ID, DU and SY components contain only variance and co-variance error while the LT component contains those errors as well as the bias error. In that sense, it is not surprising for the apportionment method to indicate that much of the error originates from the longer time scales. This finding is not an artefact of the station selection method but rather indicative of the presence of non-negligible biases in the model simulations which are considered as stemming from systematic processes by the error apportionment methodology.

3. Point (iv): CMAQ ozone error has a negligible dependence on errors in NO2, while NO2 errors are important in Chimere – a sentence on why this might be the case would be useful (e.g. are the NOx emissions in the European domain much more poorly characterized than in the North American emissions data, hence the European values are more influenced by that error?

In line with comment #44 we have now restricted the analysis to co-located stations of ozone, temperature, wind speed and NOx. New figures have been produced (one per continent). The behaviour is similar to the one discussed in the first submission as far as the error of ozone dependence on the error of $NO_2$ is concerned. We do not have an explanation for the higher $NO_2$ dependence of the European error on $NO_2$ and we prefer not to infer on possible reasons without supporting evidence.

4. Point (v): did zeroing CMAQ's winter anthropogenic emissions have no impact on the ozone concentrations (this is how I'm interpreting the word "null" on line 34) – that's a surprising result; there should have been some response due to the removal of NOx via the NOx titration path, at the least. Can the authors explain why one domain is much more sensitive to anthropogenic emissions than the other, or can these differences be ascribed to differences in the two modelling frameworks?

Null, on average, over winter. This result refers to the continent wide monitoring network-average, while the MSE variation is much larger over the individual sub-regions (Fig 6 and Fig 7). We agree that the sentence is confusing and removed in the revised manuscript.

Comments on the main part of the manuscript:

5. Page 3, lines 109 to 115: the first of the five enumerated results in the abstract is that 70 to 85% of the total ozone quadratic error is may be attributed to fluctuations associated with time scales > 1.5 days in duration. Here, the sections chosen for analysis are described as being those which cluster closely for time scales of > 1.5 days in duration. The regions for analysis have thus been selected based on the level of similarity in that time scale range, so perhaps the result that most of the error lies within that time scale is not surprising –it does however raise the issue of whether the finding is broadly applicable, or is the result of this selection procedure. If all stations are included in the analysis, rather than the subregions indicated in Figure 1, is finding (i) from the abstract still true? Can the authors provide an argument regarding why the finding is more broadly significant beyond a possible selection bias in how points were selected for comparisons?

We have complemented Table 1 with the MSE for the network-averaged ozone, over the entire continental areas and find that the fractions of MSE due to the diurnal component are in line with those of the sub-regions. The spatial associativity of the daily signal is not of real interest. At any station, the daily signal is highly correlated with that of any other station (for ozone), thus we focus on the synoptic time scale that characterises better the heterogeneities of ozone concentration. This subject is covered in Solazzo and Galmarini (2015).

6. Line 128: I think the writing might be a bit unclear – it sounds like the model values being subjected to a two-hour average when the obs are a one-hour average? Might be better to use "The model concentrations were assumed to be linear between the instantaneous on-the-hour reporting times; the integration (average) between those times was used to construct hour starting (or ending) values in order to more directly compare to the averaging used in the observations." I suspect that what is meant in the sentence, please confirm and rewrite for clarity.

Revised as suggested

7. Lines 132 to 133: Why were the data spatially averaged? This needs to be discussed: how does this aid in or improve the analysis? This will tend to smooth out local sources of short term variations, which will reinforce the tendency of the resulting data to have more error associated with the longer time scales (point (i) issue noted above) – are there other reasons why this is done?

We believe that presenting the hourly results (as originally produced by the models) for three sub-regions is the best compromise for drawing conclusions about the time scale of the error, that remains one of the main scopes of our investigation. The spatial average is done, primarily, to simplify the exposure and discussion of the massive amount of results. Since the first phase of AQMEII, we consider it a good compromise for summarising the results of a multi-model comparison. The alternative would have been to show the spatial distribution of some time-integrated metrics by means of maps, but in that case the time variations would be sacrificed. On the other hand, individual models (CMAQ and Chimere) are extensively evaluated in a number of publications.

8. Line 136: "Missing values have not been imputed" – please clarify: if you are spatially averaging data at each hour, does that not mean that missing data are being imputed spatially?

We have clarified in the text that the missing values in the time series have not been imputed temporally, i.e. before the space aggregation.

9. Line 145: shouldn't that be "BEIS"?

Done

10. Lines 142 to 152: somewhere, mention the extent of the model domain in each case (or add that to Figure 1 as an outline, perhaps?).

We have added the extensions of the domains in the revised text (section 2.1).

11. Lines 161 to 165: why was the same lateral boundary condition not used for both domains? Please include an explanation. A zero or non-zero value may have a large impact on the model results; the response of either model O3 may not be linear with respect to the magnitude of the boundary ozone value chosen. For example, a model initialized with 35 ppbv O3 would have more OH on the model inflow boundaries than a 0 ppbv boundary condition. Were any tests conducted showing the potential impact of the choice of boundary condition level on the simulations?

The scope of this study is to help the diagnostic of the error of two widely used modelling systems. In particular we do not wish to compare the two modelling systems one against the other, primarily because they are applied to different continental areas and subject to different emissions, meteorology and compared against different observational networks. Therefore, the response of each model to the changing conditions needs to be evaluated in isolation.

For convenience of the discussion and exposure of the results the two models have been tested to similar changes, when possible. For the boundary conditions, the intent was to test the response of the models in the situation when no ozone enters from the boundaries (corresponding to the case where all ozone formation stems from emissions). The reviewer is right in commenting that a constant value other than zero at the boundary makes, potentially, a large difference in the resulting ozone levels. When the modeling groups participating in this study first discussed the necessary sensitivity simulations, the CMAQ group encountered difficulties in simulating a case with zero ozone boundary conditions and instead performed a simulation with a constant boundary condition of 35 ppb, roughly representative of summertime background values. This choice was communicated to the Chimere group who opted to set the boundary conditions to the same constant value in their simulation. The CMAQ group subsequently solved their difficulties in simulating the zero boundary condition case but the Chimere group did not re-run their sensitivity simulation. Since the CMAQ zero boundary condition case matches the original intent of this sensitivity, we chose to use it in this analysis rather than using the constant 35 ppb ozone boundary condition case for CMAQ. Because of this discrepancy, the considerations about the 'const BC' run for Chimere need to be limited to the changes in variability with respect to the base case. We have highlighted these aspects in the revised section 2.2 and 3.2.

12. Line 173: Ah – I had been wondering what had been meant by "total ozone quadratic error" in the abstract. Please change the abstract to use either "mean square error" or "total ozone quadratic error (mean square

error)"; more people are familiar with that term than total ozone quadratic error, and the terms have not been introduced in the abstract.

'total' referred to the undecomposed time series, in contrast with the error of the decomposed time series. We find that 'total' might be more direct and easier to understand than 'undecomposed', but have reworded the text nonetheless. 'Quadratic' and 'square' have the same meaning and should not cause any confusion (specified at the beginning of section 2.3)

13. Section 3.1: The difference between North American and EU responses to zero emissions is very interesting. A few words describing the flow field in the relevant months in each case would be worthwhile (e.g. I'm assuming that the spring maximum in North America is likely due to mixing events over the Rockies; trop-strat folds and the like –are the EU O3 highs for zero emissions similarly a reflection of transport, or are this local formation)?

The springtime peak for the zero emissions case over NA is consistent with the springtime peak in northern hemispheric background ozone and the predominant westerly and north-westerly inflow into the NA domain. The background ozone springtime peak is indeed related to more active tropospheric/stratospheric exchange processes during that season. We also note that the summer maximum of the scenario with zeroed anthropogenic emissions is also driven by the emission of biogenic species.

14. Figures 2 and 3: modify the captions: "Average monthly (right column of panels) and diurnal (left column of panels) curves constructed. . ."

Revised as suggested

15. Lines 208 – 225: The authors mention the work of Appel here as a possible explanation for the more rapid decrease in ozone concentrations in NA1 and NA2. The timing seems less off in NA3, EU1 and EU2, but again the model leads the observations in EU3. So, the question: why is this effect more prominent in some areas rather than others? For example, one possibility is that the stomatal conductance and heat capacity values are more incorrect in regions NA1, NA2 and EU3 than elsewhere, if one assumes that the main cause of the problem is related to those two terms. Is there evidence to suggest that this might be the case (e.g. did Appel describe vegetation types where these differences would be the most noticeable, and did those correspond to the same regions with the largest differences)?

Not only the stomatal conductance and heat capacity are likely to play a role in the timing of the ozone profiles, but also the atmospheric stability and the ability of the model to deal with thermally stable and/or windless conditions. For example, we have shown in previous works that Chimere (as well as the majority of the models participating in AQMEII) has problems in dealing with the winter PM in EU3 and that the modelled surface wind speed is biased high. Separate analysis by the CMAQ group (not included in the manuscript) showed that the effects of the model updates on the ozone diurnal cycle, in particular the evening transition period, were more pronounced in NA1 and NA2 compared to NA3. Further reasons are subject of ongoing and future investigations.

16. Section 3.1: How does this rule out other possible causes of diurnal error in the simulations?

The question is not clear, the section 3.1 is mostly a description of the aggregated profiles. If the reviewer refers to the sentence about the CMAQ updates to the stomatal conductance function and the heat capacity for vegetation, we have reworded the text to reflect that those updates could be only one of the causes of daily model error.

17. Section 3.2: This section was very interesting – I'm wondering if the authors can suggest possible reasons for their results. For example, the NA3 subregion had little impact of emissions on night-time model performance for total MSE, but daytime performance showed a more significant impact. I'm wondering, for example, if there are sufficient natural emissions of NO in the San Joaquin valley to titrate the available ozone, hence little impact?

The different behaviour for the night-time ozone errors in NA3 vs. NA1 and NA2 is indeed interesting but we do not have a clear explanation for its cause. A higher amount of natural NO emissions in that area that may dampen the impact of changed anthropogenic NO emissions might indeed be a reason, especially considering that the monitors selected for analysis are 'rural' and thus located in closer proximity of such natural rather than anthropogenic sources. Given that night-time ozone is particularly sensitive to local-scale titration, the proximity of the stations relative to anthropogenic emission sources may actually be the driving factor,

regardless of the level of natural emissions, but without performing additional analysis we feel that this is speculation at this point and did not include such speculation in the revised manuscript.

18. Lines 260 – 272: Implications, please? The anthropogenic emissions affect the bias, variance and covariance of ozone formation. This seems to have taken a back seat to PBL as a source of error elsewhere in the paper.
We have clarified some possible implications in the revised text, consistent with our reply to the first general comment about the role of PBL in causing the error. The main point is that most of the error lies in the base line fluctuations; about 20% lies in the diurnal or faster fluctuations. Of this latter portion, we have proven (mainly by excluding other processes) the role of PBL dynamics ('PBL' here embraces all processes contributing to PBL growth and collapse, including radiation, and that we are unable to study in isolation). We cannot rule out (and cannot quantify exactly either) that part of the daily error is also due to emissions and/or chemistry. There is however indirect evidence that the leading cause of daily error is related to the representation of PBL dynamics (see reply to comment #1 and #41).

19. Line 267-268: The absence of a variance error in the base case tells us that the standard deviation magnitude is the same for both base case and observations – the authors ascribe this to the emissions being of the correct intensity, but the reasoning for this conclusion is unclear – please explain. For example, if two time series are identical aside from a bias offset at every time, they will have the same net standard deviation from the mean, and much of the error in the zero emissions case is in the bias term – which is what Figure 4 seems to show. That to me implies that the temporal/spatial distribution of the emissions, potentially coupled with the variability due to the meteorology, results in the two time series having the same standard deviation –but not that the emissions have the right intensity (I'm interpreting intensity here as magnitude). Maybe I should ask, "What do the authors mean by 'intensity', in this context, and how specifically does this explain the absence of a variance error?"
To be more cautious, we have reworded as suggested: "… is due to the correct interplay between the temporal/spatial distribution of the emissions, potentially coupled with the variability due to the meteorology."

20. Lines 273 to 276: yes, this makes sense; the boundary condition being a constant in this case so only the bias should be affected. However, you'd want to be careful about assuming that the behavior of the error would be the same with a time-varying boundary condition, which would presumably have variance and covariance errors associated with how well that boundary condition actually captures the inflow conditions.
We agree it is not straightforward to derive conclusions about the nature of the error from this sensitivity analysis. However, if the time-varying boundary conditions of the base case affect the variance of the ozone field, we should expect a variance error when the boundary conditions are set to be constant over time. What we observe, however, is mainly an offsetting bias with little impact on the variability (with the exception of NA2 daylight summer, 3[rd] panel of Figure 4).

21. Lines 277 to 285: Interesting: the bias is the dominant part of the deposition effect, which is what I would expect. The covariance term contributes at a few orders of magnitude lower – why? Spatial variability of the deposition field due to changes in ground cover, responses to meteorology of the resistance terms, e.g. temporal variation of the stomatal size? One implication is that getting the deposition magnitude "right" is more important than, e.g., the time dependence of surface resistance. Worth mentioning in the abstract/conclusions?
We were also surprised by the large deposition effect. This has motivated us to initiate analyses to further study the impacts of different approaches to representing deposition on model performance. Based on this initial investigation, it seems that the deposition has a profound impact on the bias (as expected) and on the variability of the signal in NA2 and NA3 (although much lower than the bias part), while only a limited effect on the timing of the signal. The reasons, so far, are unclear. As suggested by the reviewer, we have added some further considerations in the revised text (section 3.2).

22. Line 299: Please clarify this a bit, viz: "resemble those of the daylight base case (Figure 6a, left column), but reduced in magnitude during winter. . ."
Revised as suggested

23. Figure 8 and 9 and similar figures: a general comment: These figures would benefit from a common (perhaps logarithmic) colour scale being used, a scale based on something other than an even division from maximum to minimum in each figure alone (e.g. -80, -50, -30, -10, -8, -5, -3, -1, 0, 1, 3, 5, 8, 10, 30, 50, 80, 100,

300), The graphics quality is poor (images are very fuzzy in the version provided to reviewers). It's difficult to cross compare the different panels to get a feeling for the relative magnitude of the changes in each case, due to these combined issues.

Figures have been modified as suggested

24. There are two places where the figure ordering doesn't follow the standard convention of numbering the figures and presenting them in the order in which they are discussed in the text, this being the first. This should have been addressed prior to submission. Given that Figures 4, 6 and 8 are described together in the text, followed by Figures 5, 7, and 9 – they should be reordered and numbered in the order in which they are discussed.

The figures have been grouped by the type of analysis, while the discussion is presented by continental area, all in the same section. As long as the references in the text are clear we see no reason why the order of the figures must align with the text. We have checked with the editorial team of ACP, and that is not an editorial requirement. We don't believe it will cause confusions to the readers. We'd also like to note that Figures 4 – 9 are first introduced sequentially in the first two paragraphs of section 3.2 before the more detailed discussion of the figures is broken down by continent.

25. Lines 308 to 310: what is the (potential) physical significance of the differences between the different regions? Much more of the error seems associated with the covariance than with the NA simulation. Assuming that the two meteorological models are approximately the same in accuracy and timing of events, would this imply that the Chimere model has more issues with the timing of emissions than with the magnitudes?

Meteorological conditions are also different between the two areas and conditions that are most critical to a model (i.e. more difficult for a model to 'get right') might be more frequent over Europe than over North America (e.g. stagnant conditions), but this is just a speculation. We have no basis to argue that the timing of the emissions are more of an issue in Europe than in North America (see reply to next comment).

26. Line 311: "Removing the anthropogenic emissions had almost no effect on the covariance share of the MSE". This might be suggested by Figure 5, though since the scale is logarithmic the differences may not appear large. But Figure 7(a) compared to Figure 7(c) implies that the share of the covariance has decreased significantly for the zero emissions case, going from less than 10% for the continent, for the base case mean ozone, to over 75% for the zero emissions mean ozone. I agree that the variance portion is unchanged, though. Please correct or clarify the statement starting on line 311.

The share of the covariance has decreased because the share of bias has increased. The correlation coefficient staying the same is in fact a further indication that the covariance is not heavily affected by the zeroing of anthropogenic emissions. We report hereafter the error decomposition for the 'base case' and the 'zero Emi' case (legend as in second panel). These figures show the same values as figure 5 of the manuscript in linear scale. The covariance has indeed little variation.

[Figure]

27. Line 327-328: Another reason to wonder why choose 35 ppbv as the boundary condition for one model simulation and 0 for the other. It's unlikely that a model would use 0 as its boundary condition. . . it would have been better to use a constant value for CMAQ, perhaps chosen based on upwind observations. Why was 35 ppbv chosen for Chimere, and zero for CMAQ?
See reply to comment #11.

28. Line 333-334: Interesting. This implies that the variability of the boundary conditions themselves become more important in winter, as well.
That seems to be the case. We have added a sentence of that effect in the revised text

29. Line 335: RMSE or MSE? Please use consistent terminology throughout the paper. I think this should be MSE in this section.
RMSE, the root of the MSE. We have introduced it at the beginning of section 3.2

30. Line 336: ". . . and bias error during the night-time (Figure 5e)."
Revised as suggested

31. Line 341: Clarify with an additional reference to the figure demonstrating the point being made, viz. "a drastically reduced covariance error compared to the mean ozone (Figure 7a); the timing error is now shifted."
Revised as suggested

32. Figure 6e versus Figure 7e. Why is a much larger portion of the daily maximum 8hr ozone MSE for Chimere in variance, while it's almost non-existent in CMAQ? I suspect that this may represent issues with the timing of the emissions in the European domain.
Variance and covariance errors have more share of the total MSE for the DM8hr in Europe than in North America also because the bias share in Europe is smaller. It is however evident that the covariance error is larger in Europe. In light of our reply to comments #26, the timing of the emissions might not be taken as the main responsible for this. Reasons need to be sought in other processes, like radiation or PBL dynamics.

33. It would be useful for the authors to state the values of sigma-m and sigma-o for both models for 8hr daily max O3. My guess here is that the Chimere value of ïA¸s¸o is larger than the European value of sigma-m, which would indicate a larger variability of conditions in the observations than in the model (given that I'm assuming the meteorological variability is likely to be the same, this makes the emissions/AQ model variability the likely culprit). Actually stating the values of sigma-o and sigma-m would help. It would be good to compare Figure 6 e and Figure 7e in this regard, too, in the text, since they are so very different, and to provide some possible explanation for these differences.
The question is not entirely clear. We have included $\sigma_m$ and $\sigma_o$ values explicitly in the revised text. As pointed out in the reply to comment 11, the scope here is not to compare the two models, but we can learn from their difference, as suggested by the reviewer here. We have added some considerations in the revised section 3.2.

34. Figure 9: all figures look alike to casual inspection, since the colour scales differ in extent. See the comment above regarding Figures 8 and 9 and the figures of this type in the text.
Figures have been modified

35. Lines 348 to 352: Would the authors be able to explain why the impact of deposition might be so different between the two models? E.g. perhaps a paragraph in which how deposition is implemented at the code level might explain this. Are both models using deposition as a flux boundary condition on the vertical diffusion equation, for example, or is one electing to remove the mass from the lowest model layer? Are the magnitude of the deposition fluxes larger in Chimere than in CMAQ (e.g. compared the total O3 deposition from the two models)?
A short description has been added in the revised manuscript. *"In CMAQ, dry deposition is used as a flux boundary condition for the vertical diffusion equation. A review of CMAQ dry deposition model as well as other approaches is provided in Pleim and Ran (2011). In CHIMERE the dry deposition process is described through a resistance analogy (Wesely (1989)). For each model species, three resistances are estimated: the aerodynamical resistance, the resistance to diffusivity near the ground and the surface resistance. For particles, the settling velocity is added. More information is included in Menut et al. (2013)."* Comparison among the

deposition fluxes calculated by the two models is not informative in such different contexts. As we outline in response to comment #21, we have initiate analyses to further study the impacts of different approaches to representing deposition on model performance.

36. Section 4.1 and Figures 10,11: This is interesting, but I'm not sure of the relevance to one of the main issues with regards to correctly predicting air pollution: accurate prediction of high concentration ozone episodes, which are usually of a few days duration. I think this may need a bit of rewriting to focus on the implications of the analysis towards these episodes. For example, having more energy in the differences in the longer time scales implies that boundary conditions are important again (line 377); the analysis shows that much of the energy in the differences between model and obs is associated with longer time periods; whole year, 1 to 2 months, etc. However, the analysis elsewhere in the paper suggests that boundary conditions are more important only during the winter. I'm not sure, based on the discussion, what this adds to the overall analysis. A suggestion: can you expand the y-axis to focus on time scales of hourly to synoptic (e.g. 5 days), perhaps as a second column of panels in Figures 10 and 11, modify the colour scale accordingly, and focus the discussion on that part of the power spectrum? It also might help if the x-axes of Figures 10 and 11 were done in months of the year. The key question to be discussed in this section should be the extent to which this analysis can suggest deficiencies in the models on the time scales associated with ozone episodes. E.g. an expansion of lines 386 to 391, with less of a focus on the longer time scales, unless the authors can make an argument that the longer time scales are in some way influencing high ozone days, and add more to the issue of boundary conditions than has already been discussed.

In the introduction we laid out the main scope of the paper, i.e. '…*to move towards tools devised to enable diagnostic interpretation*' of model error. In the case of the wavelet analysis of the ozone error, we propose an innovative, non-parametric, analysis that can provide insights into the time scale of the error, its intensity and time scale. To the best of our knoweldge, wavelet analysis has not been applied before to help diagnose these aspects of the error of regional air quality models. Being the only non-parametric analysis of the paper, it helps boost the confidence in the results achieved by applying the other, parametric, techniques (regression, ACF, MSE decomposition). Perhaps the usefulness of this technique is not immediately evident here due to the too many degrees of freedom of the models and to our limited knowlwdge of the majority of the variables causing the ozone error, but we believe it can become a resource for model testing and development under more controlled conditions. The use of wavelet of error to detect model deficiencies in reproducing episodes is certainly feasible but it is beyond the scopes of the current study and would require a dedicated paper (also considering that one of the reviewers suggests to simplify the discussion that is already too dense).

That said, the reasons we propose to explain the high energies of the spectrum are just speculations. Since we are not in a position to control each variable/process responsible for the error of ozone independently, all of the explanations we offer are conjectures of the possible causes of model errors. The zeroing of the boundary conditions has effects throughout the year (Figure 2) although the error of the models is most sensitive to variation of the conditions at the boundary during winter. We see no contraddiction there.

Months are provided in the upper part of each panel.

In the supplementary material, we have provided plots with enlarged y-axis as suggested (Figures S7 and S8), and added some considerations in the revised text (section 3.2).

37. Line 393: How well can a model with the relatively low resolution of Chimere in these simulations (25x18 km) be able to capture the daily variation in upslope/downslope winds in the Alps/Po valley – and have there been any higher resolution simulations in the vicinity of EU3 which have better comparisons to observations than attempted here?

The Po Valley area is known to present substantial challenges for air quality models, see e.g. for instance the POMI dedicated multi-model assessment (https://link.springer.com/article/10.1007/s11869-013-0211-1). The area, located just south of the Alps, is not characterized by a strong topography, therefore the limitations are generally not attributed to orographic winds but rather to the prevailing stagnant meteorological conditions with strong thermal inversions. We are not aware of high resolution simulations with better comparisons to observations in the region.

38. Line 398: yes, fold events might be an issue as well – or their combination with upslope/downslope winds. The ozone high in the springtime in North America has been linked to Troposphere/Stratosphere fold events as well, and the events tend to be better captured at higher resolution (though some low resolution models include parameterizations for the events). Transport of fold events to the surface sometimes requires coupling

with smaller scale processes such as convection or upslope/downslope winds; the resolution may thus play a factor in this issue.
Thanks for the remark.

39. The second place where the figure ordering does not match the order of appearance in the text occurs at this point in the analysis. This is disturbing in a multi-author paper which has seen one level of pre-peer-review at the ACPD stage and presumably peer review in some of the writers' institutions; the figures should be ordered and numbered according to their appearance in the text. Figures 12, 13 and 14 appear after Figures 15 and 16 in the text – please renumber the figures to appear in the sequence they appear in the text, and reorder and re-caption the figures accordingly. Perhaps these were last minute changes to the manuscript?
Yes, this was done as a last minute change made by the first author, it has been corrected.

Section 4.2, 4.3:
40. Line 413: "explains" is too generous here. "The majority of the variance is accounted for in the 24 hour lag autocorrelation" might be a better way to phrase this.
Rephrased as suggested

41. This sort of analysis identifies the time scales associated with the model error – which is a very useful thing to do – but does not describe the causes of those errors. There are a number of variables listed (lines 414-417) which have been known to affect ozone formation for several decades, many of which are interlinked; e.g. the incoming shortwave radiation may be affected by the cloud cover in turn affected by the relative humidity, etc.. The variables are not independent. The key finding of these sections is that a 24 hour (or multiple of 24 hour) periodicity exists within the ozone errors. A similar periodicity is found for the wind speed and temperature errors – but the authors imply a causal link to the meteorological modules (line 433-434), which I don't think is justified based on their analysis ("correlation /= causation"). Here's two alternative possibilities (neither of which may be the "actual" cause, my point here is that unless there's direct evidence, the linkage suggested is hypothesis rather than the result of analysis):
(1) Suppose the diurnal temporal allocation of the emissions data is incorrect, so that the model emissions peak at the wrong time of day – this would result in a diurnal error variation which has nothing to do with the meteorology. The authors later acknowledge this possibility later in section 4.3, line 483 – but in the Conclusions (lines 605 to 609) focus only on the PBL possibility – where really, it could be either, or something else altogether. The only solid conclusion is that the error does have that periodicity – which is very valuable information for model developers! However, the evidence for PBL being the main culprit is not clear from the analysis presented here.
Surface wind speed, surface temperature, 'base case' ozone, and 'zero emi' ozone, primary pollutants, all reveal the same daily periodicity. To us, that implies that the problem (or one of the problems) lies in the dynamics of the PBL and/or radiation. All of these results, on the other hand, do not allow us to screen out potential contributions from emission and/or chemistry errors (we would need the equivalent of the 'zero PBL' run to prove it!), but do prove that one of the causes of the daily error is the in the dynamics of the PBL and/or radiation (see discussion in new section 5).
Please also consider the difference between the time shift of the base case and the zeroed emission scenario:

[Figure]

time shift "base" - time shift "zero Emi"

This plot shows the difference (in hour) at each receptor (the number of receptors is reported at the top) between the time shift of the base and the 'zero Emi' case. The median value over a large number of receptors is zero in all cases (for EU the median equals the mean, confirming our previous argument that the covariance error stays the same, see reply to comment #26), with the only exception of winter in NA, for which the median is -1. Our conclusion is that the timing of the emissions is not responsible for (or contributes very little to) the daily error we see (also in line with the findings of Menut et al., 2012: Impact of realistic hourly emissions profiles on modelled air pollutants concentrations, Atmospheric Environment, pp. 233-244, where the timing of anthropogenic emission was found to matter mainly for morning/evening traffic peaks, while ozone was much better synchronised with photochemistry, which is also synchronised with biogenic emissions). The plot above has been added to the supplementary material (figure S15) and discussed in the revised section 4.3.

(2) Suppose that the actual radiation reaching the surface was attenuated more than in the model, in a fashion where the amount of attenuation depends on the solar zenith angle? This would also result in a diurnal signal to the error (and possibly influence some of the other meteorological processes noted as well). And again have little to do with the PBL.
Yes, radiation is indeed a process we encompass in the term 'PBL dynamics', due to our limitation in separating the forcing from the process. We have now clarified in the revised text.

The authors go on to link the analysis (which only shows the temporal nature of the errors) with the PBL dynamics in both models, and present that link as one of the conclusions of the paper (line 605-609) and in the abstract. This link is not justified by their analysis, which identifies time scales only, not specific processes. The suggestion that the issue observed in the current work has been addressed in a more recent version of CMAQ (line 436) is unfounded – unless the authors make use of that specific version of CMAQ and repeat their analysis, and find that the error is completely removed. The conclusions and the 4.2/4.3/4.4 analysis need the text modified to make it clear that there are, based on the analysis carried out thus far, multiple possibilities for the time offset noted in the errors (e.g. modifying their text to remove phrases like "likely" with regards to the cause of the errors, using instead, "possibly, subject to further investigation"). Alternatively, they should redo the analysis (if not for the whole period, for a shorter test period in order to better make their case) with the new version of CMAQ mentioned, and/or modify Chimere in order to test out the hypothesis that the PBL parameterization.
One misunderstanding here is the meaning we assign to 'PBL', which is not simply 'PBL height' but rather the whole chain of dynamical forcing and processes that govern the PBL formation and collapse including radiation. We acknowledge it encompasses a wide range of processes, but reflects what the reviewer noted, i.e. that we are unable to treat each process independently. That said, we believe that all possibilities mentioned by the reviewer can be right (in turn or altogether, and also including the photolysis cycle), and we have indeed mentioned it clearly in the revised manuscript (section 4.2 and 5). However, the error structure of the zeroed-emission case having the same periodicity as the base case provides a 'causal' link between ozone error and daily processes in the model other than anthropogenic emissions. To us, the only remaining, plausible process is the PBL (in the inclusive sense outlined above). We are also convinced that the problem

cannot possibly lie with only one or two processes given the high level of non-linear interdependence of boundary layer dynamic from radiation, surface description, surface energy balance, heat exchange processes, development or suppression of convection, shear generated turbulence, entrainment and detrainment processes at the boundary layer top for heat and any other scalar. Most likely it could depend from imprecisions or errors in all of these or some with repercussions on the all PBL system.

42.    The authors mention that the European measurement data themselves may have timing errors, referencing an earlier paper (lines 480 to 481), as a possible explanation for the grouping by country of the model errors which can be seen in Figure 13(a). Explain why timing issues in the observations doesn't invalidate or weaken the analyses in this section. This is a serious issue – the authors have made an effort to ensure that the model values are specific and representative of specific hours – but if the measurement data time stamps can't be trusted, that adds additional uncertainty to the analysis and its conclusions. For example, if the time series in the observations are all off by 2hours in some parts of the domain, would that not show up as an "error" signal with a strong diurnal variation? I also wonder about time zone errors in reporting (local time, standard time, daylight savings time, etc.).

The likely issue with the AIRBASE European monitoring stations is not (or not only) the timing, since all measures are reported hourly, but rather the lack of harmonisation in the measurements methods and sampling protocol. We noted the same issue when analysed the associativity of the diurnal signal of the Canadian and US networks. We have removed the word 'timing' to avoid confusion.
Yes, it is a further source of uncertainty, but what is the alternative?
All observational and modelling data can be retrieved in local standard time from the ENSEMBLE system, which is the service provider of the AQMEII datasets.

43. Line 480: harmonization misspelled.
Corrected, thanks

44.    Section 4.4:
Line 518-519: The authors need to explain why the data were not limited to co-located measurements of air pollutants and meteorology, in this section's analysis. The authors are averaging different sets of locations between air pollutants and meteorology – and making the assumption that the resulting averages will both represent the same region to the same degree. Worst case scenario, for example: air pollution measurements in valley bottoms and meteorological measurements on mountain-tops.
The analyses have been repeated and restricted to stations in a horizontal radius of 1000 m and maximum vertical displacement of 250m. The resulting number of stations is of 61 in Europe and of 45 in North America. The results are now discussed for the continental areas only.

45. Line 525: The authors use the phrase "overwhelming daily memory of the error" –please explain. The term implies that the results of day 1 are in some way affecting day 2 – but the analysis thus far has dealt with periodicity, which does not necessarily imply a causal link between one day and the next.
The comment here is not entirely clear. The analysis reveals that the error of day $d$ at hour $h$ is highly correlated with the error at $h \pm 24d$, with $h$ in [1,24] and $d$ in [1,4] (and even higher values), based on the PACF analysis. Given that the PACF peaks periodically each 24 hours and that the values in between are small/non-significant, the link is not causal (the error at hour $h$ DOES NOT affect the error at $h$+24, it is 'just' highly correlated) but rather the source of the error is the same and has a 24 hours periodicity. Either causal or casual, the daily periodicity is striking.

46. Line 531-532: At what point would these collinearities invalidate the analysis?
Good question, but we do not have the answer. Maybe some analytical threshold for significance or some 'rules of thumb' have been established elsewhere, but we are not aware of it. We have provided the table where the co-linearities are quantified   to help the readers drawing informed conclusions.

47. Line 538-540: The authors again attribute the errors to PBL issues, without sufficient cause. All that can safely be concluded by the analysis thus far is that the errors have a significant diurnal component, and that the PBL parameterization is one of many possible contributors to that diurnal component.
See replies to comments 1, 41.

48. Lines 544-546 and lines 554-558: Filtering out the shorter time-scales removes much of the collinearity of the fields examined – which says that those fields are interrelated for the shorter time scales more than the longer time scales. This does not imply that errors with CMAQ are associated with longer time scales, however (line 555) – the reasoning behind this statement made by the authors is not clear.

The comment there refers to the 'no-DU' case, for which no high associativity has been found with the available error fields. Thus, our point is that 1) the causes of the error are others and 2) these other causes need to have a time scale longer than 1.5 days (because we refer to the 'no-Du case'). We have clarified this in the revised text.

49. Line 566: Anything with a diurnally repeating signal in the error could be the root cause here.

Point taken

50. Conclusions: The authors state (line 577) that other methods of analysis do not help target the causes of model error. I'm not completely sure that the work they have presented does a better job – the main contribution (and a very valuable one!) is that the time scales of the errors may be identified with the methodologies the authors present. However, the causes of those errors remain speculation at this point in time, since a causal link between specific processes and the time scale analysis has not been demonstrated. Nevertheless, the analysis methods introduced here are an excellent route by which to reach that endpoint, and worthy of publication in ACP on that basis alone. Coupled with process analysis (their suggestion for further work), it may be possible to tease out the causes of the errors: e.g. output the change in concentration due to each operator at each time step, and analyzing the rates of change of the different components.

We have modified the wording in the Conclusions by clearly stating that the final step of linking error to processes has not been achieved and that future analyses need to include sensitivity runs: *"At the current stage, the methods we propose help identify the time scale of the error and its periodicity. The step to link the error to specific processes can only be reached by integrating the analysis with sensitivity model runs. For instance, we can infer that the timing error of the diurnal component is (at least partially) associated to the dynamics of the PBL, but further analyses are necessary to isolate the components of the PBL responsible for that error".*